# Mosaic anatomy in an early fossil squamate

Roger B. J. Benson[1✉], Stig A. Walsh[2], Elizabeth F. Griffiths[3], Zoe T. Kulik[1], Jennifer Botha[4], Vincent Fernandez[5], Jason J. Head[6] & Susan E. Evans[7✉]

Squamates (lizards and snakes) comprise almost 12,000 living species, with wide ecological diversity and a crown group that originated around 190 million years ago[1,2]. Conflict between morphology and molecular phylogenies indicates a complex pattern of anatomical transformations during early squamate evolution, which remains poorly understood owing to the scarcity of early fossil taxa[1,3]. Here we present *Breugnathair elgolensis* gen. et sp. nov., based on a new skeleton from the Middle Jurassic epoch (167 million years ago) of Scotland, which is among the oldest relatively complete fossil squamates. *Breugnathair* is placed in a new family, Parviraptoridae, an enigmatic group with potential importance for snake origins, that was previously known from very incomplete remains. It displays a mosaic of anatomical traits that is not present in living groups, with head and body proportions similar to varanids (monitor lizards) and snake-like features of the teeth and jaws, alongside primitive traits shared with early-diverging groups such as gekkotans. Phylogenetic analyses of multiple datasets return conflicting results, with parviraptorids either as early toxicoferans (and potentially stem snakes) or as stem squamates that convergently evolved snake-like dental and mandibular traits related to feeding. These findings indicate high levels of homoplasy and experimentation during the initial radiation of squamates and highlight the potential importance of convergent morphological transformations during deep evolutionary divergences.

Squamates (lizards and snakes) diverged from their closest living relative, the tuatara (*Sphenodon*), by the early Triassic period[1], and molecular clocks suggest that their crown group originated by the Early Jurassic epoch, around 190 million years ago[2] (Ma). However, confident records of crown- or near-crown squamates are not known before the Middle Jurassic[1,3], and patterns of ecological diversification during early squamate evolution remain poorly understood. There is currently little consensus on the relationships of many early squamate fossils (for example, refs. 1,4–6), and difficulties in resolving the phylogenetic affinities of these specimens are compounded by the large incongruence between morphological and molecular hypotheses of squamate evolution[2].

Parviraptorids are an enigmatic group of extinct, predatory squamates that persisted for more than 20 million years, from the Middle Jurassic–Early Cretaceous epoch of North America and Europe. Recent works have proposed that parviraptorids may be the earliest members of the snake stem lineage[7]. However, they have also been identified at various times as anguimorphs[8], gekkonomorphs[9,10], members of a 'scincomorph'-anguimorph group[11], or potentially as stem squamates[12]. These hypotheses have been difficult to evaluate, because parviraptorid specimens so far are relatively incomplete, leaving important anatomical questions unanswered, including the extent to which bones from across the skeleton share features with snakes or other groups, and whether or not parviraptorids had long, limbless (or limb-reduced) bodies like snakes.

Recent work retained only the most snake-like elements within the parviraptorid hypodigm—primarily tooth-bearing bones and vertebrae[7]. However, earlier works reported other bones that together show a unique combination of snake-like and non-snake-like features[8,13]. If parviraptorids are stem snakes, then this may provide evidence of mosaic evolutionary changes along the snake stem lineage. Alternatively, it may indicate that parviraptorids are not closely related to snakes, and evolved their snake-like features convergently. However, discussion of these evolutionary hypotheses has been eclipsed by concerns that some of the more informative specimens may be chimeric associations of multiple taxa[7] (Supplementary Discussion). Here we report a relatively complete specimen of an early parviraptorid (Fig. 1 and Extended Data Fig.1) that resolves these concerns, adds substantially to knowledge of parviraptorid anatomy, and limits hypotheses of their phylogenetic affinities and significance for early squamate evolution.

## Systematic palaeontology

**Pan-Squamata** Gauthier & de Queiroz, 2020[12]
**Parviraptoridae** new family

**Diagnosis.** Squamates (or stem squamates) with well-developed limbs and a unique combination of primitive and derived traits including paired, shallow, unsculptured parietals enclosing a parietal foramen and possessing a postparietal (posteromedian) process; teeth on all palatal bones; double tooth row on palatine and absence of a choanal fossa; no vomer–maxillary contact; strongly recurved marginal teeth implanted

[1]Division of Paleontology, American Museum of Natural History, New York, NY, USA. [2]Natural Sciences Department, National Museum of Scotland, Edinburgh, UK. [3]Education Services, University of Cambridge, Cambridge, UK. [4]Evolutionary Studies Institute, University of the Witwatersrand and GENUS:DSTI-NRF Centre of Excellence in Palaeosciences, Johannesburg, South Africa. [5]European Synchrotron Radiation Facility, Grenoble, France. [6]Department of Zoology, University of Cambridge, Cambridge, UK. [7]Department of Cell and Developmental Biology, University College London, London, UK. ✉e-mail: rbenson@amnh.org; s.e.evans@ucl.ac.uk

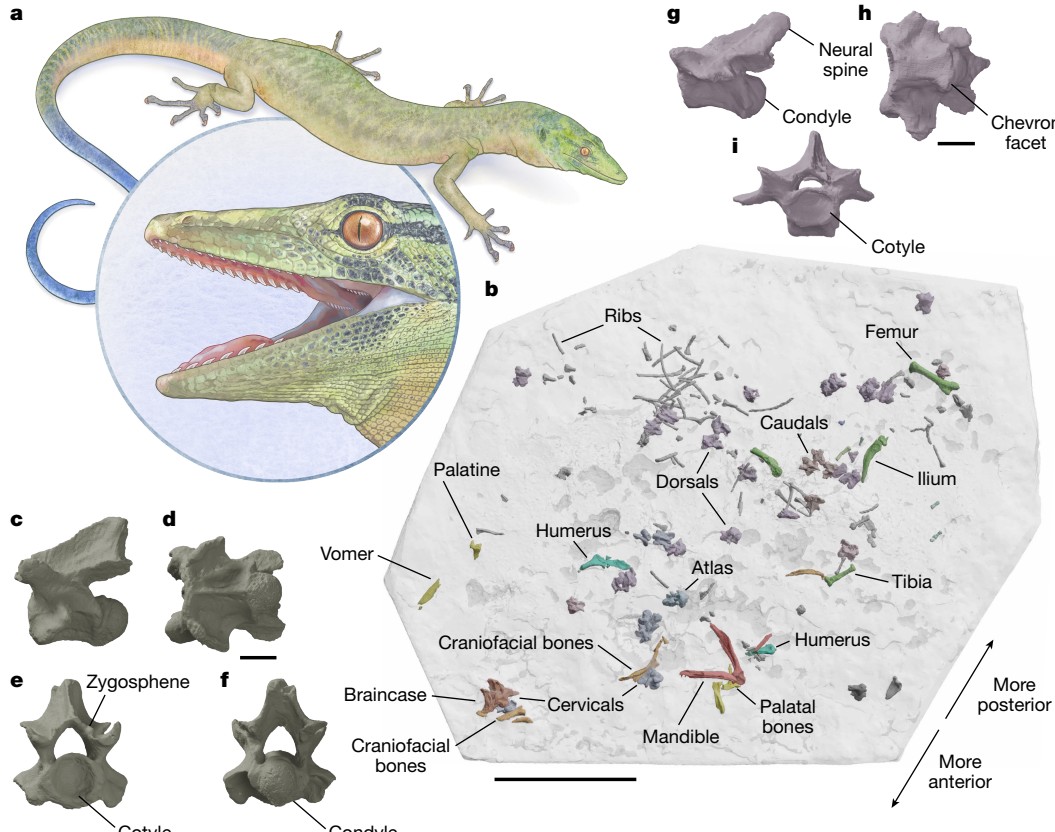

**Fig. 1 | Reconstruction of *Breagnathair elgolensis* from NMS G.2023.7.1.**
**a**, Life reconstruction of *Breugnathair elgolensis* based on measured proportions of NMS G.2023.7.1. **b**, Digital render of the bones as originally preserved in NMS G.2023.7.1, using information from the pilot scan (Supplementary Data 1 and 2). **c–f**, Digital renders of cervical vertebra (CEb in Extended Data Fig. 5) in left lateral (**c**), ventral (**d**), anterior (**e**) and posterior (**f**) views. **g–i**, Caudal vertebra (CAa in Extended Data Fig. 5) in left lateral (**g**), ventral (**h**) and anterior (**i**) views. Scale bars: 50 mm (**b**), 2 mm (**c–i**). Life reconstruction reproduced with permission from Mick Ellison (American Museum of Natural History).

in shallow rounded sockets and separated by interdental ridges; no erosion of mature teeth by replacements; and procoelous vertebrae in adults, with a weakly developed zygosphene–zygantral system.
**Comment**. The term 'parviraptorid' has been used informally in previous works (for example, ref. 7), but has not been erected formally as we do here.
**Included taxa**. *Parviraptor estesi* (type genus; Extended Data Figs. 2 and 3), *Diablophis gilmorei*, *Portugalophis lignites* and *Breugnathair elgolensis* gen. et sp. nov., as well as the holotype and at least some specimens referred to the nomen dubium *Eophis underwoodi*, from the Middle Jurassic of Kirtlington[8] (Extended Data Fig. 4 and Supplementary Discussion).

*Breugnathair elgolensis* gen. et sp. nov.

**Etymology**. *breug-nathair* (brʲiag Nahɪrʲ; adapted from Scottish Gaelic): false snake; specific name comes from the locality of discovery, north of the village of Elgol on the Strathaird Peninsula of the Isle of Skye.
**Holotype**. NMS (National Museums of Scotland, Edinburgh, UK) G.2023.7.1, a disarticulated partial skeleton from the Middle Jurassic (Bathonian, 166 Ma) Kilmaluag Formation of the Elgol Coast SSSI[14], collected in 2015 under permit from NatureScot, with permission from the landowner John Muir Trust.
**Diagnosis**. Parviraptorid that differs from Early Cretaceous *Parviraptor estesi* in having proportionally narrower parietals that bear a nuchal shelf and lack a deep ventral concavity between the base of the postparietal process and the base of the supratemporal process; differs from Late Jurassic *Portugalophis lignites* by having shorter interdental ridges

and more sharply recurved tooth crowns; and differs from Late Jurassic *Diablophis gilmorei* in the less bulbous morphology of tooth bases, and substantially more recurved crowns (Supplementary Discussion).

## Description

NMS G.2023.7.1 is an association of disarticulated bones spread over a diameter of approximately 19 cm in a rostral to caudal pattern, with skull elements and anterior vertebrae at one side of the block, and hindlimb and caudal vertebrae at the opposite edge (Fig. 1b, Extended Data Fig. 5 and Supplementary Data 1 and 2). Assignment to a single individual is well-supported, based on the matching morphologies of paired elements from different parts of the block, including parietals and humeri, as well as the consistent morphology of all preserved vertebrae, the shared morphology of teeth on the dentary, vomer, palatine and pterygoid, and the consistently large relative sizes of bones compared with other squamates from the same locality[3,14] (Methods). We provide high-resolution 3D models of the complete skeletal anatomy derived from X-ray micro-computed tomography (μCT) as well as synchrotron phase-contrast μCT (Methods), available via MorphoSource (Supplementary Data 1). Further anatomical description is provided in the Supplementary Discussion.

The skeleton has lizard-like proportions, with large, well-developed limbs. The limb bone sizes and a presacral vertebral count are within the range of limbed squamates (Fig. 1, Methods, Supplementary Discussion and Supplementary Data 2). Thirty-two vertebrae are preserved, of which 24 are definite presacrals, 3 are possible presacrals, 3 are definite caudals and 2 are possible caudals or sacrals (Extended Data Fig. 5 and

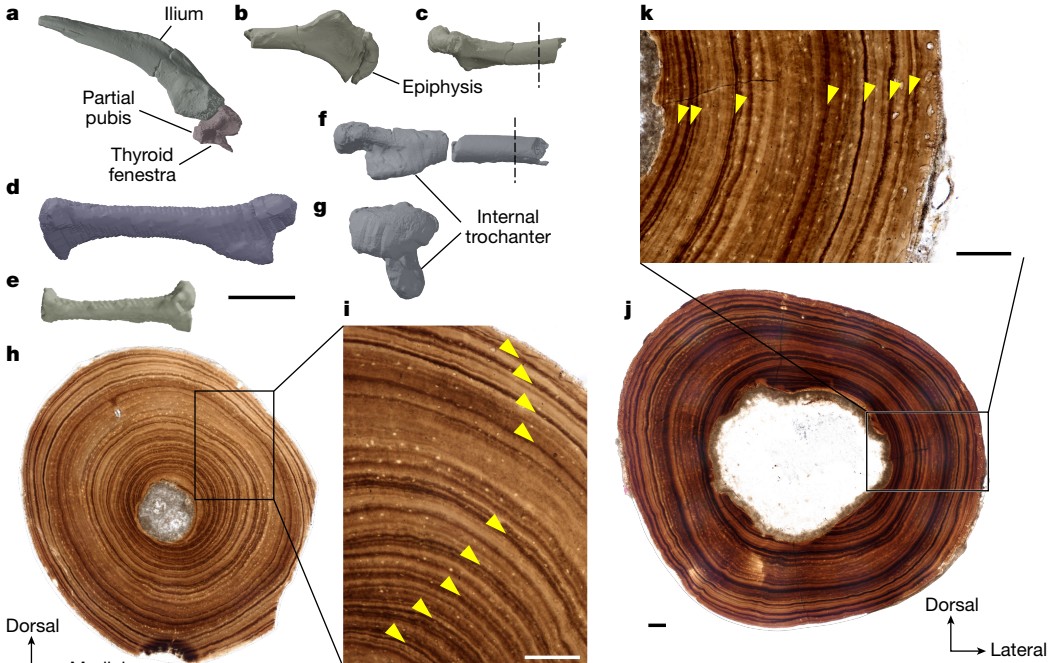

**Fig. 2 | Appendicular morphology and osteohistology of NMS G.2023.7.1.** **a**–**g**, Digital renders of right ilium and partial pubis in lateral view (**a**), proximal part of right humerus in ventral (**b**) and posterior (**c**) views, left femur in posterior view (**d**), right tibia in anterior view (**e**) and proximal part of right femur in posterior (**f**) and proximal (**g**) views. **h,i**, Transverse thin section of the right humerus (**h**; location as indicated in **c**), showing a remarkably thick cortex composed of lamellar bone and nine incremental growth marks indicated by yellow arrowheads (**i**). **j,k**, Transverse thin section of the right femur showing resorbed endosteal edge (**j**; location as indicated in **f**), resulting in a thinner cortex that is similarly composed of lamellar bone with seven growth marks (arrowheads, **k**). Scale bars: 5 mm (**a**–**g**), 100 μm (**h**–**j**).

Supplementary Data 2). This gives a minimum presacral count of 25–28 (including the missing atlas). The preserved materials indicate that at least 7–8 cervicals (6 preserved, plus the atlas and 1 cervicodorsal) and 17–21 dorsals (17 preserved, plus the cervicodorsal and 3 possible dorsals) were present. Even given that some vertebrae are likely to be missing, this suggests that the presacral–precloacal vertebral count was much lower than in limbless squamates (at the lower end, up to 68 in the anguid *Pseudopus*; and more in other taxa – for example, 120–320 in snakes[15]). The limbs and limb girdles are disarticulated and incomplete, but several phalanges are preserved, including an ungual phalanx (Extended Data Fig. 6). Other preserved limb and girdle elements are well-developed including the partial right scapulocoracoid (Extended Data Fig. 6), both humeri, missing their distal portions, the partial right side of the pelvis, left femur, partial right femur, right tibia and possible partial fibula (Fig. 2a–g).

The bone histology of the humerus and femur shows slow, cyclical growth (Fig. 2 and Supplementary Discussion). An almost complete record of primary growth is visible in the humerus, in which nine distinct growth marks interrupt lamellar and parallel-fibred bone, indicating a minimum age of nine years. Cortical remodelling in the femur has erased the earliest growth, recording only six growth marks. Decreasing growth mark spacing toward the outer bone edges suggests that NMS G.2023.7.1 was at or near skeletal maturity. Further evidence of osteological maturity includes the full ossification of the vertebral condyles (Fig. 1) and co-ossification of the surangular, prearticular and articular into a compound bone (Fig. 3), although the humeral epiphyses remain unfused to the shaft, consistent with prolonged continuation of growth (Fig. 2). The prolonged duration of growth and late skeletal maturity resembles that of varanids[16,17].

Preserved cranial bones include parietals, squamosals, a postfrontal, much of the braincase, a jugal, vomer, palatine and pterygoid (Fig. 3 and Extended Data Figs. 6 and 7). A postorbital is not preserved, but its presence is inferred from facets on both the postfrontal and squamosal.

Digital reconstruction suggests long, low cranial dimensions with a proportionally long snout (Fig. 3a,b and Methods), similar to the skull proportions of extant varanids[18], mosasauroids[19] and some gekkotans[20] (for example, *Lialis*[21]). These proportions are also consistent with the long, low maxilla preserved in some other parviraptorids[7,8]. The position of the maxillary facet on the jugal shows that the maxilla terminated anterior to the postorbital bar (Fig. 3 and Extended Data Fig. 6; in contrast to ref. 7), differing from the condition in snakes, including the early stem snakes *Najash* and *Dinilysia*, in which the maxilla extends to the postorbital bar or more posteriorly[22–24].

The parietals share features with early-diverging squamate groups, including the presence of a parietal foramen (Fig. 3c), which is absent in snakes and crown gekkotans, but present in many other reptiles including possible stem gekkotans (for example, ref. 25) and most other squamates. The parietals are proportionally long with flat, unsculptured dorsal surfaces, and are paired, similar to rhynchocephalians, some gekkotans, and xantusiids[18], plus early-diverging fossil squamates such as *Eichstaettisaurus*[25] and *Dalinghosaurus*[26]. This differs from other squamates, including anguimorphs, snakes and iguanians, in which the parietals are fused across the midline[18,27]. A postparietal process is present (Fig. 3 and Extended Data Fig. 6), similar to dibamids, gekkotans and scincoids, plus early-diverging fossil squamates such as *Bellairsia*[3] and *Ardeosaurus*[4]. Parietal features of snakes are absent, including those present in *Najash* and *Dinilysia*, such as ventrolateral laminae that enclose the anterior braincase[23,24,28].

The preserved palatal bones bear teeth that are morphologically similar to, but smaller than, those of the dentary (Fig. 3d,e). The vomerine tooth row is single, whereas the contiguous palatine–pterygoid tooth row is double for most of its length (Fig. 3b and Extended Data Fig. 7). Vomerine teeth are also present in early rhynchocephalians[29], but are absent in almost all crown squamates except the extant anguine *Pseudopus*[30], some Eocene anguimorphs[31] and the possible stem scincoid *Eoscincus*[5]. Palatine teeth are also present in *P. estesi*. These were

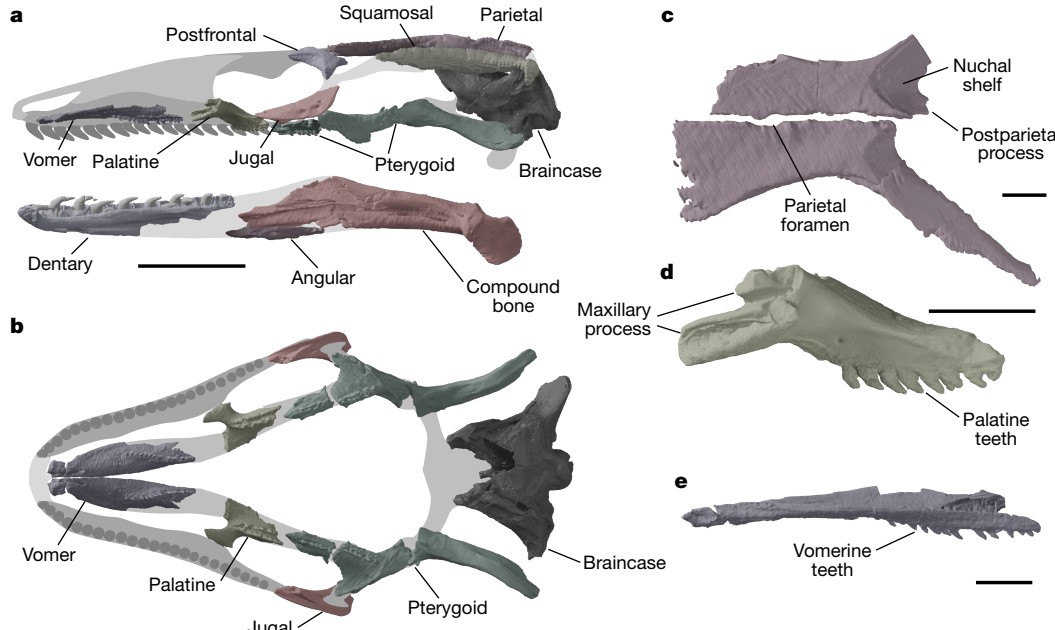

**Fig. 3 | Skull reconstruction and details of cranial bones of NMS G.2023.7.1. a,b**, Digital render of cranium and mandible reconstruction (Methods) in left lateral (cranium) and medial (mandible) views (**a**) and cranium reconstruction in ventral view showing the palate and braincase (**b**). Reconstructed areas are shown with grey shading. **c–e**, Digital renders of parietals in dorsal view (**c**), left palatine in lateral view (**d**) and right vomer in medial view (**e**). Scale bars: 10 mm (**a**,**b**) and 2 mm (**c–e**). Note that the jugal, postfrontal, vomer, palatine and pterygoid have been mirrored to enable reconstruction.

not evident in previous studies but are shown here using computed tomography (CT) scans, confirming that previously reported palatines[8] belong to parviraptorids and not to a distinct squamate group (in contrast to ref. 7; Extended Data Figs. 2 and 3 and Supplementary Data 1). Palatine teeth are also present in stem lepidosaurs, rhynchocephalians and some early-diverging squamates[26,29,32], as well as most snakes and some iguanians and anguimorphs[18,28].

The vomer is proportionally long and narrow and lacks any evidence of a contact with the maxilla between the vomeronasal fossa and fenestra exochoanalis (Fig. 3b and Extended Data Fig. 7), similar to both snakes and early-diverging lepidosaurs[27–29], including the stem squamate *Oculudentavis*[33]. This interpretation is corroborated by the morphology of the maxilla in other parviraptorids, which shows little development of a medial shelf[7,8] (Extended Data Fig. 2). The palatine lacks any development of a choanal fossa and is thus similar to squamate outgroups such as the stem lepidosaur *Marmoretta*[32], rhynchocephalians[27], and the stem squamate *Bellairsia*[3].

The braincase of NMS G.2023.7.1 is similar to those of non-snake squamates[18,28] (Fig. 4b,c and Extended Data Fig. 7). Anteriorly, the prootic possesses a blunt, free-ending alary process above a shallow trigeminal notch. A distinct, convex, tab-like crista prootica, similar to the one in some anguimorphs, scincoids, lacertoids and iguanians is present anterior to the fenestra vestibuli. The fenestra is separated from the lateral aperture of the recessus scala tympani (LRST) by a broad crista interfenestralis. The oto-occipital lacks a well-defined crista tuberalis behind the LRST, unlike the condition in most anguimorphs[18]. The paroccipital process of the oto-occipital is short and deep (Fig. 4b).

The morphology of the prootic and parietal indicates the presence of a membranous anterior braincase as in most lepidosaurs, but unlike all known modern and fossil snakes, as well as head-first burrowers such as amphisbaenians and dibamids, in which the parietal, prootic and basisphenoid form a bony anterior margin[28]. The morphologies of the middle and inner ear (Extended Data Fig. 7) are different from those of all known snakes, in which the neurocranial cristae join to form a complete, or nearly complete, crista circumfenestralis that partially encloses the fenestra vestibuli and LRST as a component of a re-entrant perilymphatic fluid circuit in the ear[34]. Dorsally, the morphology of the supraoccipital–parietal articulation, including the elongate, rod-like processus ascendens (Extended Data Fig. 7), indicates the retention of a flexible metakinetic axis, unlike the rigid articulation in all known snakes[28,35].

The surangular, prearticular and articular are fused to form a compound bone (Fig. 3a), as seen in many other squamates, including snakes, dibamids, xantusiids and amphisbaenians[36], as well as many gekkotans, some scincids, some gymnophthalmids, anguimorphs[18] and the stem squamate *Oculudentavis*[33]. The nature of the joint between the dentary and compound bone is uncertain owing to breakage and the lack of a preserved splenial.

Dentary tooth implantation (Extended Data Fig. 7) is similar to that of many snakes, in which the tooth base sits in a shallow alveolus[37,38], is symmetrical, and has the pulp cavity opening ventrally[39], differing from the pleurodont condition (as in refs. 37,39) of many non-snake squamates and some stem snakes[35]. Replacement pits are absent, as in other parviraptorids including the Late Jurassic *Portugalophis*, which has been interpreted as showing early initiation of internal tooth resorption similar to snakes[39]. A wide subdental gutter is present ventrolingual to the alveoli, as present in many non-squamates but absent in snakes[28] and mosasauroids[40].

The marginal teeth of NMS G.2023.7.1 are conical and strongly recurved (Fig. 3a and Extended Data Fig. 7), similar overall to those of other parviraptorids[7,8] as well as many snakes[7,27], mosasauroids[19], the pygopodid *Lialis*[41], and some living[18,20] (for example, *Anguis fragilis*, *Heloderma* and many *Varanus* species) and extinct[42] anguimorphs.

Vertebrae of NMS G.2023.7.1 are procoelous (Fig. 1c–i and Extended Data Fig. 6), as in other adult parviraptorids[7,8] (although immature vertebrae may retain an open notochordal canal[8]). The cotyle and condyle are rounded in outline, similar to snakes. However, the edges of the condyle are not expanded beyond the margins of the posterior centrum, unlike in stem and crown snakes, teiids, varanoids and some mosasauroids. Instead, the diameter of the condyle is smaller than the posterior centrum, as in scincoids and procoelous gekkotans[15].

The neural spines are mostly triangular in lateral view, with the greatest height posteriorly. Zygosphene–zygantrum articulations are present but simple; zygosphenes are at a distinct angle to the main

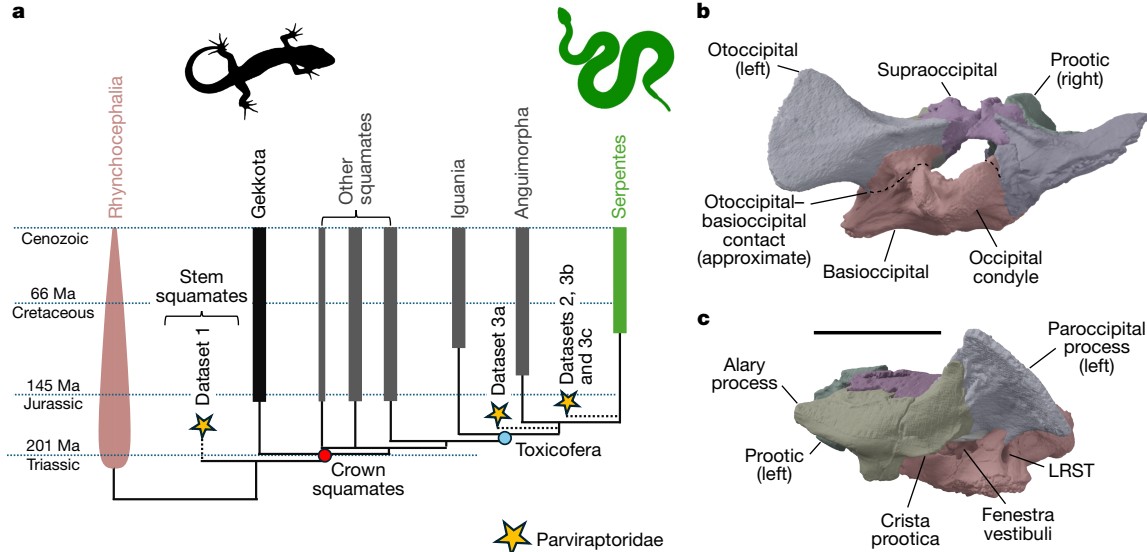

**Fig. 4 | Hypotheses of the phylogenetic affinities of parviraptorids and braincase morphology of NMS G.2023.7.1. a**, Summary of phylogenetic results provided in Extended Data Figs. 8–10, with general topology adopted from Extended Data Fig. 8a (Iguania, Anguimorpha and Serpentes form a polytomy in other analyses). The phylogenetic positions of Parviraptoridae returned by analyses of Datasets 1 (modified from ref. 3), 2 (modified from ref. 6) and 3 (modified from ref. 47) are indicated by stars. Dataset 3a refers to results from Dataset 3 when Anguimorpha and Serpentes are constrained as sister taxa (Extended Data Fig. 9a), Datasets 3b and 3c refer to results when Iguania plus Anguimorpha and Iguania plus Serpentes, respectively, are constrained as sister taxa (Extended Data Figs. 9b and 10). **b,c**, Digital render of the braincase of NMS G.2023.7.1 in posterior (**b**) and left lateral (**c**) views. Scale bar, 5 mm. Silhouettes were obtained from Phylopic (https://www.phylopic.org/) under a CC0 1.0 Universal Public Domain licence: *Gymnodactylus geckoides* created by J. C. Arenas-Monroy (https://www.phylopic.org/images/11729fc4-30d2-40bf-a5a3-fd490a74589e/gymnodactylus-geckoides) and *Crotalus viridis*, created by B. Perry (https://www.phylopic.org/images/b58a7ca7-8d14-4d7b-88a5-f2b05de79c85/crotalus-viridis).

prezygapophyseal surface but remain contiguous with it. This condition is widely distributed among squamates (ref. 27, character 468) but differs from the well-separated and more dorsally placed accessory facets of some other taxa, including teiids, mosasauroids and snakes[15].

Vertebral articulations with dorsal ribs consist of small synapophyses that are laterally oriented and well dorsal to the ventral margin of the centrum in mid-trunk vertebrae. This morphology is similar to many limbed squamates and unlike the condition in snakes, where synapophyses are larger, possess differentiation of articular surfaces into diapophyseal and parapophyseal facets, are ventrolaterally angled, and extend to or below the ventral extent of the centrum[15,43].

Cervical intercentra contact both of their adjacent vertebrae via low-contact surfaces, as in squamates and their outgroups. This differs from the condition in mosasauroids, some anguimorphs, and the early-diverging stem snakes *Najash* and *Dinilysia*[44], in which cervical intercentra articulate entirely or primarily with a distinct hypapophysis (that is, with 'fore-and-aft margins' (ref. 27, character 465)) on the posteroventral surface of the preceding centrum[15,19,45]. The axis shows a three-fold hypapophysis morphology (Extended Data Fig. 6), which is also present in some other early-diverging crown squamate groups, including the dibamids *Anelytropsis* and *Dibamus*, as well as some scincids (for example, *Acontias* and *Melanoseps*)[46].

## Phylogenetic results

We included NMS G.2023.7.1 and the more complete associations of parviraptorid material from the Early Cretaceous of the UK (*Parviraptor estesi* and cf. *Parviraptor estesi*; for which we provide new CT data; Extended Data Figs. 2 and 3 and Supplementary Data 1) in three phylogenetic analyses with different foci (Methods): (1) the early reptile dataset of ref. 3, which includes many snake and non-snake squamates alongside a rich sample of early lepidosaurs and lepidosauromorphs; (2) the squamate dataset of ref. 6, which contains a substantive sample of Jurassic squamates; and (3) the snake-focussed dataset of mainly toxiceferan squamates from ref. 47, with considerable added data from non-toxiceferan squamates and various early squamate fossils. Parviraptorid specimens from the Late Jurassic epoch of the USA (*Diablophis*) and Portugal (*Portugalophis*) were not included in these analyses, pending discovery or description of more comprehensive material. We analysed these datasets using Bayesian inference tip dating (Methods).

Parviraptorids form a clade in all analyses, but we found support for multiple different hypotheses of their relationships to other squamates (Fig. 4a). We find parviraptorids either as stem squamates (Dataset 1; Extended Data Fig. 8a), on the stem of snakes (Dataset 2; Extended Data Fig. 8b; and Dataset 3 when iguanians are constrained as the sister taxon of snakes or anguimorphs; Extended Data Figs. 9b and 10) or on the stem of anguimorphs and snakes (Dataset 3; Extended Data Fig. 9a; when anguimorphs and snakes are constrained as sister taxa). This uncertainty remains despite our considerable effort to resolve it by attempting to include complete samples of characters and taxa relevant to deep squamate divergences.

These conflicting phylogenetic results from different datasets lead to uncertainty about the first appearance date of crown squamates in the fossil record. Molecular clock studies imply an Early Jurassic or Late Triassic age for the squamate crown group[2]. However, except for parviraptorids and the undisputed stem squamate *Bellairsia*[3], almost all Middle Jurassic squamates are known only from disarticulated and limited portions of the skeleton obtained by sieving of bulk sediments, which contain little information on their phylogenetic affinities (including Middle Jurassic specimens that share dental features with the candidate stem scincoid group Paramacellodidae such as cf. Paramacellodidae, from the Elgol Site of Special Scientific Interest[14] (SSSI)). Current analyses therefore suggest that crown squamates are not known with certainty until the Late Jurassic[3,6,45,48–50]. As a candidate stem snake, *Breugnathair* may be the oldest crown toxiceferan. However, phylogenetic uncertainties suggest that this and other parviraptorids should be treated cautiously when choosing fossil calibrations for molecular clock studies.

## Discussion

NMS G.2023.7.1 decisively resolves uncertainties on the anatomy of parviraptorids, refuting previous claims that partial skeletons from the Early Cretaceous Purbeck Limestone Formation of the UK are chimeric associations of bones from multiple, phylogenetically distinct groups[7] (Extended Data Figs. 2 and 3 and Supplementary Discussion). Instead, parviraptorids show a unique combination of traits that is highly distinct from any living group, with snake-like dental morphology alongside plesiomorphies shared with stem squamates, gekkotans or other early-diverging groups, combined with varanid-like overall head and body proportions (a long, low skull without substantial body elongation or limb loss).

Several aspects of parviraptorid anatomy are uniquely shared with early-branching squamates and so are inconsistent with affinities to either stem snakes or Toxicofera more broadly. These include aspects of parietal morphology, such as lack of fusion between contralateral elements and the presence of a prominent postparietal process[8–10,27]. Furthermore, parviraptorids lack the cervical intercentral morphology of stem snakes and some anguimorphs, in which cervical intercentra articulate with prominent cervical hypapophyses primarily on their preceding centrum[15,44]. The absence of a choanal fossa on the palatine differ from all crown-group squamates and the presence of vomerine teeth differs from almost all crown squamates bar some living and extinct anguimorphs[30,31] and *Eoscincus*[5].

Other features of parviraptorids are shared with both snakes and with some stem squamates or early-diverging squamate groups such as gekkotans, and so provide equivocal phylogenetic information. These include: (1) the presence of an anteroposteriorly elongate maxilla and low ascending process of the maxilla, also seen in the stem squamate *Oculudentavis*[33]; (2) the probable lack of contact between the vomer and maxilla anterior to the fenestra exochoanalis, also seen in the stem squamate *Oculudentavis* and squamate outgroups[29,33]; (3) the presence of palatine teeth, also seen in squamate outgroups[29,32] as well as some early-diverging squamates[26] and a few extant anguimorphs and iguanians[18]; (4) the fusion of posterior mandibular bones into a compound bone, which is widespread among squamates; and (5) the presence of a distinct, medially projecting palatine process of the maxilla[7], which is also widespread among squamates[20].

Snake-like features are limited to the teeth, some tooth-bearing bones, and dental replacement[7,39]. These may be ecologically relevant; highly recurved teeth evolved independently in various groups of predatory squamates (described above). The same may be true for the posterior embayment of the lateral wall of the dentary, seen in the parviraptorid *Portugalophis*[7], which is also present in mosasauroids and anguimorphs, the stem squamate *Oculudentavis*[33], and in *Dalinghosaurus*[26]. Parviraptorids lack snake-like braincase characters. Notably, the absence of a bony enclosure of the anterior braincase in NMS G.2023.7.1 contradicts the identification of a suboptic shelf of the frontal as evidence for an enclosed braincase in *Parviraptor estei*[7].

Mosaic anatomy in parviraptorids illustrates the complexity of morphological evolution during early squamate diversification. *Breugnathair* combines highly derived, snake-like features with deep squamate plesiomorphies also seen in stem squamates and gekkotans. This causes discordance among our phylogenetic analyses. Such complexities may be expected given the conflicts between morphological and molecular inferences of living squamate phylogeny[2]. Nevertheless, molecular phylogenetic topologies do not predict that early stem snakes should share features with gekkotans and stem squamates, which are separated from snakes by large phylogenetic distances (Fig. 4). The occurrence of such features in parviraptorids could indicate that the snake ancestor had considerably different traits than expected based on the anatomy of other living toxicoferans (anguimorphs and iguanians; based on the phylogenies from Datasets 2 and 3; Fig. 4a). Alternatively, these features may indicate that parviraptorids are stem squamates and not stem snakes (based on Dataset 1; Fig. 4a; see also ref. 12). If so, then the predatory traits of parviraptorids provide evidence of an early ecological radiation among stem squamates, occurring by the Middle Jurassic, before crown squamates had attained substantial diversity. Under either hypothesis the character conflicts evident in parviraptorid anatomy illustrate important uncertainties about morphological transformations during early squamate evolution. This emphasizes the need to obtain high-resolution anatomical data for many more taxa, and for renewed progress in the discovery of new fossil squamates that may shed light on the initial diversification of this important group.

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

## Methods

### Preservation and taphonomy of NMS G.2023.7.1

Disarticulated bones of NMS G.2023.7.1 are spread over a diameter of approximately 19 cm on an undulating bedding surface. This preservation is similar to that of specimens of *P. estesi* (NHMUK (Natural History Museum, London, UK) PV OR 41388) and cf. *P. estesi* (NHMUK PV R8851), from the Early Cretaceous Purbeck Limestone Group[8]. Various elements are visible at the surface of the block, including a right mandible, braincase, various other skull bones, vertebrae, ribs, a partial right coracoid, humeri, a right ilium and right femur (Fig. 1b). Numerous additional bones are present in the matrix (Fig. 1b). Of the skull, NMS G.2023.7.1 includes the left jugal, left postfrontal, left and right parietals, left and right squamosal, braincase, left vomer, left palatine, right pterygoid, right dentary, right angular, and right compound bone incorporating the surangular, prearticular and articular. Of the axial skeleton NMS G.2023.7.1 includes 32 vertebrae or partial vertebrae, many dorsal ribs, and a cervical intercentrum (Extended Data Fig. 5 and Supplementary Data 2). Of the appendicular skeleton, NMS G.2023.7.1 includes the right coracoid, right and left humeri missing distal ends, right ilium, fragment of right pubis, right femur missing distal end, left femur missing epiphyses, right tibia missing epiphyses, one metapodial, and three phalanges, including one ungual phalanx. Details of all these elements, including links to 3D digital models, are included in Supplementary Data 1.

The morphologies and lack of duplication of all squamate elements preserved in NMS G.2023.7.1, are consistent with assignment to a single relatively large-bodied individual (see main text). Although the bones are disarticulated, they are spread in a rostral to caudal pattern with skull elements and anterior vertebrae at one side of the block, and hindlimb and caudal vertebrae at the opposite edge, with dorsal vertebrae, ribs, and forelimb elements between (Fig. 1b and Extended Data Fig. 5). NMS G.2023.7.1 is isolated from other skeletons recovered from the same bedding surface by at least two metres distance. Those skeletons belong to mammals, amphibians and fish, but not squamates. Thus, it is unlikely that any squamate elements reported here represent distinct squamate individuals or species, transported from adjacent areas of the lagoon floor.

The presence of large numbers of easily transported Voorhies Group 1 elements (for example, vertebrae and ribs[51]) indicates that the skeleton was subjected to currents that were neither strong enough to remove these elements, nor strong enough to transport allochthonous Group 2 and 3 elements (that is, limb bones and skull elements[51]) from other individuals into the bone scatter before it was buried. Other non-squamate vertebrate elements such as bone crumbs and a tritylodontid tooth are present in the region of the skeleton, but these Group 3 lag elements are most likely to have been present on the lagoon floor when the skeleton was deposited, and similar material is very common throughout vertebrate-bearing levels of the sequence and around other skeletons.

These observations, the rarity of squamate remains compared to other groups in the Kilmaluag Formation assemblage[14], and the rarity of large squamate remains specifically (of almost 200 specimens collected from the Elgol Coast SSSI from 2014–2024, NMS G.2023.7.1 is the only specimen to preserve any large-bodied squamate remains) provide strong support for the view that NMS G.2023.7.1 represents a single squamate individual.

### Reconstruction of skull and body proportions

The skull reconstruction shown in Fig. 3 was conducted in Blender 3.5.0 by arranging elements of NMS G.2023.7.1 in 3D space, with reference to extant squamate anatomy. The outline of the maxilla was based on the maxilla of NHMUK PV OR 48388, the holotype of *P. estesi*. The skull length is estimated at 41.4 mm and primarily uses information from the preserved portions of the dentary and compound bone. The reconstructed skull has long, low proportions. Evidence for this comes from the braincase dimensions, relative to the lengths of the combined palatal elements and mandibular elements. We allowed additional vertical height at the back of the skull to accommodate slight crushing of the braincase.

The life reconstruction shown in Fig. 1a was produced by Mick Ellison at the American Museum of Natural History, in consultation with R.B.J.B. and S.E.E., using measurements derived from the specimen. Measurements were made from 3D digital models, using Meshlab 2023.12[52], and are reported in Supplementary Data 2. The length of the presacral vertebral column was estimated based on the summed lengths of the 23 definite presacral vertebrae with measurable lengths (84.5 mm, excluding the condyles), giving an estimated presacral length of 99.2–110.2 mm if 27–30 presacral vertebrae were originally present (allowing for the missing atlas and the possibility of missing cervicals or dorsals). Of this, the summed cervical lengths give an estimated neck length of 25.9 mm or more. The straight-line length of the longest complete dorsal rib (19 mm) and ilium length (15 mm) informed reconstruction of body depth, and the partial humerus length (10.1 mm), femur length (19 mm) and tibia length (11.5) informed reconstruction of limb lengths. Other aspects, such as hand and foot morphology, are not informed by evidence and should be considered as generalized.

### Discovery, preparation and imaging of NMS G.2023.7.1

NMS G.2023.7.1 was discovered by S.A.W. in March 2015, at the Elgol Coast SSSI, during fieldwork led by R.B.J.B. and S.A.W., assisted by A. Wolniewicz, with permission of the landowner, the John Muir Trust, under permit from NatureScot (then, Scottish Natural Heritage). It was extracted as a block of micritic limestone approximately 220 mm long, 180 mm wide and 150 mm deep. This block was embedded in silicone and prepared from behind using acetic acid by S. Moore-Fay of Wavecut Platforms, to its current thickness of 15–30 mm.

We scanned the full slab, using the Nikon Metrology XT H 225 ST X-ray μCT scanner at the School of Earth Sciences X-ray Tomography Facility, University of Bristol, UK, providing a pilot scan of the whole specimen with all bones in their original positions. Segmentation of this and other scans in the current work was conducted using the software Mimics 19.0 (Materialise) primarily by E.F.G. The scan and parameters for this and all other μCT scans described in the current work are available on MorphoSource (links provided in Supplementary Data 1).

The pilot CT scan is the basis of the digital map of the specimen shown in Fig. 1b and Extended Data Fig. 5 and was also used to identify regions of matrix free of preserved bones that were then removed with a table-mounted disc cutter. The resulting reduction of slab diameter allowed a better signal-to-noise ratio in subsequent episodes of CT scanning. We also used the pilot scan to separate some portions of the specimen in smaller blocks for high-resolution μCT scanning at the University of Bristol facility. These blocks, and the remaining portion of the specimen, were given subpart numbers that follow from NMS G.2023.7.1: (1) NMS G.2023.7.1.1, the main part of the slab, excluding the following sections; (2) NMS G.2023.7.1.2, a small portion including the braincase, left jugal, right parietal, left postfrontal and a cervical vertebra; (3) NMS G.2023.7.1.3, small portion including the left palatine and a dorsal rib; (4) NMS G.2023.7.1.4, small portion including the left vomer; and (5) NMS G.2023.7.1.5, small portion including a tritylodontid tooth and unidentified bone fragments (grey elements Fig. 1b, bottom right).

We also targeted regions of the remaining slab (NMS G.2023.7.1.1) for phase-contrast synchrotron X-ray tomography on beamline ID19 of the European Synchrotron Radiation Facility (ESRF), Grenoble, France (described below): (1) the right dentary, angular and compound bone, right pterygoid, right squamosal, left parietal and left humerus; (2) the right ilium, partial pubis and femur, phalanges including an ungual phalanx, dorsal ribs, and eight vertebrae, including dorsals, caudals and a cervical intercentrum; and (3) the right humerus and scapulocoracoid, plus a phalanx and a cervical and two dorsal vertebrae.

Finally, the main slab (NMS G.2023.7.1.1) was split into four portions that were scanned separately at School of Earth Sciences, University

of Bristol. This allowed higher quality 3D models of a few elements for which models from our other scans were not of sufficient quality, including the tibia, atlas and some other vertebrae.

We also completed CT scans of the holotype (NHMUK PV OR 48388) and referred (NHMUK PV R8851) specimens of *P. estesi* using a Nikon XTEK H 225 ST MicroCT scanner at Cambridge Biotomography Centre, University of Cambridge, UK, also available via MorphoSource (Supplementary Data 1).

### Phase-contrast synchrotron X-ray tomography

For synchrotron X-ray tomography, the beamline was set up for filtered white beam (W150-B wiggler gap 39 mm; filtered with 10 mm Cu and 0.5 mm W) resulting in a total integrated detected energy of approximately 170 keV. Images were recorded with an indirect detector comprising a 500 μm LuAG scintillator, a set of two Hasselblad lenses (100 and 150 mm; Victor Hasselblad) set for a 0.67× magnification, and a PCO.edge 4.2 sCMOS camera (PCO), resulting in a measured pixel size of 8.96 μm. The sample-detector distance was set to 13.2 m for propagation phase contrast. We used 5,000 projections over a 360° rotation, with an exposure time of 0.06 s per projection (taking the average of 3 frames of 0.02 s each), 41 flat-field images and 40 dark-field images were used as calibration. The recorded field of view in this configuration was 8.89 mm vertically and 18.35 mm horizontally (992 × 2,048 pixels). The area scanned during each rotation was increased by shifting the centre of rotation by around 8 mm horizontally (corresponding to 900 pixels on the detector), allowing us to reconstruct tomograms across a field if view spanning 34.59 × 34.59 mm (3,861 × 3,861 pixels). We then combined scans of multiple fields of view to image wider areas. Data were processed using PyHST2 with the single distance phase retrieval approach[53,54]. Post-processing included a ring artefact correction[55], change of dynamic range from 32-bits to 16-bits using the maximum and minimum 0.001% histogram clipping values of all the datasets of the series, and weighted averaging of duplicated tomograms resulting from the overlap on the vertical axis.

### Osteohistology

Three elements were thin sectioned from the associated partial skeleton NMS G.2023.7.1. The midshafts of a partial humerus and partial femur (Fig. 2c,f), and unknown region of the body of a rib were manually prepared from the specimen block with a carbide needle. The resulting pieces of bone were embedded in EpoThin 2 low viscosity epoxy resin. Embedded specimens were cut transversely at or nearest the mid-diaphysis using an Isomet 1000 precision saw. After being mounted onto frosted glass slides with clear Gorilla superglue gel, specimens were ground to optical clarity (~30 μm) on a lapidary wheel and photographed in plane-polarized light on a Nikon Eclipse LV100POL microscope fitted with a Prior ProScan III automated stage adaptor. Composite images were processed using Nikon NIS-Elements BR (version 5.24.03) imaging software using the Extended Depth Focus function to autofocus and z-stack photomicrographs to improve image quality of microstructural details. Two thin sections were made of the humerus and femur, and one thin section was made from the rib. High-resolution photomicrographs are available from MorphoSource (Supplementary Data 1).

### Phylogenetic analyses

We included NMS G.2023.7.1 and two other parviraptorid specimens in three phylogenetic datasets. Parviraptorids were represented in these datasets by three operational taxonomic units (OTUs): (1) the holotype of *Breugnathair elgolensis* (NMS G.2023.7.1); (2) the holotype of *P. estesi* (NHMUK PV OR 48388); and (3) the referred specimen of cf. *P. estesi* (NHMUK PV R8551). Parviraptorid specimens from other localities were not included in the analyses as they are less complete (or incompletely described) but are discussed in the Supplementary Discussion. Whiteside et al.[48] recently reported the Late Triassic *Cryptovaranoides*

*microlanius* as an anguimorph, and therefore deeply nested within the squamate crown group. However, Brownstein et al.[50] presented highly differing anatomical observations and suggested that *Cryptovaranoides* may instead be an archosauromorph. Given the ambiguity of published anatomical data[48–50], we did not test the affinities of *Cryptovaranoides* here and await more well-resolved anatomical data. We also did not include the proposed stem lepidosaur *Taytalura*[56], also because of phylogenetic uncertainties[3]. Note that we closely considered the phylogenetic scores proposed by Caldwell et al.[7] for parviraptorids when scoring these matrices (explained in Supplementary Discussion).

Phylogenetic inference was carried out using Bayesian inference in MrBayes 3.2.7a[57], using a fossilized birth–death tree prior[58,59] with a proportion of extant species sampled of 0.038 under diversified sampling, and default priors for speciation rate, extinction rate and fossil sampling rate. We used an offset exponential tree age prior with a minimum age of 240 Ma (Middle Triassic) and mean age of 250 Ma (Early Triassic), and a relaxed clock transition rate model with independent gamma rates with a default variance increase prior of 10 and a log-normal clock rate with a mean of −2.56 log units and variance of 1.08. The ages of all operational taxonomic units were specified using a uniform distribution between their minimum and maximum possible stratigraphic ages, extended from ref. 3 to encompass the wider set of fossil taxa included in the analyses conducted here. We constrained the ingroup relationships of squamates to reflect current consensus based on molecular phylogenetic analyses[2], by specifying a series of partial constraints matching those shown by ref. 3. The resulting backbone constraint is consistent with the phylogeny of ref. 2, with trichotomies representing areas of uncertainty among recent analyses (that is, enforcing no specific relationships between the three constituent groups). For example, Gekkota and Dibamidae in an unresolved trichotomy with the clade including all other squamates. The toxicoferan clade (Iguania, Anguimorpha and Serpentes) was left as an unresolved trichotomy in analyses of datasets 1 and 2, but constrained to represent three different phylogenetic hypotheses of toxicoferan ingroup relationships for three separate analyses of dataset 3 (explained below).

**Dataset 1.** The early reptile dataset of Talanda et al.[3] (modified from refs. 32,45,60), which includes an extensive sample of early members of the reptile crown group, as well as living and fossil rhynchocephalians and squamates, focussing on deep lepidosaur and deep squamate divergences, but includes relatively few snakes: three extant species plus the early fossil snakes *Najash*, *Pachyrachis* and *Dinilysia*. To this we added our three parviraptorid OTUs, the hypothesized early anguimorph *Dorsetisaurus*, and the Late Jurassic squamate *Eoscincus*[5], resulting in a dataset of 125 taxa and 382 characters.

**Dataset 2.** An extended version of the squamate dataset of Meyer et al.[6] (modified from Gauthier et al.[27]). Meyer et al.[6] extended the taxon and character sample of Gauthier et al.[27] by adding characters and taxa relevant to Jurassic squamate divergences, including new fossil taxa. The resulting matrix includes a strong sample of early squamates and relevant characters, as well as a strong sample of snakes. To this we added fifteen OTUs: the possible stem lepidosaurs *Velbergia bartholomaei*, *Fraxinisaura rozynekae*, *Paliguana whitei*, *Sophineta cracoviensis* and *Marmoretta oxoniensis*, the early rhynchocephalians *Gephyrosaurus bridensis* and *Kallimodon pulchellus*, the stem squamates *Bellairsia gracilis*, *Oculudentavis naja*, *Oculudentavis khaungraae*, the possible early anguimorph *Dorsetisaurus purbeckensis*, and our three parviraptorid OTUs. This resulted in a dataset of 637 characters and 168 tips, compared to the 155 tips included by Meyer et al.[6].

**Dataset 3.** An extended version of the dataset of Zaher & Smith[47] (modified from Hsiang et al.[61]; after ref. 27), to which we added various early squamate and non-squamate fossils. The matrix of Zaher & Smith[47] included a large sample of living and fossil snakes and relevant

characters, as well as 11 iguanians, 21 anguimorphs and two rhynchocephalians. This sample was intended to focus on toxicoferans, with relevance to snake ingroup relationships.

We modified the matrix of Zaher & Smith[47] by adding many species of non-toxicoferan squamates, adopting the scores of ref. 62 (modified from ref. 27 based on revisions proposed in more recent works such as refs. 63,64) when, as in most cases, the identical characters were used in these studies. We also added fossil taxa intended to increase representation of the stem groups of Lepidosauria, Squamata, and squamate subgroups: the possible stem lepidosaurs *V. bartholomaei*, *Taytalura alcoberi*, *F. rozynekae*, *P. whitei*, *S. cracoviensis* and *M. oxoniensis*, the stem squamates *B. gracilis*, *Huehuecuetzpalli mixtecus*, *O. naja* and *O. khaungraae*, the early-diverging Jurassic and Cretaceous squamates *Hongshanxi xiei*, *Yabeinosaurus tenuis*, *Dalinghosaurus longidigitus*, *Liushusaurus acanthocaudata*, *Scandensia ciervensis*, *Meyasaurus faurai*, *Jucaraseps grandipes*, *Chometokadmon fitzingeri*, *Ardeosaurus brevipes*, *Eichstaettisaurus schroederi*, *D. purbeckensis*, and our three parviraptorid OTUs. We omitted the candidate stem snake *Tetrapodopis amplectus* from analyses due to substantial doubts about its anatomy and phylogenetic affinities[65,66].

In addition to this, we added two characters. Character 789: gastralia present (0); absent (1). Character 790: entepicondylar foramen present (0); absent (1). This resulted in a dataset of 790 characters and 189 tips, compared to 788 characters and 90 tips by Zaher & Smith[47]. Because analyses of this matrix returned parviraptorids as toxicoferans we ran three separate analyses of this dataset, using topological constraints that represent different phylogenetic hypotheses for toxicoferan ingroup relationships: (1) snakes constrained as sister to anguimorphs; (2) snakes constrained as sister to iguanians; and (3) anguimorphs and iguanians constrained as sister taxa, to the exclusion of snakes.

**Evaluating convergence.** Analysis of dataset 1 was run for 133 million generations, dataset 2 was run for 160 million generations, and dataset 3 was run for up to 180 million generations prior to convergence (at an average standard deviation of split frequencies of 0.01 or less). Chains were sampled every 10,000th generation, with a burn-in of 50%. The effective sample size was greater than 200 for all tested parameters, and an average potential scale reduction factor was 1.01 or less on all parameters, indicating convergence.

**Reporting summary**
Further information on research design is available in the Nature Portfolio Reporting Summary linked to this article.

## Data availability
All computed tomography data and three-dimensional models reported in this paper are available at MorphoSource (https://www.morphosource.org/projects/000354820). The phylogenetic datasets analysed here are available at https://doi.org/10.17605/OSF.IO/NVZUD.

## Code availability
Our phylogenetic scripts, including full analytical settings, are available at https://doi.org/10.17605/OSF.IO/NVZUD.

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

**Acknowledgements** The authors thank John Muir Trust and NatureScot for permission to collect on the Elgol Site of Special Scientific Interest; A. Wolniewicz for assistance in the field; S. Moore-Fay for preparation of the specimen; D. MacLeod for advice on Scottish Gaelic construction; M. Ellison for the illustration of *Breagnathair elgolensis* in Fig. 1a; R. Araújo and A. LeBlanc for data on *Portugalophis*; and T. Davies and K. Smithson for assistance with CT scanning. We acknowledge the European Synchrotron Radiation Facility (ESRF) for provision of synchrotron radiation facilities under proposal number LS-2687. J.B. was funded by the National Research Foundation, Genus: DSTI-NRF Centre of Excellence in Palaeosciences and the Palaeontological Scientific Trust (PAST).

**Author contributions** R.B.J.B., S.E.E. and J.J.H. designed the project. R.B.J.B. and S.A.W. led the fieldwork. S.A.W. found the specimen. R.B.J.B, J.J.H. and V.F. collected μCT and synchrotron CT data. E.F.G. carried out digital segmentation. R.B.J.B., S.E.E. and J.J.H. made anatomical interpretations and descriptions. R.B.J.B. and S.E.E. developed the phylogenetic datasets and analyses. Z.T.K. made histological thin sections. Z.T.K. and J.B. contributed histological data and interpretation. R.B.J.B. and S.E.E. led manuscript preparation and writing and all authors contributed to editing the manuscript.

**Competing interests** The authors declare no competing interests.

**Additional information**
**Correspondence and requests for materials** should be addressed to Roger B. J. Benson or Susan E. Evans.

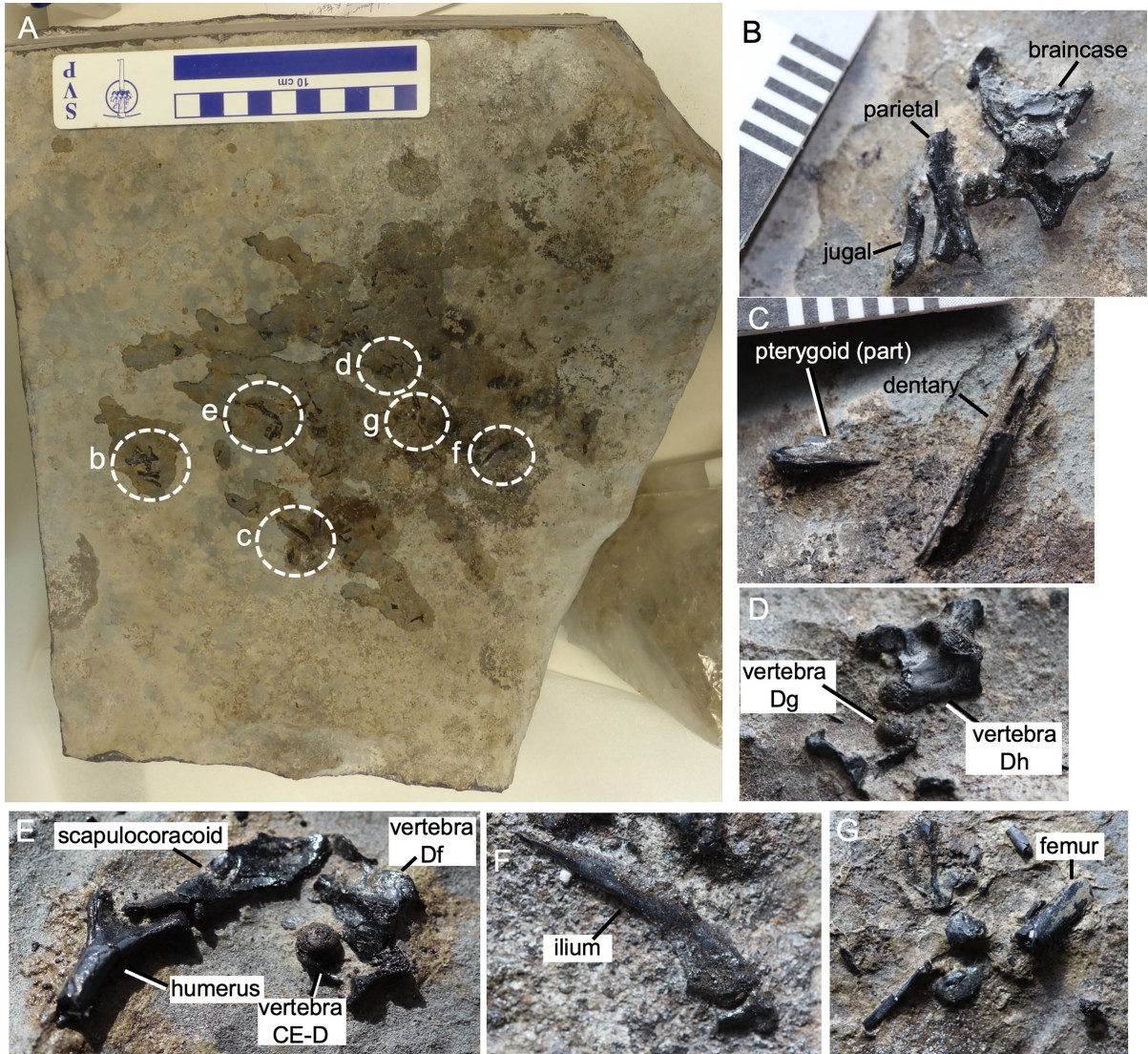

**Extended Data Fig. 1 | Photos of the holotype of *Breugnathair elgolensis* NMS G.2023.7.1.** (A) Whole specimen with locations of panels (B)–(G) labelled. (B) Braincase and associated cranial elements, (C) dentary and quadrate ramus of pterygoid, (D) dorsal vertebrae Dg and Dh, (E) partial humerus and scapulocoracoid with two vertebrae (cervicodorsal CE-D and dorsal Df), (F) ilium with attached part of pubis, (G) femoral shaft and associated bones.

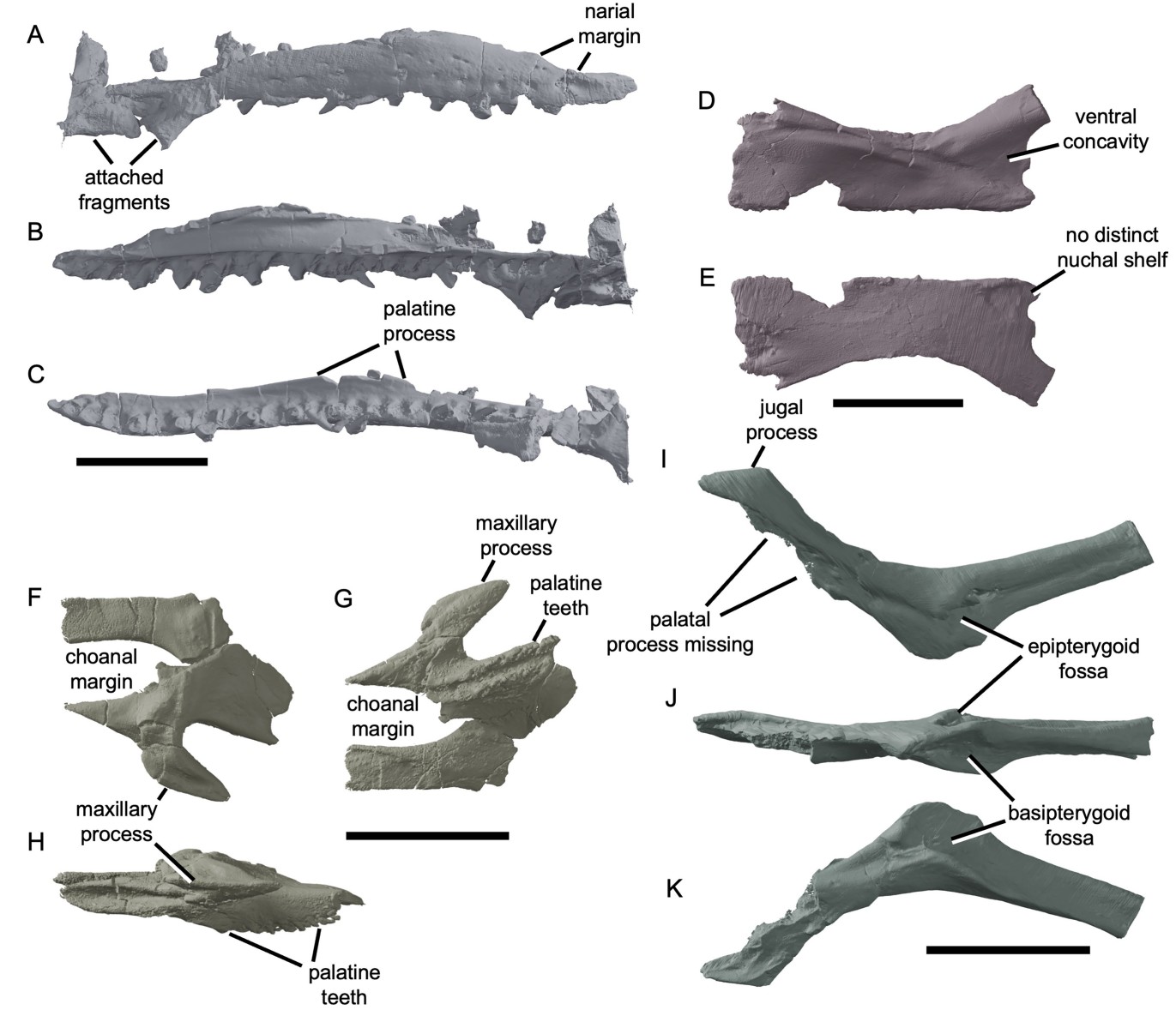

**Extended Data Fig. 2 | Holotype of *Parviraptor estesi* (NHMUK PV OR 48388) from the Early Cretaceous Purbeck Limestone Group.** Digital renders of (A–C) right maxilla in lateral (A), medial (B) and ventral (C) views; (D–E) left parietal in ventral (D) and dorsal (E) views; (F–H) left palatine in dorsal (F), ventral (G) and lateral (H) views; and right pterygoid (I–K) in dorsal (I), medial (J), and ventral (K) views. Scale bars equal 5 mm.

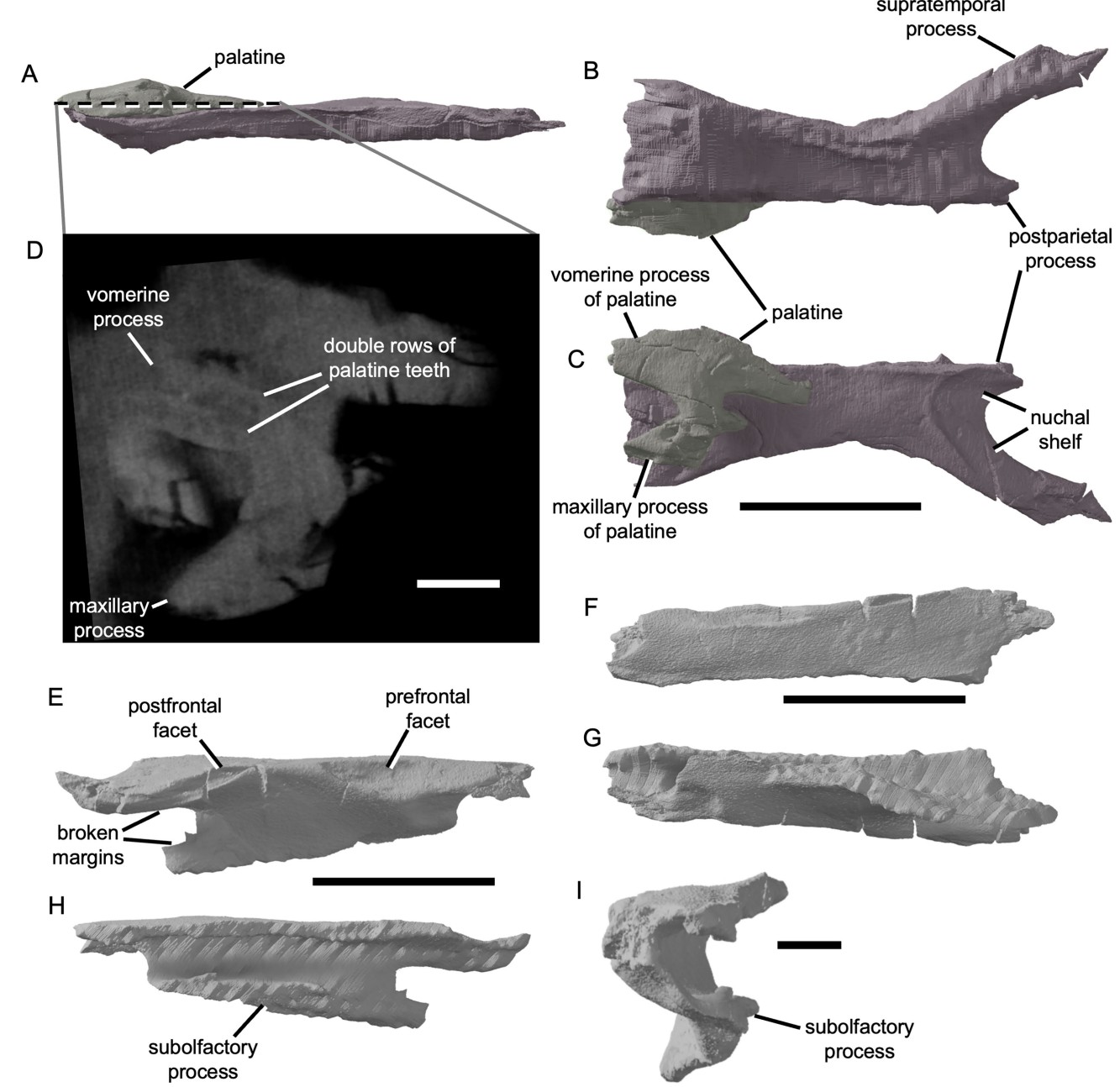

**Extended Data Fig. 3 | Left parietal, left palatine and right frontal of parviraptorid specimen (NHMUK PV R8511; cf. *Parviraptor estesi*) from the Early Cretaceous Purbeck Limestone Group.** (A–C) Digital renders of parietal and attached palatine in medial (A), ventral (B), and dorsal (C) views; (D) computed tomographic image showing ventral part of vomerine process of palatine appressed to dorsal surface of parietal and therefore not visible in the physical specimen; (E–I) digital renders of right frontal in lateral (E), dorsal (F), ventral (G), medial (H) and anterior (I) views. Dashed line in (A) indicates plane of section (D). Scale bars equal 5 mm (A–C, E–H) and 1 mm (D, I).

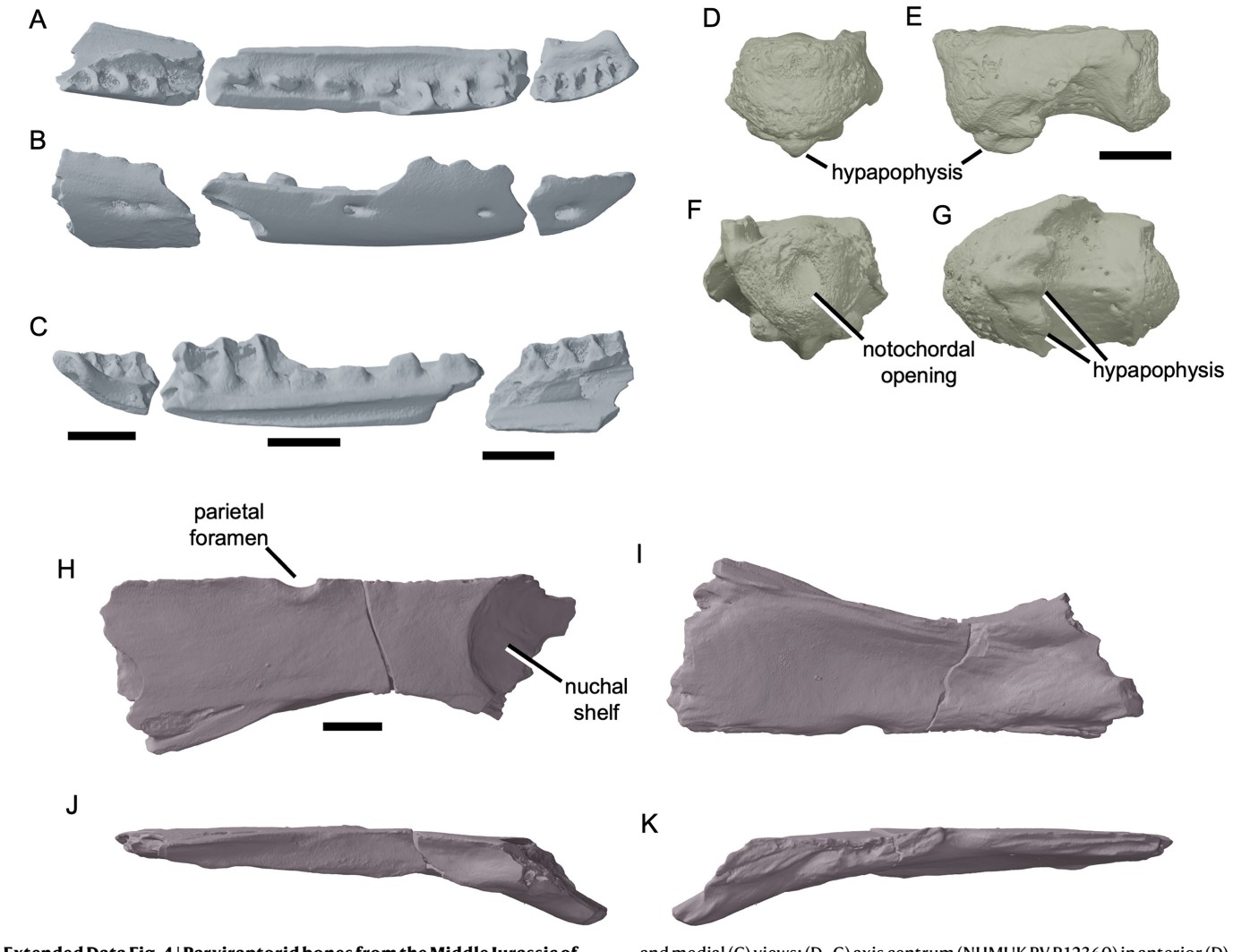

**Extended Data Fig. 4 | Parviraptorid bones from the Middle Jurassic of Kirtlington quarry, Oxfordshire, UK, including the holotype dentary symphysis of *Eophis underwoodi* (NHMUK PV R12355; a nomen dubium as Parviraptoridae indet.).** Digital renders of (A–C) dentary portions (NHMUK PV R12355, R12354, R12370 from anterior to posterior) in dorsal (A), lateral (B) and medial (C) views; (D–G) axis centrum (NHMUK PV R12360) in anterior (D), left lateral (E), posterior (F) views, and ventral (G) views; and (H–K) left parietal (NHMUK PV R12353) in dorsal (H), ventral (I), lateral (J) and medial (K) views. Scale bars equal 1 mm.

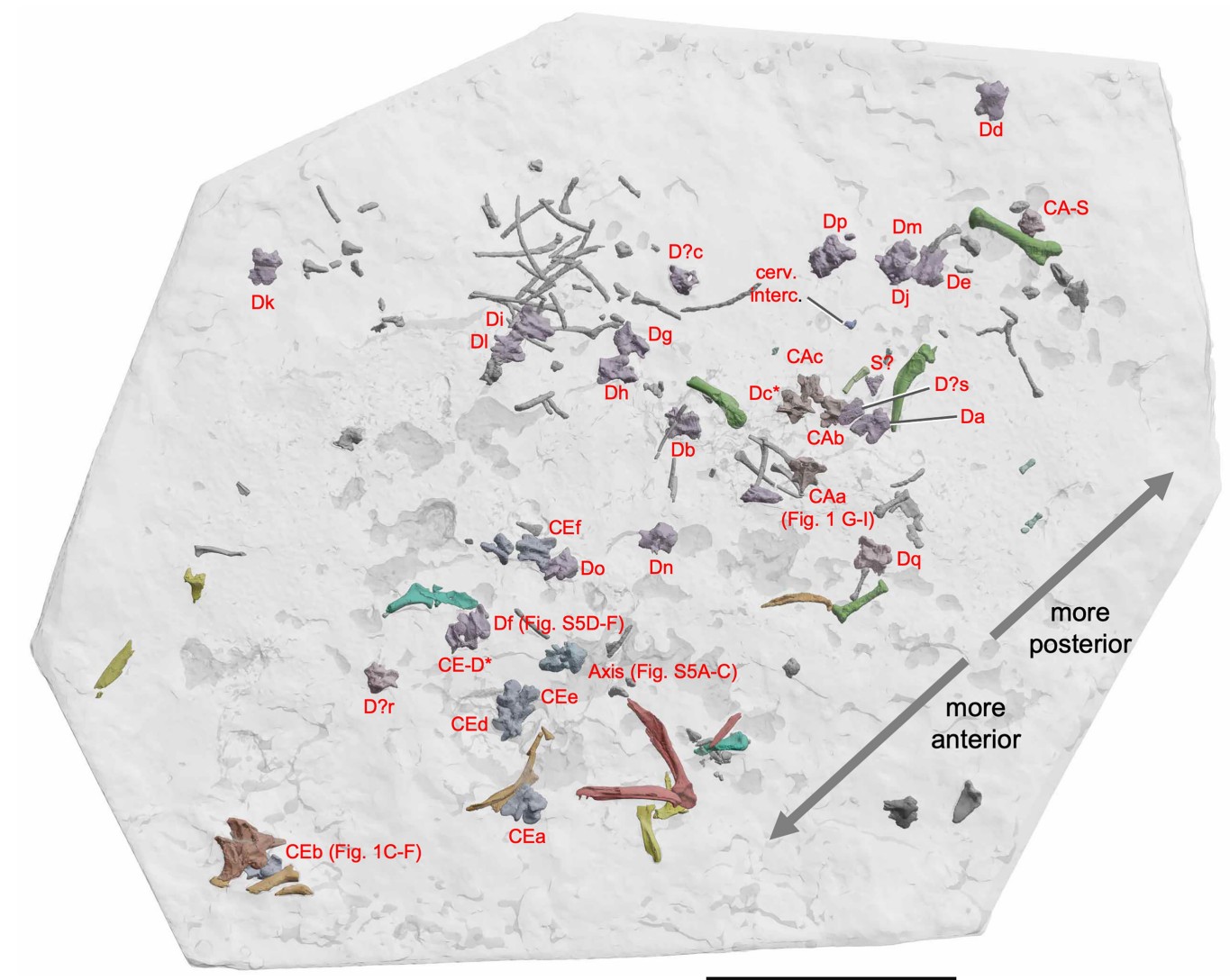

**Extended Data Fig. 5 | Digital rendering of NMS G.2023.7.1 as preserved, indicating vertebral identities.** Each vertebra is indicated by an alphanumeric code that is also used in Supplementary Data 2. Within this code, 'CE' indicates cervical vertebrae, 'CE-D' indicates cervicodorsal (i.e. posterior cervical or anterior dorsal), 'D' indicates dorsal vertebrae, 'S' indicates sacral, 'CA-S' indicates caudal or sacral, and 'CA' indicates caudal vertebrae. Scale bar equals 50 mm.

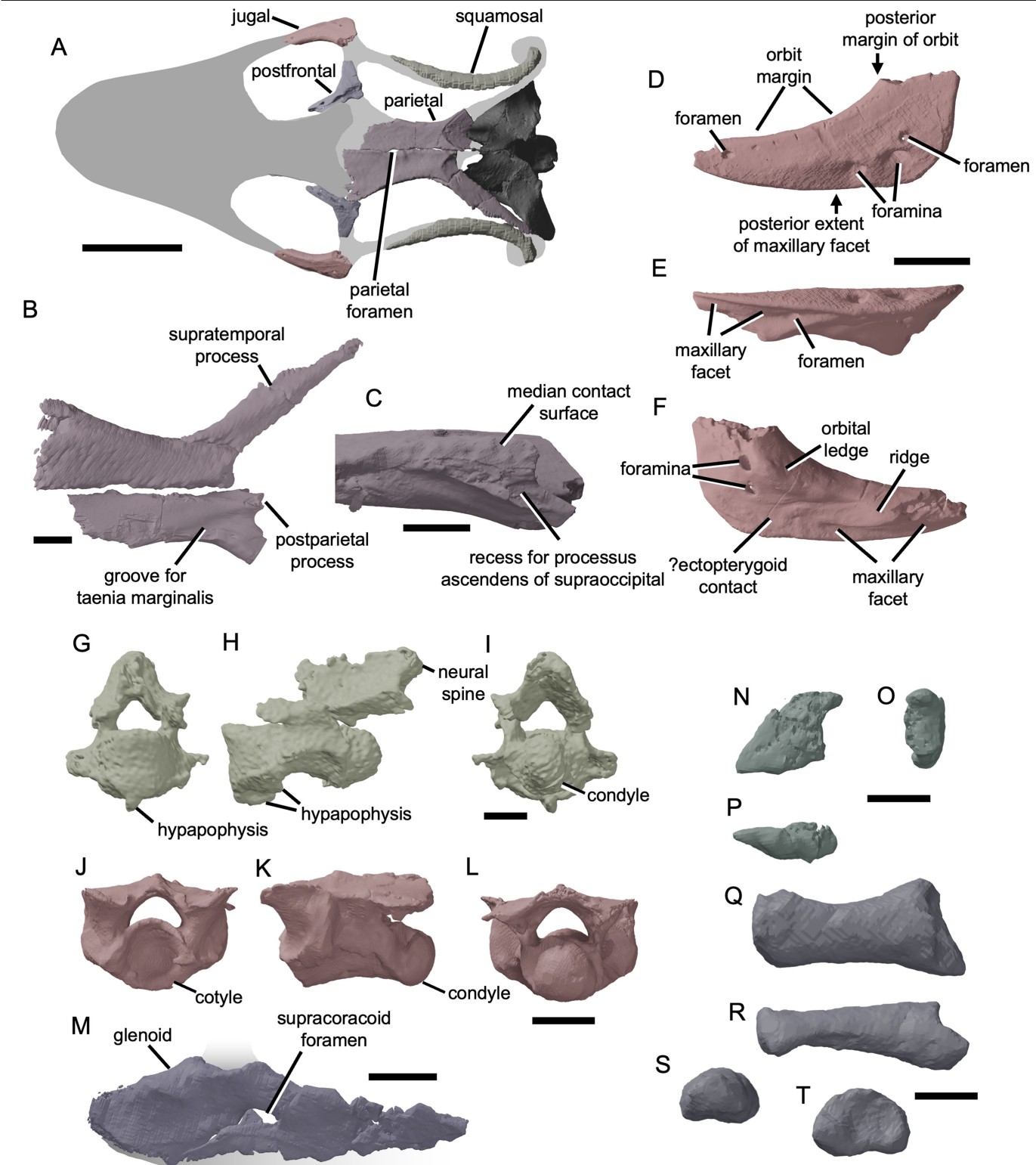

**Extended Data Fig. 6 | Additional details of NMS G.2023.7.1.** Digital render of cranium in dorsal view (A) plus digital renders of: (B) paired parietals in ventral view; (C) posterior part of right parietal in medial view; (D–F) the left jugal in lateral (D), ventral (E) and medial (F) views; (G–I) the axis in anterior (G), left lateral (H) and posterior (I) views; (J–L) dorsal vertebra (Df; see Extended Data Fig. 5) in anterior (J), left lateral (K) and posterior (L) views; (M) partial right scapulocoracoid in lateral view, with grey shaded areas indicating missing portions; (N–P) ungual phalanx (bone 22) in lateral or medial (N), proximal (O) and ventral (P) views; and (Q–T) phalanx (bone 9) in ventral (Q), lateral or medial (R), distal (S) and proximal (T) views; Scale bars equal 10 mm (A), 2 mm (B, D, M) and 1 mm (C, G–L, N–T).

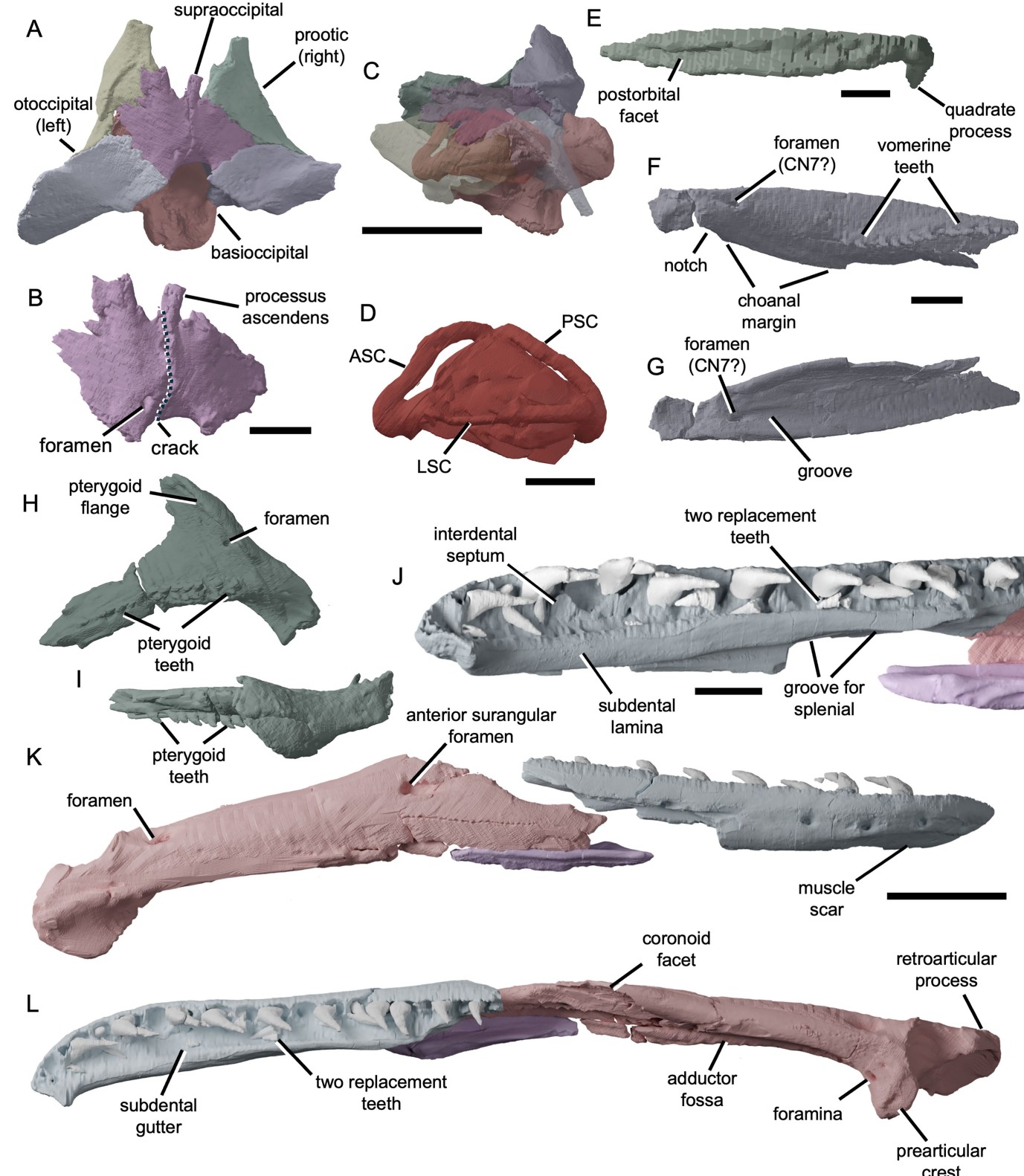

**Extended Data Fig. 7 | Additional details of NMS G.2023.7.1.** Digital renders of (A) braincase in dorsal view with separate bones colours as in Fig. 3; (B) supraoccipital in dorsal view; (C) semi-transparent braincase in left lateral view showing labyrinth endocast internally; (D) left labyrinth endocast in lateral view; (E) right squamosal in medial view; (F–G) right vomer in ventral (F) and dorsal (G) views; (H–I) left pterygoid in ventral (H) and lateral (I) views; (J) anterior part of right mandible in medial view showing details of teeth and replacement teeth; (K–L) right mandible in lateral (K) and dorsal (L) views. Scale bars equal 5 mm (A, C, J–L) and 2 mm (B, D, E–I).

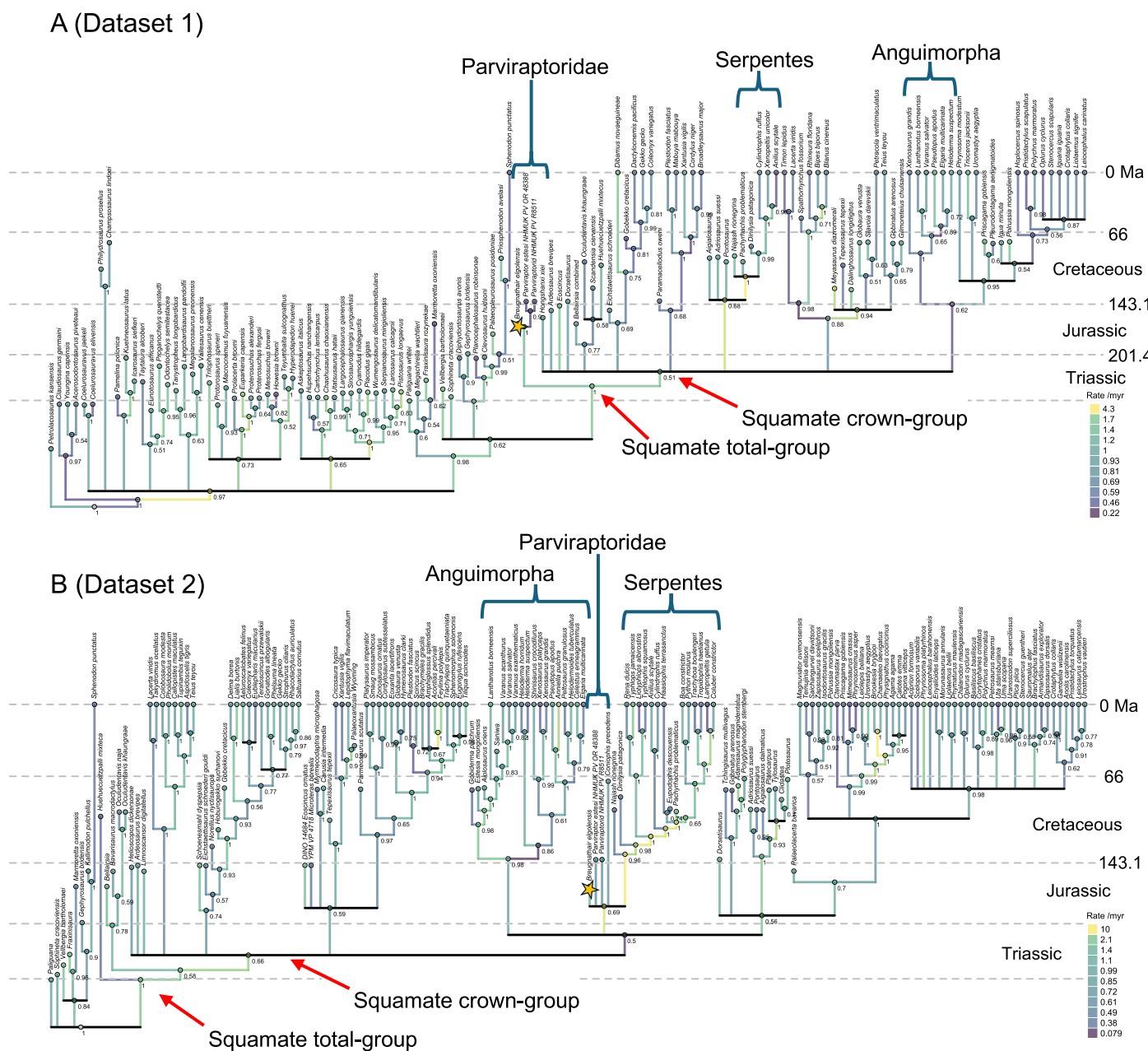

## A (Dataset 1)

**Extended Data Fig. 8 | Phylogenetic results of Datasets 1 (A) and 2 (B).**
Time-scaled majority rule consensus trees derived from posterior distribution of Bayesian phylogenetic inference (*Methods*) are shown, with branches colours indicating transition frequencies. Numbers adjacent to node are posterior probabilities derived from the full sample of posterior topologies with the first 50% discarded ad burning, sampling every 10,000 generations.

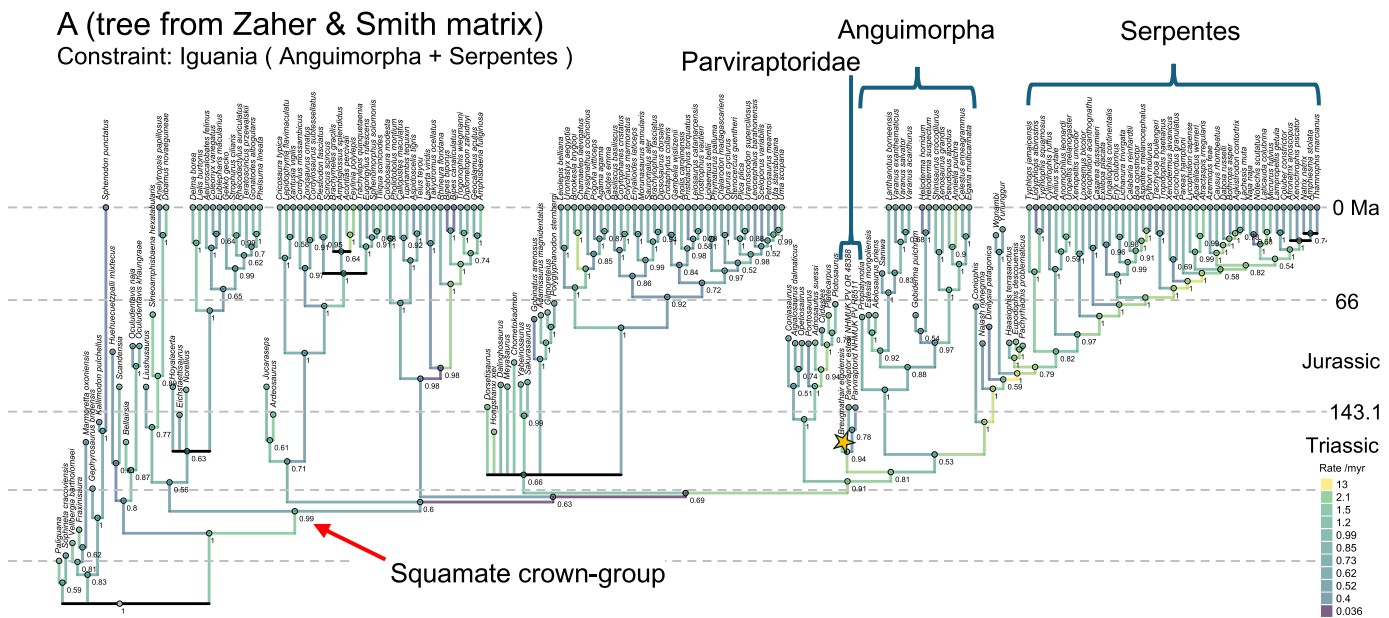

## A (tree from Zaher & Smith matrix)
Constraint: Iguania ( Anguimorpha + Serpentes )

Anguimorpha

Parviraptoridae

Serpentes

Squamate crown-group

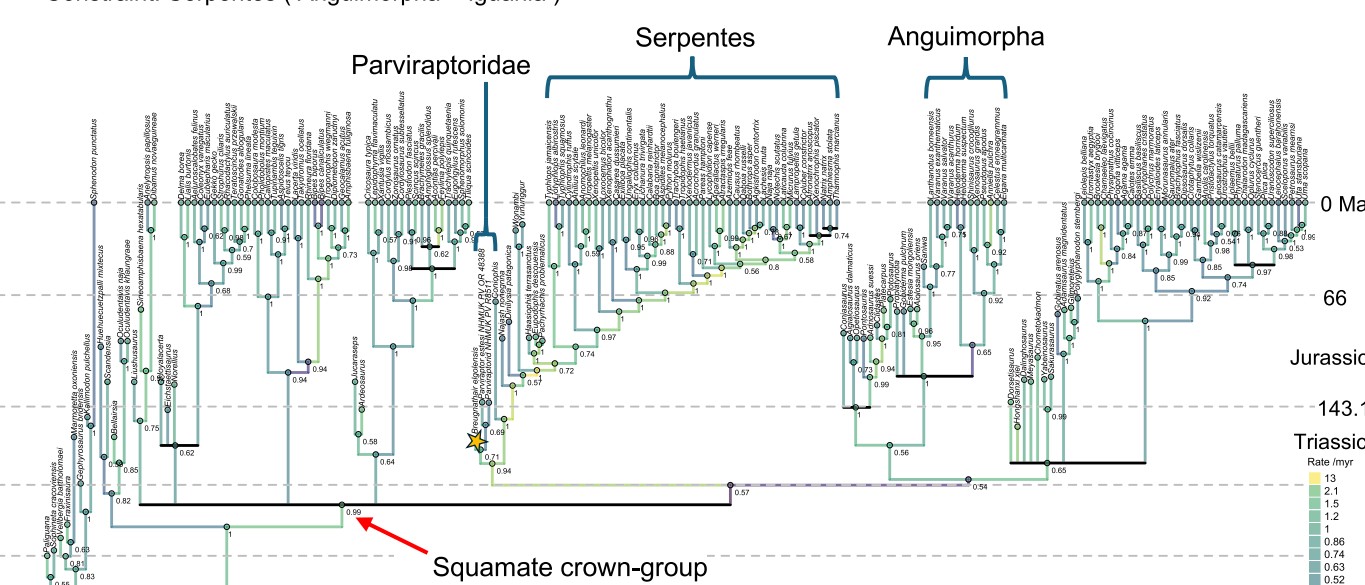

## B (tree from Zaher & Smith matrix)
Constraint: Serpentes ( Anguimorpha + Iguania )

Parviraptoridae

Serpentes

Anguimorpha

Squamate crown-group

**Extended Data Fig. 9 | Phylogenetic results of Dataset 3.** (A) Constraining Anguimorpha as the sister of Serpentes within Toxicofera; (B) Constraining Anguimorpha as the sister of Iguania within Toxicofera. Timescaled majority rule consensus trees derived from posterior distribution of Bayesian phylogenetic inference (*Methods*) are shown, with branches colours indicating transition frequencies. Numbers adjacent to node are posterior probabilities derived from the full sample of posterior topologies with the first 50% discarded ad burning, sampling every 10,000 generations.

## Tree from Zaher & Smith matrix
Constraint: Anguimorpha ( Iguania + Serpentes )

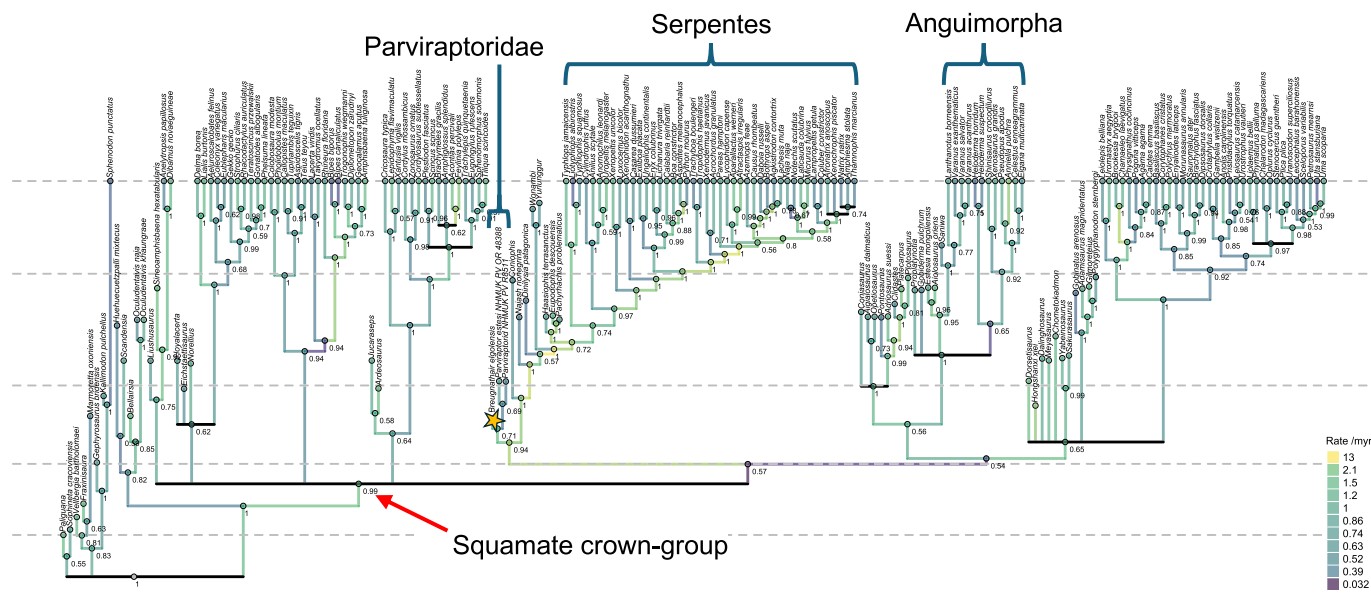

**Extended Data Fig. 10 | Phylogenetic results of Dataset 3.** constraining Iguania as the sister of Serpentes within Toxicofera. The timescaled majority rule consensus tree derived from posterior distribution of Bayesian phylogenetic inference (*Methods*) is shown, with branches colours indicating transition frequencies. Numbers adjacent to node are posterior probabilities derived from the full sample of posterior topologies with the first 50% discarded ad burning, sampling every 10,000 generations.

# Reporting Summary

## Statistics

For all statistical analyses, confirm that the following items are present in the figure legend, table legend, main text, or Methods section.

| n/a | Confirmed | |
|---|---|---|
| ☒ | ☒ | The exact sample size (*n*) for each experimental group/condition, given as a discrete number and unit of measurement |
| ☒ | ☐ | A statement on whether measurements were taken from distinct samples or whether the same sample was measured repeatedly |
| ☒ | ☐ | The statistical test(s) used AND whether they are one- or two-sided<br>*Only common tests should be described solely by name; describe more complex techniques in the Methods section.* |
| ☒ | ☐ | A description of all covariates tested |
| ☒ | ☐ | A description of any assumptions or corrections, such as tests of normality and adjustment for multiple comparisons |
| ☒ | ☐ | A full description of the statistical parameters including central tendency (e.g. means) or other basic estimates (e.g. regression coefficient) AND variation (e.g. standard deviation) or associated estimates of uncertainty (e.g. confidence intervals) |
| ☒ | ☐ | For null hypothesis testing, the test statistic (e.g. *F*, *t*, *r*) with confidence intervals, effect sizes, degrees of freedom and *P* value noted<br>*Give P values as exact values whenever suitable.* |
| ☐ | ☒ | For Bayesian analysis, information on the choice of priors and Markov chain Monte Carlo settings |
| ☒ | ☐ | For hierarchical and complex designs, identification of the appropriate level for tests and full reporting of outcomes |
| ☒ | ☐ | Estimates of effect sizes (e.g. Cohen's *d*, Pearson's *r*), indicating how they were calculated |

*Our web collection on statistics for biologists contains articles on many of the points above.*

## Software and code

Policy information about availability of computer code

| Data collection | *Provide a description of all commercial, open source and custom code used to collect the data in this study, specifying the version used OR state that no software was used.*  Mimics 19.0 (Materialise, Leuven), Blender 3.5.0, Meshlab 2023.12, Nikon NIS-Elements BR 5.24.03 |
|---|---|
| Data analysis | Phylogenetic inference was carried out using Bayesian inference in MrBayes 3.2.7a (Ronquist et al. 2012) ~~Our phylogenetic scripts, including full analytical settings, are available at https://doi.org/10.17605/OSF.IO/NVZUD~~ |

For manuscripts utilizing custom algorithms or software that are central to the research but not yet described in published literature, software must be made available to editors and reviewers. We strongly encourage code deposition in a community repository (e.g. GitHub). See the Nature Portfolio guidelines for submitting code & software for further information.

## Data

Policy information about availability of data

All manuscripts must include a data availability statement. This statement should provide the following information, where applicable:

- Accession codes, unique identifiers, or web links for publicly available datasets
- A description of any restrictions on data availability
- For clinical datasets or third party data, please ensure that the statement adheres to our policy

All computed tomography data and three-dimensional models reported in this paper are available at Morphosource (https://www.morphosource.org/projects/000354820).   The phylogenetic datasets analysed here are available at https://doi.org/10.17605/OSF.IO/NVZUD.

# Research involving human participants, their data, or biological material

Policy information about studies with human participants or human data. See also policy information about sex, gender (identity/presentation), and sexual orientation and race, ethnicity and racism.

| | |
|---|---|
| Reporting on sex and gender | *Use the terms sex (biological attribute) and gender (shaped by social and cultural circumstances) carefully in order to avoid confusing both terms. Indicate if findings apply to only one sex or gender; describe whether sex and gender were considered in study design; whether sex and/or gender was determined based on self-reporting or assigned and methods used. Provide in the source data disaggregated sex and gender data, where this information has been collected, and if consent has been obtained for sharing of individual-level data; provide overall numbers in this Reporting Summary. Please state if this information has not been collected. Report sex- and gender-based analyses where performed, justify reasons for lack of sex- and gender-based analysis.*<br>NA |
| Reporting on race, ethnicity, or other socially relevant groupings | *Please specify the socially constructed or socially relevant categorization variable(s) used in your manuscript and explain why they were used. Please note that such variables should not be used as proxies for other socially constructed/relevant variables (for example, race or ethnicity should not be used as a proxy for socioeconomic status). Provide clear definitions of the relevant terms used, how they were provided (by the participants/respondents, the researchers, or third parties), and the method(s) used to classify people into the different categories (e.g. self-report, census or administrative data, social media data, etc.) Please provide details about how you controlled for confounding variables in your analyses.*<br>NA |
| Population characteristics | *Describe the covariate-relevant population characteristics of the human research participants (e.g. age, genotypic information, past and current diagnosis and treatment categories). If you filled out the behavioural & social sciences study design questions and have nothing to add here, write "See above."*<br>NA |
| Recruitment | *Describe how participants were recruited. Outline any potential self-selection bias or other biases that may be present and how these are likely to impact results.*<br>NA |
| Ethics oversight | *Identify the organization(s) that approved the study protocol.*<br>NA |

Note that full information on the approval of the study protocol must also be provided in the manuscript.

# Field-specific reporting

Please select the one below that is the best fit for your research. If you are not sure, read the appropriate sections before making your selection.

☐ Life sciences          ☐ Behavioural & social sciences          ☒ Ecological, evolutionary & environmental sciences

For a reference copy of the document with all sections, see [nature.com/documents/nr-reporting-summary-flat.pdf](http://nature.com/documents/nr-reporting-summary-flat.pdf)

# Ecological, evolutionary & environmental sciences study design

All studies must disclose on these points even when the disclosure is negative.

| | |
|---|---|
| Study description | The study reports a fossil lizard skeleton (National Museums of Scotland [NMS] G.2023.7.1) imaged using 3D digital models derived from X-ray micro-computed tomography. |
| Research sample | One specimen is studied (National Museums of Scotland [NMS] G.2023.7.1), with qualitative comparisons to additional specimens, and phylogenetic analysis based on existing datasets |
| Sampling strategy | Taxa included in the phylogenetic analysis were selected to represent a broad, phylogenetically over-dispersed sample of squamates and their evolutionary relatives. |
| Data collection | R.B.J.B, J.H.H and V.F. collected micro-CT and synchrotron CT data. E.F.G. carried out digital segmentation. R.B.J.B., S.E.E. and J.H.H. made anatomical interpretations and descriptions. R.B.J.B. and S.E.E. developed the phylogenetic datasets and analyses. Z.K. made histological thin sections. Z.K. and J.B. contributed histological data and interpretation |
| Timing and spatial scale | National Museums of Scotland [NMS] G.2023.7.1 is from the Middle Jurassic of Scotland. Comparative specimens span from the Triassic to the present (250 million years ago – present) |
| Data exclusions | The Early Cretaceous fossil Tetrapodophis was excluded from phylogenetic analyses because of published doubts about its anatomy. |
| Reproducibility | Not relevant to the present study (not experimental) |
| Randomization | Not relevant to the present study (not experimental) |
| Blinding | Not relevant to the present study (not experimental) |

Did the study involve field work?  ☒ Yes   ☐ No

## Field work, collection and transport

| Field conditions | Field conditions are not relevant to the study |
|---|---|
| Location | Elgol Coast Site of Special Scientific Interest, Isle of Skye, Scotland, United Kingdom |
| Access & import/export | Fieldwork was conducted with permission of the landowner, the John Muir Trust, under permit from NatureScot (then, Scottish Natural Heritage). |
| Disturbance | Fossil extraction site was naturalised using a lump hammer |

# Reporting for specific materials, systems and methods

We require information from authors about some types of materials, experimental systems and methods used in many studies. Here, indicate whether each material, system or method listed is relevant to your study. If you are not sure if a list item applies to your research, read the appropriate section before selecting a response.

### Materials & experimental systems

| n/a | Involved in the study |
|---|---|
| ☒ | ☐ Antibodies |
| ☒ | ☐ Eukaryotic cell lines |
| ☐ | ☒ Palaeontology and archaeology |
| ☒ | ☐ Animals and other organisms |
| ☒ | ☐ Clinical data |
| ☒ | ☐ Dual use research of concern |
| ☒ | ☐ Plants |

### Methods

| n/a | Involved in the study |
|---|---|
| ☒ | ☐ ChIP-seq |
| ☒ | ☐ Flow cytometry |
| ☒ | ☐ MRI-based neuroimaging |

## Antibodies

| Antibodies used | Describe all antibodies used in the study; as applicable, provide supplier name, catalog number, clone name, and lot number. |
|---|---|
| Validation | Describe the validation of each primary antibody for the species and application, noting any validation statements on the manufacturer's website, relevant citations, antibody profiles in online databases, or data provided in the manuscript. |

## Eukaryotic cell lines

Policy information about cell lines and Sex and Gender in Research

| Cell line source(s) | State the source of each cell line used and the sex of all primary cell lines and cells derived from human participants or vertebrate models. |
|---|---|
| Authentication | Describe the authentication procedures for each cell line used OR declare that none of the cell lines used were authenticated. |
| Mycoplasma contamination | Confirm that all cell lines tested negative for mycoplasma contamination OR describe the results of the testing for mycoplasma contamination OR declare that the cell lines were not tested for mycoplasma contamination. |
| Commonly misidentified lines (See ICLAC register) | Name any commonly misidentified cell lines used in the study and provide a rationale for their use. |

## Palaeontology and Archaeology

| Specimen provenance | Elgol Coast Site of Special Scientific Interest, Isle of Skye, Scotland, United Kingdom. Fieldwork was conducted with permission of the landowner, the John Muir Trust, under permit from NatureScot (then, Scottish Natural Heritage). |
|---|---|
| Specimen deposition | National Museums of Scotland [NMS] G.2023.7.1 |

| Dating methods | No new dates |
|---|---|

☐ Tick this box to confirm that the raw and calibrated dates are available in the paper or in Supplementary Information.

| Ethics oversight | No ethical approval or guidance was required. Study of fossils conducted legally, under permit. |
|---|---|

Note that full information on the approval of the study protocol must also be provided in the manuscript.

# Animals and other research organisms

Policy information about studies involving animals; ARRIVE guidelines recommended for reporting animal research, and Sex and Gender in Research

| Laboratory animals | *For laboratory animals, report species, strain and age OR state that the study did not involve laboratory animals.* |
|---|---|
| Wild animals | *Provide details on animals observed in or captured in the field; report species and age where possible. Describe how animals were caught and transported and what happened to captive animals after the study (if killed, explain why and describe method; if released, say where and when) OR state that the study did not involve wild animals.* |
| Reporting on sex | *Indicate if findings apply to only one sex; describe whether sex was considered in study design, methods used for assigning sex. Provide data disaggregated for sex where this information has been collected in the source data as appropriate; provide overall numbers in this Reporting Summary. Please state if this information has not been collected. Report sex-based analyses where performed, justify reasons for lack of sex-based analysis.* |
| Field-collected samples | *For laboratory work with field-collected samples, describe all relevant parameters such as housing, maintenance, temperature, photoperiod and end-of-experiment protocol OR state that the study did not involve samples collected from the field.* |
| Ethics oversight | *Identify the organization(s) that approved or provided guidance on the study protocol, OR state that no ethical approval or guidance was required and explain why not.* |

Note that full information on the approval of the study protocol must also be provided in the manuscript.

# Clinical data

Policy information about clinical studies
All manuscripts should comply with the ICMJE guidelines for publication of clinical research and a completed CONSORT checklist must be included with all submissions.

| Clinical trial registration | *Provide the trial registration number from ClinicalTrials.gov or an equivalent agency.* |
|---|---|
| Study protocol | *Note where the full trial protocol can be accessed OR if not available, explain why.* |
| Data collection | *Describe the settings and locales of data collection, noting the time periods of recruitment and data collection.* |
| Outcomes | *Describe how you pre-defined primary and secondary outcome measures and how you assessed these measures.* |

# Dual use research of concern

Policy information about dual use research of concern

## Hazards

Could the accidental, deliberate or reckless misuse of agents or technologies generated in the work, or the application of information presented in the manuscript, pose a threat to:

| No | Yes | |
|---|---|---|
| ☒ | ☐ | Public health |
| ☒ | ☐ | National security |
| ☒ | ☐ | Crops and/or livestock |
| ☒ | ☐ | Ecosystems |
| ☒ | ☐ | Any other significant area |

## Experiments of concern

Does the work involve any of these experiments of concern:

| No | Yes | |
|----|-----|--|
| ☒ | ☐ | Demonstrate how to render a vaccine ineffective |
| ☒ | ☐ | Confer resistance to therapeutically useful antibiotics or antiviral agents |
| ☒ | ☐ | Enhance the virulence of a pathogen or render a nonpathogen virulent |
| ☒ | ☐ | Increase transmissibility of a pathogen |
| ☒ | ☐ | Alter the host range of a pathogen |
| ☒ | ☐ | Enable evasion of diagnostic/detection modalities |
| ☒ | ☐ | Enable the weaponization of a biological agent or toxin |
| ☒ | ☐ | Any other potentially harmful combination of experiments and agents |

# Plants

| | |
|---|---|
| Seed stocks | *Report on the source of all seed stocks or other plant material used. If applicable, state the seed stock centre and catalogue number. If plant specimens were collected from the field, describe the collection location, date and sampling procedures.* |
| Novel plant genotypes | *Describe the methods by which all novel plant genotypes were produced. This includes those generated by transgenic approaches, gene editing, chemical/radiation-based mutagenesis and hybridization. For transgenic lines, describe the transformation method, the number of independent lines analyzed and the generation upon which experiments were performed. For gene-edited lines, describe the editor used, the endogenous sequence targeted for editing, the targeting guide RNA sequence (if applicable) and how the editor was applied.* |
| Authentication | *Describe any authentication procedures for each seed stock used or novel genotype generated. Describe any experiments used to assess the effect of a mutation and, where applicable, how potential secondary effects (e.g. second site T-DNA insertions, mosiacism, off-target gene editing) were examined.* |

# ChIP-seq

## Data deposition

☐ Confirm that both raw and final processed data have been deposited in a public database such as GEO.

☐ Confirm that you have deposited or provided access to graph files (e.g. BED files) for the called peaks.

| | |
|---|---|
| Data access links<br>*May remain private before publication.* | *For "Initial submission" or "Revised version" documents, provide reviewer access links.  For your "Final submission" document, provide a link to the deposited data.* |
| Files in database submission | *Provide a list of all files available in the database submission.* |
| Genome browser session<br>(e.g. UCSC) | *Provide a link to an anonymized genome browser session for "Initial submission" and "Revised version" documents only, to enable peer review.  Write "no longer applicable" for "Final submission" documents.* |

## Methodology

| | |
|---|---|
| Replicates | *Describe the experimental replicates, specifying number, type and replicate agreement.* |
| Sequencing depth | *Describe the sequencing depth for each experiment, providing the total number of reads, uniquely mapped reads, length of reads and whether they were paired- or single-end.* |
| Antibodies | *Describe the antibodies used for the ChIP-seq experiments; as applicable, provide supplier name, catalog number, clone name, and lot number.* |
| Peak calling parameters | *Specify the command line program and parameters used for read mapping and peak calling, including the ChIP, control and index files used.* |
| Data quality | *Describe the methods used to ensure data quality in full detail, including how many peaks are at FDR 5% and above 5-fold enrichment.* |
| Software | *Describe the software used to collect and analyze the ChIP-seq data. For custom code that has been deposited into a community repository, provide accession details.* |

# Flow Cytometry

## Plots

Confirm that:

☐ The axis labels state the marker and fluorochrome used (e.g. CD4-FITC).

☐ The axis scales are clearly visible. Include numbers along axes only for bottom left plot of group (a 'group' is an analysis of identical markers).

☐ All plots are contour plots with outliers or pseudocolor plots.

☐ A numerical value for number of cells or percentage (with statistics) is provided.

## Methodology

| | |
|---|---|
| Sample preparation | *Describe the sample preparation, detailing the biological source of the cells and any tissue processing steps used.* |
| Instrument | *Identify the instrument used for data collection, specifying make and model number.* |
| Software | *Describe the software used to collect and analyze the flow cytometry data. For custom code that has been deposited into a community repository, provide accession details.* |
| Cell population abundance | *Describe the abundance of the relevant cell populations within post-sort fractions, providing details on the purity of the samples and how it was determined.* |
| Gating strategy | *Describe the gating strategy used for all relevant experiments, specifying the preliminary FSC/SSC gates of the starting cell population, indicating where boundaries between "positive" and "negative" staining cell populations are defined.* |

☐ Tick this box to confirm that a figure exemplifying the gating strategy is provided in the Supplementary Information.

# Magnetic resonance imaging

## Experimental design

| | |
|---|---|
| Design type | *Indicate task or resting state; event-related or block design.* |
| Design specifications | *Specify the number of blocks, trials or experimental units per session and/or subject, and specify the length of each trial or block (if trials are blocked) and interval between trials.* |
| Behavioral performance measures | *State number and/or type of variables recorded (e.g. correct button press, response time) and what statistics were used to establish that the subjects were performing the task as expected (e.g. mean, range, and/or standard deviation across subjects).* |

## Acquisition

| | |
|---|---|
| Imaging type(s) | *Specify: functional, structural, diffusion, perfusion.* |
| Field strength | *Specify in Tesla* |
| Sequence & imaging parameters | *Specify the pulse sequence type (gradient echo, spin echo, etc.), imaging type (EPI, spiral, etc.), field of view, matrix size, slice thickness, orientation and TE/TR/flip angle.* |
| Area of acquisition | *State whether a whole brain scan was used OR define the area of acquisition, describing how the region was determined.* |

Diffusion MRI     ☐ Used     ☐ Not used

## Preprocessing

| | |
|---|---|
| Preprocessing software | *Provide detail on software version and revision number and on specific parameters (model/functions, brain extraction, segmentation, smoothing kernel size, etc.).* |
| Normalization | *If data were normalized/standardized, describe the approach(es): specify linear or non-linear and define image types used for transformation OR indicate that data were not normalized and explain rationale for lack of normalization.* |
| Normalization template | *Describe the template used for normalization/transformation, specifying subject space or group standardized space (e.g. original Talairach, MNI305, ICBM152) OR indicate that the data were not normalized.* |
| Noise and artifact removal | *Describe your procedure(s) for artifact and structured noise removal, specifying motion parameters, tissue signals and physiological signals (heart rate, respiration).* |

| Volume censoring | *Define your software and/or method and criteria for volume censoring, and state the extent of such censoring.* |

## Statistical modeling & inference

| Model type and settings | *Specify type (mass univariate, multivariate, RSA, predictive, etc.) and describe essential details of the model at the first and second levels (e.g. fixed, random or mixed effects; drift or auto-correlation).* |

| Effect(s) tested | *Define precise effect in terms of the task or stimulus conditions instead of psychological concepts and indicate whether ANOVA or factorial designs were used.* |

Specify type of analysis: ☐ Whole brain ☐ ROI-based ☐ Both

| Statistic type for inference

(See Eklund et al. 2016) | *Specify voxel-wise or cluster-wise and report all relevant parameters for cluster-wise methods.* |

| Correction | *Describe the type of correction and how it is obtained for multiple comparisons (e.g. FWE, FDR, permutation or Monte Carlo).* |

## Models & analysis

| n/a | Involved in the study |
|---|---|
| ☐ | ☐ Functional and/or effective connectivity |
| ☐ | ☐ Graph analysis |
| ☐ | ☐ Multivariate modeling or predictive analysis |

| Functional and/or effective connectivity | *Report the measures of dependence used and the model details (e.g. Pearson correlation, partial correlation, mutual information).* |

| Graph analysis | *Report the dependent variable and connectivity measure, specifying weighted graph or binarized graph, subject- or group-level, and the global and/or node summaries used (e.g. clustering coefficient, efficiency, etc.).* |

| Multivariate modeling and predictive analysis | *Specify independent variables, features extraction and dimension reduction, model, training and evaluation metrics.* |

