## [Peer Review file · Nature]

Mosaic anatomy in an early fossil squamate

Corresponding Author: Professor Roger Benson

Version 0:

Reviewer comments:

Referee #1

(Remarks to the Author)

The manuscript, entitled "Mosaic anatomy in an early fossil squamate," presents the description of a new Middle Jurassic squamate, *Breugnathair elgolensis*, from the Isle of Skye in Scotland. The specimen is confidently assigned to "parviraptorids," a group previously known only from highly fragmentary material, mostly comprising isolated cranial elements. Parviraptorids have long occupied a contentious position in discussions of squamate affinities and, more recently, in debates concerning the origin of snakes. Initially interpreted by some authors as stem squamates or anguimorphs, these fossils have more recently been linked to snakes due to a handful of jaw and dental characters, and proposed as a means of bridging a critical temporal gap in the early snake fossil record, extending their presence into the Jurassic. However, the limited and poorly preserved nature of most available specimens has significantly constrained the robustness of such interpretations. This issue has been further complicated by recent claims that the only relatively complete specimen attributed to this group may, in fact, be a chimaera—comprising cranial and postcranial elements from different, and possibly distantly related, taxa. Conversely, critics of a snake affinity have emphasized that reliance on a few snake-like traits—while disregarding postcranial anatomy that often appears inconsistent with snake morphology—casts doubt on the relevance of parviraptorids to snake evolution, leading to renewed scrutiny of their phylogenetic placement.

In this context, the new specimen described in the present manuscript constitutes a major advance. It provides key anatomical elements that decisively resolve previous uncertainties, effectively falsifying claims of chimaeric associations in known parviraptorid material. The authors argue that *Breugnathair* exhibits a striking mosaic of anatomical traits, combining primitive features, snake-like dental and jaw structures, and head-body proportions reminiscent of varanids. This combination is interpreted as strong evidence of widespread homoplasy and ecological experimentation during the early evolutionary radiation of squamates. The manuscript successfully addresses persistent conflicts of anatomical interpretations and contributes critical new data to the ongoing debate on squamate diversification and the origins of snakes. Given the significance of the material, the quality of the anatomical analysis, and the broader implications of the findings, I fully endorse the results presented in this study and strongly recommend its publication in *Nature*, pending minor revisions and incorporation of the suggestions detailed below.

Hussam Zaher

Lines 92, 111. Change *Diabolophis* to *Diablophis*.

Line 192 "present IN stem lepidosaurs, ...

Lines 223-225: The authors may consider citing Zaher et al. (2022), who reviewed the loss of the metakinetic joint in snakes in the context of the braincase anatomy of the stem-snake *Sanajeh indicus*. Including this reference would provide additional context and strengthen the discussion of cranial kinesis evolution in early snakes.

Lines 231-234: Contrary to the statement presented by the authors, the development of a deep lamina of the prearticular that walls the adductor fossa medially appears to be restricted to crown-Alethinophidia. In contrast, early fossil snakes such as *Najash* and *Dinilyisia* retain the plesiomorphic condition of a shallow adductor fossa, similar to that of non-snake squamates (Garberoglio et al., 2019; Zaher & Scanferla, 2012). I suggest revising this portion of the text.

Lines 237-240: a pleurodont tooth implantation was also reported in stem-snakes *Najash* (Zaher et al., 2009) and *Sanajeh* (Zaher et al., 2022), and in crown-snake scolecophidians (Zaher and Rieppel, 1999).

Line 267 change Hypopophysis to hypapophysis.

Line 736: change parviraptorids to parviraptorids, and toxicoferances to toxicoferans.

Lines 736-738: The sentence "Because analyses of this matrix returned parviraptorids as toxicoferans, we ran three separate analyses of this dataset using different phylogenetic hypotheses for Toxicoferan ingroup relationships" would benefit from clarification. Specifically, could the authors elaborate on what is meant by "different phylogenetic hypotheses"? Were the three analyses conducted under topological constraints reflecting alternative toxicoferan relationships? And how was that accommodated with the larger constraint for the ingroup relationship of squamates mentioned in lines 745-748. Please specify the methodological approach used to implement these constraints and how they were derived.

Line 886: ..., and so it WILL not be discussed further ...

Lines 1014-1015: change Diabolophis to Diablophis

Referee #2

(Remarks to the Author)

Revision of Mosaic anatomy in an early fossil squamate by Benson et al.

This paper describes an incredible new fossil that sheds light to 1) Early evolution of Squamata, 2) the complex nature of the new defined family Parviraptoridae.

This paper is of interest for a general audience and is clearly presented and illustrated. The authors made an effort to test the position of this family by retrofitting this new taxon in three existing data sets.

I have some comments that can be considered to improve the paper, but the authors are free to include them or not.

1) The preservation of Breugnathair is very similar to the original block of Parviraptor. I am not an expert on taphonomy, but maybe the authors can add a section of taphonomy and discuss the similarity of the preservation of these taxa? Squamates are generally known by isolated elements and some rare articulated fossils (hard rock, or amber embedded), but it is curious the disarticulated, but complete nature of these fossils. Could similar depositional environment of the two fossils be interpreted as a similar ecological environment for Parviraptorids?

2) I made a comment on the phylogenetic part, and I still think that the authors exhausted the resources to test the position of Parviraptorids. I still think the data provided in Caldwell et al. (<https://www.nature.com/articles/ncomms6996>) should be either explored in the analyses or at least comment why these were not included.

3) Can you provide also a dorsal view of the articulated skull?

After reading the article, I ended a little bit disappointed that the position of Parviraptorids might remain still uncertain, but it is also common that fossils that show mixed traits are found in such alternative positions. Oculudentavis could be an interesting fossil to add, as it exhibits a similar behavior (Basal or with mosasaurs), at least in the Gauthier et al data set (Modified in Meyer et al.)

Specific comments:

Line 23: 11989 according to the Reptile data base

Line 28: although the ICZN don't have a rule that prevents writing species names in the abstract, it is generally not advisable to do so, cause it makes uncertain the place in the publication where the new species is described

Line 30: maybe say, The new taxon is placed in a new Squamate family, Parviraptoridae, an... Or something like that.

Line 34: add: using different data sets

Line 37, maybe convergence sound better than homoplasy?

Line 43: Some molecular estimates of this divergence to 250 million years, or the end of the Permian. See:

<https://www.nature.com/articles/s41586-020-2561-9>

Line 51: Add Megachirella wachtlei to this list?

<https://www.nature.com/articles/s41586-018-0093-3>

Line 71: Since the position of snakes in the combined analyses is generally stable within Toxicofera, I think the main issue is the mosaic nature of Parviraptorids.

Line 75, instead of constrains use maybe use: narrows down their phylogenetic position. Constrain in systematics refers mainly to ad hoc decision to force some groups .

Line 84: Since Lizard is pretty much a name that apply to Squamates (See Camp 1923), perhaps don't include Lizard-like squamates. Can you could just say Limbed squamates?

Line 85: smooth instead on unsculptured?

Line 98: something seems wrong on this sentence, it is defining the Formation and location, but also that is of special scientific interest. Is that the full name of the locality?

Line 113: Are you adding a justification for the nomen dubium below, if so, maybe add See below?

Line 125 methods: are these references to sessions required by the journal?

I also noticed that citations are numbered and in the author year format.

Line 133: make sure you clarify the numbers of vertebral regions.

Line 153: Is this difference on onset of humeral epiphyses, also seen in other limb bones? Could this be a fracture?

Line 155: maybe change to basicranium?

Line 158: A good reference for this would be: Stephenson NG. 1962. The comparative morphology of the head skeleton, girdles and hind limbs in the Pygopodidae. Journal of the Linnean Society (Zoology). 44: 627-644.

I also see your point, of all Gekkotans, *Lialis* is more comparable, however the similarities are just superficial. *Lialis* has smaller and more teeth, narrower skulls, different snouts, different palates, and different basicranium, just to cite a few.

Line 169: True, but comparisons would make more sense to Early Pan-Gekkotans, such as *Norellius* and *Eichstaettisaurus* who have the foramen.

Line 170, see my comment on smooth. Also most gekkotans have smooth parietals see: Elizabeth Glynne, Juan D Daza, Aaron M Bauer, Surface sculpturing in the skull of gecko lizards (Squamata: Gekkota), *Biological Journal of the Linnean Society*, Volume 131, Issue 4, December 2020, Pages 801–813, <https://doi.org/10.1093/biolinnean/blaa144>

Line 188: *Norellius* also have palatine teeth.

Line 207: Maybe use non-ophidian squamates, which seems to be more widely used.

Line 207: basicranium

Line 209: to be consistent with the figures, use alary process?

Line 211: This is a great difference with snakes

Line 213: I cannot tell if this is lack of definition or poorly preserved.

Line 219: Maybe mention that this is also the case of many head first burrowers eg. amphisbaenians, dibamids,

Line 238: add this reference after Edmund: Zaher H, Rieppel O. 1999. Tooth implantation and replacement in squamates, with special reference to mosasaur lizards and snakes. *American Museum Novitates* 3271: 1-19.

Line 283: The selection of data sets is appropriate to test the position of Parviraptorids, however, I have a recommendation: here you should either write a sentence of why the scores for the maxilla of Parviraptor done by Caldwell et al. (<https://www.nature.com/articles/ncomms6996>) and added to the Gauthier et al data set were not considered, are these not compatible with the data set of Meyer et al? Perhaps you can do an additional run including these scores? Maybe this could be added to the supplementary material? The authors might disagree with this idea, but I know from personal experience that the group of Caldwell states all the time that their data is ignored. If this paper is published, what is going to happen is that they will re-score the data and provide a new analyses. It is hard to predict the results of this, but they can use argument to that Parviraptor is just a snake.

Line 285: reword: But we recovered this group in different positions in the squamate tree.

Line 494: ?

Line 581: You stated that there are 32 vertebrae preserved, but from these 24 are attributed to presacral. Here you estimated 30 presacral, are the 2 additional sacral or caudal? Also 30 presacral is in the higher end of many limbed squamates, groups, this number excludes Iguanians, limbed cordylids, Teiids and Gymnophthalmids. See: Daza JD, Bauer AM, Stanley EL, Bolet A, Dickson B**, Losos JB. 2018. An enigmatic miniaturized and attenuate complete lizard from the Mid-Cretaceous amber of Myanmar. *Breviora* 563:1–18

Referee #3

(Remarks to the Author)

I have reviewed the manuscript entitled "Mosaic anatomy in an early fossil squamate", by Professor Benson et al., under consideration for publication in *Nature*.

A. Summary of key results: The study deals with the description of a disarticulated specimen (thus, corresponding to associated elements of a single specimen) of a squamate from the middle Jurassic of the UK, through high resolution CT-Scanning. The authors use this specimen as the holotype of a new genus of parviraptorid, a family of squamates that they formally erect here after being informally used in a series of papers and that is important in having been regarded in previous studies as the oldest snakes in the fossil record. Besides being one of the best preserved squamate specimens on record (only surpassed in preservation by one specimen of *Bellairsia* described by some of the same authors of the present manuscript in a recent publication in *Nature*), the specimen is highly interesting in showing that the reasons for considering previous material of Parviraptor a chimaera were wrong, and that a former interpretation of Parviraptor and other parviraptorids as stem-snakes (and thus, the oldest snakes on record)(Caldwell et al., 2015), were biased by the election of removing those elements that did not have snake features from the taxon.

B. Originality and significance: The specimen is highly interesting in showing that 1) the claimed chimaeric nature of other parviraptorids is highly unlikely; 2) that at the very least the new genus of parviraptorid *Breugnathair* had well-developed limbs; and 3) that parviraptorids consistently show a mixture of snake-like characters (mostly related to macropredatory feeding) and primitive squamate features.

C. Data and methodology: The paper is excellently written and figures, which mostly rely on the high resolution CT-scans performed, including those performed through Synchrotron, are nice and highly informative.

D. Appropriate use of statistics and treatment of uncertainties: The methodologies used are appropriate, and the authors performed up to three different morphological matrices in order to test different phylogenetic schemes. They provide support values for the nodes, and use a rather conservative consensus tree (50% majority rule consensus, which is standard in Bayesian analyses).

E. Conclusions: As a summary of my assessment of the paper results, I can say that I agree with the recognition of *Breugnathair* as a new genus of parviraptorid and with the erection of parviraptoridae. The authors provide three alternative positions for parviraptorids: 1) stem-snakes (which would confirm the position proposed by Caldwell et al. 2015); 2) forming a polytomy with *Anguimorpha* and *Serpentes*, which would not regard them as stem-snakes but would confirm them as toxicoferans anyway; and 3) as stem-squamates. Although several decisions made through the manuscript seem to suggest they favour a non-stem-snake position for parviraptorids, the truth is that the final section of the manuscript is somewhat equidistant between possibilities. I think that being slightly more assertive in defending the stem-squamate position would have been clearer to the reader (if that position was actually favoured by the authors). The importance of the paper beyond the simple addition of a new genus of fossil squamate relies on 1) the ancient age of the deposits where it was found, because there are few confirmed squamates of this age, and only one contemporaneous genus (described by the same authors) is known from a more complete specimen; 2) parviraptorids, claimed to be stem snakes by Caldwell et al. (2015) after separation of elements associated in the same block as belonging to different taxa, and selection of only the snake-like bones, are here shown to represent instead squamates with a mosaic of features, some of which are ?convergently shared with snakes. The truth is that the authors are a bit inconsistent in their interpretation, because some decisions made seem to reveal that they favor interpretation of parviraptorids as stem-squamates (or at the very least, as forms not closely related to snakes), whereas other parts of the manuscript seem to try to remain equidistant between the three options (stem-squamate, stem-snake, or more broadly toxicoferan, see my comments below). Although I personally would favor interpretation of parviraptorids as not closely related to snakes, mainly in the view of the presence of well-developed limbs and other characters described by the authors, what worries me more is this slight lack of consistency. Note that removing parviraptorids from the snake stem would have deep consequences for our understanding of the evolutionary origins and timing of divergence of not only snakes, but also of toxicoferans (snakes, iguanians and anguimorphs) as a whole, because parviraptorids represented the oldest fossil record of Toxicofera. *Dorsetisaurus*, which would be the oldest toxicoferan if parviraptorids are removed from the clade, is not mentioned in the manuscript. The implications of this particular view, where parviraptorids are not toxicoferans or, at least, not stem-snakes, concern anyone involved in the evolution of squamates as a whole. It also hints at the importance of mid and late Jurassic squamates as examples of morphologies that do not easily fit any of the crown groups, something that is not new but is shown more clearly in the specimen described here than in others.

F. Suggested improvements: Although I regard the study as very well executed and of great interest, not just to paleoherpetologists, but also to those interested in the evolution of Mesozoic faunas in general, the manuscript is not free of places that could get benefited of some review. More details are provided below, but aspects to be improved include: splitting the results of phylogenetic analyses shown in fig. 4A into different trees (or, alternatively, adding a warning about the fact that parts of the depicted topology are not 100% consistent with the obtained results and that the figure should be regarded as an informal tree). Additionally, the readers should be referred to the actual trees in figures S7-S9 to interpret the results; adding descriptions of elements of *Breugnathair* that are discussed in the text, but not described; adding some more weight on the importance of the possible reinterpretation of parviraptorids as unrelated to snakes; maybe checking the synapomorphies supporting nodes of interest (or at the very least, check them for additional information that could be of interest in diagnosing *Breugnathair* in particular or parviraptorids in general, or that could help in deciding which of the alternative phylogenetic positions is more likely; and, finally, a series of minor observations (see my comments below).

G. References. The manuscript provides appropriate credit to previous work in form of numerous and adequate references.

H. Clarity and context: The writing in the manuscript is excellent, easy to follow, and very clear in all of its sections.

To sum up, I strongly recommend the publication of this work, after careful consideration of comments raised in this review that I hope will help the authors improve a very interesting and strong manuscript.

Specific comments:

Abstract. I am not sure that the expression early diverging toxicoferan is unambiguously conveying what the authors mean. They clearly mean a toxicoferan that is recorded early in the fossil record, but an early diverging toxicoferan can also be interpreted as one that diverges early along the evolution of toxicoferans (i.e. a stem toxicoferan) which is clearly not what they mean here. They might want to rephrase to avoid such ambiguity.

There is some sense of contradiction in the fact that the authors seem to be much more cautious in their interpretation of the group placement (with e.g. sentences like “This may provide evidence of mosaic evolutionary changes along the snake stem lineage, or that parviraptorids are not closely related to snakes, but independently evolved some snake-like features”), than with the election of the name. What I mean is that in some parts of the manuscript it is rather clear that they do not favor a stem-snake position, but in others they do not seem to take parts between conflicting interpretations. Keeping a name that means “false snake” is perfectly fine, but it seems to be in slight contradiction with the decision made by the authors of avoiding taking sides between the alternative positions of parviraptorids in some parts of the manuscript. If the authors favor one of the interpretations, I recommend that they express it with more confidence, in order to avoid this sensation of inconsistency. The title feels, again, slightly ambiguous. I would tend to interpret the mosaic nature of the anatomy of parviraptorids (particularly *Breugnathair*, the only for which the presence of limbs has been demonstrated), as a mixture of genuinely plesiomorphic characters (consistent with a placement on the stem of Squamata) with derived snake-like characters acquired through convergence, possibly in relation to similar feeding habits. I guess that the alternative scenario, which would involve true snake characters mixed with plesiomorphic characters (at least at the level of Toxicofera) is still possible to be interpreted as a mosaic anatomy, but I think that in that case researchers tend to interpret retention of plesiomorphic characters as something expected from an early-diverging form of a lineage (in this case, snakes), and not truly a mosaic anatomy. My point here is that, again, the authors are not being too explicit in expressing an interpretation that underlies some of the decisions made, and that should be more clearly and consistently expressed.

Line 92, *Diablophis* is misspelled *Diabolophis* here and in lines 111, 1013 and 1014.

Line 94. The taxon name Parviraptoridae has been previously informally used (e.g. Caldwell et al., 2015; Davis et al., 2018; Head et al., 2022), also in the form of parviraptorids (e.g. Pancirolli et al., 2020; Klein et al., 2021) or “parviraptorids” (e.g. Caldwell, 2020; Fachini et al., 2020; Marjanović, 2021; Zaher et al., 2023; Paulina-Carabajal et al., 2023; Čerňanský et al., 2023) although the clade had not formally been erected. The authors state this previous use of the term in the supplementary text, but it might be worth mentioning this as a note placed between the family composition and the new genus name, or at

least somewhere in the main text.

Line 109: closing parenthesis missing after S2

Line 150. Compound bone is sometimes between quotes and sometimes not, please be consistent with one or the other.

Line 168: "The parietals share features with early-diverging squamate groups, including the presence of a parietal foramen, which is absent in snakes and crown gekkotans (Fig. 3C)." Yes, but this is rather widespread, not only shared with early-diverging squamates.

Line 171: the authors probably mean xantusiid AND anguimorphs, not xantusiid anguimorphs

Line 188: regarding the sentence "Palatine teeth are also present stem lepidosaurs, rhynchocephalians and some early-diverging squamates (28,31,34 Whiteside 1986; Evans & Wang 2005; Griffiths et al. 2021) as well as most snakes and some anguimorphs (30 Cundall and Irish, 2008).", they are also variably present in several iguanians (see Mahler, L., Kearney, M. "The Palatal Dentition in Squamate Reptiles: Morphology, Development, Attachment, and Replacement," *Fieldiana Zoology*, 2006(108), 1-61).

Line 192: "in" is missing between "present" and "stem"

Line 246: "The marginal teeth of NMS G.2023.7.1 are conical and strongly recurved (Figs 3A, S6), similar overall to those of other parviraptorids (8,9 Evans 1994; Caldwell et al. 2015) as well as many snakes (8,29 Gauthier et al. 2012; Caldwell et al. 2015), mosasauroids (21 Russell 1967), amphisbaenians (41 Gans & Montero 2008), the pygopodid *Lialis* (42 Lordansky 2004), and some living (e.g. 20,23,32 *Anguis fragilis* Klembara et al. 2014; *Heloderma horridum*, many *Varanus*; Evans 2008; The Deep Scaly Project 2002–2019) and extinct (43 Norell et al. 1992) anguimorphs." Although the teeth in amphisbaenians can be considered conical and slightly recurved, they are in general very different from the rest of taxa mentioned here (they are much less recurved, closer to the condition in typical non-snake squamates). If what the authors are trying is to establish a comparison of parviraptorid with other squamates with typical features of macropredators, I would not include amphisbaenians here. If the authors want to keep amphisbaenians in the comparison, they should state "some amphisbaenians", because others (e.g. trogonophids) do not have recurved teeth at all. On the other hand, they might want to add the extant scincoid *Sphenomorphus muelleri* (see Kosma R. 2004. The dentitions of recent and fossil scinciform lizards (Lacertilia, Squamata) - systematics, functional morphology, paleoecology. PhD dissertation, Hannover (Germany): University of Hannover, p. 187.) as an example of a squamate with recurved teeth (this taxon approaches more closely the dental morphology *Lialis* than amphisbaenians in general).

Line 254. No mention to the overall snake-like look of the vertebral condyle, which is in some cases almost perfectly spherical.

Some cranial bones (e.g. the squamosal, postfrontal) and most limb and girdle bones are mentioned in the main text, but unless I missed them, they are never described. Particularly in the case of the postcranial elements, which are extremely interesting in showing that at least this parviraptorid did not have a snake-like (limbless) postcranium, it is weird that there is no description at all. The same applies to the vertebrae. Do the caudals show a fracture plane for autotomy? Autotomy was inferred for Parviraptor on the basis of the fact that many caudals represented only one half of the vertebra (presumably due to split through the autotomal plane), so CT-scans slices would be a good place to check for such planes in both Breugnathair and the CT-Scanned Parviraptor (if caudals are preserved in the latter).

Line 287. Regarding the sentence "We find parviraptorids either as stem squamates (in Datasets 1 and 2; Fig. S7), or as either stem snakes or on the stem of anguimorphs plus snakes (in Dataset 3; Fig. S8–S9)", the latest statement is not entirely true based on what is reported in the complete trees (Fig. S8-S9). What the trees show is a polytomy between parviraptorids, anguimorphs and snakes. One would need to check the original trees (prior to calculation of the consensus) in order to check which are the real positions, which I suspect would be related to anguimorphs in some trees and related to snakes in others. A position on the stem of Anguimorpha+Snakes is still possible, but not necessarily true, but I cannot check because I do not have access to the trees resulting from the full length analysis. In any case, what is stated (that the trees in figs. S8-S9 recover parviraptorids on the stem of anguimorphs plus snakes) in the text is inaccurate, and this same issue is replicated when summarizing the tree in Fig. 4a.

Line 297. "Current analyses therefore suggest that crown squamates are not known with certainty until the Late Jurassic (e.g. 4,7,45,48,49,50 Simões et al. 2018; Meyer et al. 2022; Talanda et al. 2022; Whiteside et al. 2022, 2024; Brownstein et al. 2023)." What about previous claims of the presence of paramacellodids in middle Jurassic assemblages? I know that they are too fragmentary to be included in phylogenetic analyses but, paramacellodids are often regarded as crown squamates (in particular related to scincoids). Do the authors think that they could actually represent stem squamates too? I see the position of *Eoscincus* in the results of the present manuscript, but as I say in another part of my review, I am not too confident that a clade formed by *Hongshanxi*, *Dorsetisaurus*, *Eoscincus* and parviraptorids is a real clade, because these taxa are very different from one another.

Line 332. Regarding "(3) the presence of palatine teeth, seen also in squamate outgroups (31,34 Evans 1980; Whiteside 1986; Griffiths et al. 2021) as well as some early-diverging squamates (28 Evans & Wang 2005)", as the authors themselves state in another sentence, palatine teeth are also present in some anguimorphs (not mentioned here), and I also stated in one of my comments above that some iguanians variably present palatine teeth too.

Lines 326-338. I think that of the 5 characters supposed to represent characters "shared with both snakes and with some stem squamates or early-diverging squamate groups such as gekkotans", some (3, 4, and 5) are shared also with other members of crown groups (e.g. anguimorphs, iguanians), and this might need to be stated somewhere.

Line 545. "and" is missing between "ribs," and "a"

Line 580. Details on which taxa were used for the reconstruction are given, but what about the manus and pes? Are they based on any information of the fossil? If not, is it based on any extant taxon, and which one (possibly varanids?)?

Line 606. The question mark besides postfrontal seems to convey a lack of confidence on identification, but it is missing in another part of the manuscript (line 542).

Line 634. Check "adjusting the 63 parameters"

Line 663. Check "the overlapping part. 64 Both final"

Fig. 3. The postfrontal is the only bone that is not labeled.

Fig. 3 caption. I would place a mention stating that some bones have been mirrored to create copies of the complementary

side, and specify which ones.

Only the new taxon and two OTU's for Parviraptor are included in the analyses. What about Portugalophis and Diablophis? Since they are considered parviraptorids by including them in the erected family, the reason for not including them should be reported.

Line 735, "toxicoferans", not "toxicoferances"; "parviraptorids" instead of "parvriaptorids"

Line 798. Since the summary of phylogenetic results includes the positions for Dataset 3, the caption should read Figs S7-S9, not S6-S7. See however, my comment about the fact that I have doubts about using a single tree to show the results of analyses with different constraints (this is, the two stars labeled as Dataset 3 are marking two positions, but it is not possible that they both came from a tree with the same relationships between Iguania, Anguimorpha and Serpentes, because the point of running three different analyses was to force such relationships to be different. At the very least, if the authors want to keep this particular tree as the main figure for the phylogenetic analyses results, a warning stating that this is an informal tree and any interpretation of the results should be made on the supplementary trees (figs. S7-S9) should be added.

Line 800. "and" should possibly be deleted, or the sentence needs rewording.

Fig. S6. The label pointing at "two replacement teeth" should be better placed in G, rather than in E, because they are better seen in the former view.

Fig. S7-S9. These trees are apparently timescaled (as a result of the use of tip-dating), but a numerical time scale (or labelling) is not provided.

Fig. S8A. Where the constraints are stated, it should read Iguania, not Iguana.

Line 882. The authors state: "Taxonomic rank (as Parviraptor or parviraptorids) is a matter of subjective judgement that cannot be resolved objectively, but also has no impact on phylogenetic relationships or their implications for squamate evolution, and so it not discussed further here." However, they formally erect Parviraptoridae in the manuscript, what seems to be in slight contradiction (they actually confirm and adopt Caldwell et al.'s interpretation, at least regarding the composition of the clade, not necessarily the interpretation of its position if in the squamate tree).

Line 885. Caldwell et al. is sometimes cited as 2015, and sometimes as 2014. In most cases this does not matter because such citations will end up replaced by the corresponding reference number, but the authors should check that there are no places where this could generate a problem and, particularly, make sure that these two different years are not given different reference numbers. Also, at least on one instance (line 929), Caldwell et al. is given the reference number 9, when it should be 8; and in another one, line 971-972, the reference has no number. Evans (1994) seems to present a similar problem, where it is sometimes given reference number 9 (most cases), but sporadically number 8 (e.g. line 190).

line 895. "and group of skull bones" should read "a group of skull bones..."

Line 896. I think that the "," should be a "."

Line 900. Note that in other parts of the supplementary text, the links are in place, so the call to (...links...) should probably be replaced by the actual links

Line 908. "show" instead of "shows".

Line 918. Check if the "but" should be removed

Line 984. Can the authors point to somewhere that justifies the statement about slow growth? Maybe a specific part of the text dealing with the osteohistology like line 142, and/or cite Evans (1994)?

Line 988. I agree with the statement about Eophis underwoodi as a nomen dubium, but it might be worth explicitly stating to which taxon in particular are these specimens finally referred (in the caption of Fig. S3 only parviraptorid bones are mentioned, so this seems to suggest that the intended referral in the present study for these specimens would be Parviraptoridae indet., but this would be a change from the original referral to Parviraptor cf. estesi by Evans (1994), which is not dealt with in the text.

Lines 994-1004. Although the fact that they regarded Eophis (but not Portugalophis or Diablophis) as nomen dubium, and the inclusion of the latter in the composition of parviraptoridae, make me think that the authors support both Diablophis and Portugalophis as separate genera from Parviraptor, it would possibly be worth adding a statement in this regard.

Line 1033. Remodelling, not remodeling

Figure 1. Unless I am getting it wrong, the labels for procoelous on the vertebrae are pointing at the cotyle (concave anterior surface) in Fig1E and Fig1I. Since what makes a vertebra procoelous is the presence of a condyle (convex posterior surface), I would rather point the label at the condyle of either Fig1C or Fig1F and of either Fig1G or Fig1H. The same applies to figS5D.

Figure 3. Is it possible that what the authors have labeled as ectopterygoid fossa is actually the epipterygoid fossa? This would fit better with the scoring of the epipterygoid in both P. estesi and Breugnathair as present (although the dorsal view of the pterygoid is not figured for the latter, and there is no mention to such structure in the text).

Figure 4. As I said before, I have a couple of problems with the tree corresponding to figure 4A. One is that, as currently displayed, what are labeled as stem squamates are actually drawn in a polytomy with crown squamates. This is what happens in fig. S7A, but not in the rest of trees, and drives me to a more profound problem, which is if it is acceptable to draw the multiple positions of parviraptorids in a topology that does not fit all the results. I mean, it is not the same to draw two alternative positions of a wildcard taxon in a tree that is, for the rest of taxa, fundamentally the same (at least when taxa are collapsed into large groups), than to select one of the topologies and place all the alternative positions there. The problem is that, by doing that, the authors are showing relationships that do not necessarily occur in the results. Just as an example, the position of parviraptorids as stem snakes occurs in figures S8B and S9, but in these figures Anguimorpha and Serpentes are not sister taxa as shown in Fig. 4A. Actually, fig. S8B shows the tree resulting from an analysis where Anguimorpha and Iguania are forced to be sister taxa, and in fig. S9, Iguania is forced to be sister of Serpentes, even if such relationships are not evident in the final tree because the movement of other taxa (possibly fossils, which are not included in the constraints) create a huge polytomy including Iguania and Anguimorpha. The topology regarding toxicoferan internal relationships shown in Fig. 4A corresponds to one of the results, the other two having different combinations of relationships for these three clades, so while one of the stars shows a true result of an analysis, the other star never happened in this particular topology (it happened in a topology that is completely different). In my opinion, this can be fixed in two different ways: one would be to use a different tree for each of the positions; the other one would be to place a warning in the figure stating that

the different positions are placed in one particular topology, indicate which one, and place a warning that this is a simplification of the actual relationships recovered, and point towards the actual results in figures S7-S9.

Figure S6. Since the retroarticular process is scored in the matrix as present, it is worth labelling in this figure, and maybe add a mention to its presence in the text too.

Figures S7-S9. I am not sure if this is done on purpose, but it is weird that the OTU's in the trees correspond to the specimen number, and not to the taxon name to which such specimens are referred. This would be more justifiable in the case of specimens referred with low confidence, but it is particularly weird in the case of Breagnathair, because it results in trees where the new taxon is not easy to find. Why not naming them Breagnathair, Parviraptor estesi (holotype) and Parviraptor estesi (referred specimen number xxxx) that would be much clearer? I think that naming them with the taxon name (and maybe Breagnathair in bold or signaled somehow) would improve readability. Also, Anguimorpha is labeled in just one of the trees. I think that there is space to label Iguania and Anguimorpha in all trees, but if the authors do not want to do it, it might be worth being consistent and remove the label for Anguimorpha from FigS8A.

Just for figure S8A. Although Fig. 4A depicts Parviraptoridae on the stem of Anguimorpha+Serpentes, what the majority rule consensus tree shows in Fig. S8A is a polytomy between parviraptorids, anguimorphs and snakes. It is not possible for me to check the alternative positions of parviraptorids because the authors did not provide the resulting trees. It is quite important that the authors confirm that parviraptorids are actually recovered on the stem of Anguimorpha+Serpentes, and not just as sister to Anguimorphs in some trees, and sister to Serpentes in others (what would effectively place them in a polytomy with them, as shown in Fig. S8A, but not on the stem as shown in Fig.4A). Related to this, have the authors checked the allcomp consensus tree? Just to see which relationships are favoured in the presented polytomies. This might be particularly informative for the polytomy of Fig. S8A.

There is not a single actual photograph of the new specimen. Although this is not mandatory, I would recommend providing at least one photograph (as a supplementary figure) of the specimen, where one can see the colors and other properties of both the bones and surrounding rock matrix.

There is a mention to the deletion of Tetrapodophis in the statement report (and at least there is a line in the scripts of dataset3 that effectively deactivates the taxon), but this is not mentioned in the methods. Similarly, Taytalura is mentioned as an addition to datasets 2 and 3, but it is deactivated in the scripts of such datasets, so it is not effectively used. This should be mentioned in the methods.

Of the three scripts for dataset3, in the script where the constraints enforce the monophyly of snakes+anguimorphs to the exclusion of iguanians, the second line reads "constraint:snakes+anguimorphs to the exclusion of anguimorphs". The second instance where anguimorphs is written, it should read iguanians instead. This has no effect on the results of the analyses because this is not a line of code, but an informative statement describing which constraint will be enforced below, but I thought that the authors might want to fix this anyways.

After the authors replaced the old matrices by the final updated ones I ran short versions of the analyses and everything went smoothly. My short versions of the analyses recover results that are in line with the full-length analyses results reported in the manuscript (recovering parviraptorids as stem-snakes in some occasions), but not the polytomy reported by the authors in fig. S8A). However, this is possibly related to the fact that my analyses did not run long enough to reach convergence. This is unfortunate, because I haven't been able to check the exact positions of parviraptorids that result in this polytomy (something that the authors should be able to do with the .t files of their results). This is important because such positions might contradict what is stated in the text (see my other comments regarding this).

I checked the scorings for parviraptorids, and I only have some minor questions:

1) At least in Dataset 1, both the postorbital and postfrontal are scored as present (and in Dataset 2 are scored as separate).

A bone in the approximate position of those bones appears in fig. 3A but it is not labelled and only the postfrontal is mentioned in the text (line 543). Other places where the postfrontal is mentioned, it is accompanied by a ? (line 607). If the presence of the postorbital is scored indirectly (e.g., inferred on the basis of the presence of a facet on the postfrontal/parietal), this should be mentioned in the text. Even in that case, the authors should look for inconsistencies in how the identification of the postfrontal is expressed in the text (with or without a question mark).

2) There are other instances where the score is possibly assessed indirectly (e.g., char 68 of dataset 1, the presence of frontal parietal tabs in NMS G.2023.7.1, a specimen that does not preserve the frontals, and the presence of which is possibly deduced from facets on the parietal). The same occurs with the presence of the ectopterygoid (char. 113 of dataset 1) or the epipterygoid (char 115 of dataset1). It might be worth mentioning in the text when such indirect scorings are used, mainly in those cases (e.g. presence of the epipterygoid) in which the character state is not shared with snakes.

Breagnathair is scored as lacking contact between ectopterygoid and maxilla (dataset 1, char 362), but how is this known if none of these elements is preserved in NMS G.2023.7.1?

3) The scored presence of frontal tabs in the parietal of NMS G.2023.7.1 (char 78, dataset 1) is not mentioned in the description of the element.

4) In character 123 of dataset 2: Maxilla_pivots_on_prefrontal_to_erect_fang / absent present, none of the taxa are scored 1, and only a few of them (Paliguana, Limnoscansor and Magnuviator) are scored ?. I know this is a snake character, but how the authors know about this state in Breagnathair if the maxilla and prefrontal are not preserved? The same applies to other maxilla characters.

5) How is the quadrate lateral conch (Dataset 2, character 188) scored as absent (state 1) if the quadrate is not preserved in Breagnathair?

6) In Dataset 2, characters 373 and 374, if Dentary_subdental_gutter_is scored absent, then character 374 (Dentary_subdental_gutter_development) should be unscorable (but see my comment 11 below).

7) The coronoid is scored as present in dataset 2 (character 404) and other datasets, but this bone (or corresponding facets on other lower jaw bones) are not mentioned in the text for Breagnathair.

8) In Dataset 2, character 583: 583 '570. Osteoderms on body (and/or tail) / not_imbricate 'imbricate, with gliding surface anteriorly' 'imbricate anteroposteriorly, but interdigitate laterally', shouldn't this character be scored – instead of 0 for Breagnathair?

9) In Dataset 3, I am not sure why character 54 Medial frontal pillar, relationship to subolfactory process, is scored 0 instead

of missing for Breugnathair.

10) In Dataset 3, character 221 presence of lateral conch is scored as missing in Breugnathair, which I think is correct but it is in contradiction with character 188 of dataset 2.

11) In Dataset 3, subdental shelf/gutter is scored as pronounced for Breugnathair, whereas in Dataset 2 it was scored first absent (character 373 state 1) and then pronounced (character 374, state 1). According to this, I now realise that regarding what I said in my point 6 above, the score that is possibly wrong is that character 373 should be 0, and then character 374 is fine.

12) In Dataset 3, character 724, again, osteoderms are scored 0 (not imbricate), when they should possibly be scored -.

13) If in Dataset 3, as it is stated in the text, new character 789 corresponds to the presence/absence of gastralialia, I do not understand how this was scored in the matrix, because all extant squamates and at least those fossil squamates preserving a portion of the postcranium should have been scored 1 (absent), whereas most taxa (including extant squamates known to lack gastralialia!) are scored ?. Additionally, many snakes are scored 0.

Possibly with exception of the last point, which might hint at something that needs revision, I regard these as minor potential issues that do not necessarily mean that the phylogenetic analyses should be run again, but might require some further explanations in the text.

Other points that might deserve attention:

1) I wonder what is the favored interpretation of the authors regarding the position of parviraptorids as either stem squamates, stem snakes, or a less defined toxicoferan? It does not matter if the phylogenetic analyses provide three different placements, the authors can have a preferred interpretation of the observed morphology in the fossils. On the other hand, they could possibly have a look at the synapomorphies supporting each of the results, because some characters are more clearly distributed across phylogenies than others and might help in tipping the balance towards one interpretation or another. I am not requesting a deep discussion of characters on a one by one basis, but a quick look at the characters (which can be easily mapped, and it might be worth providing as additional supporting figures) could provide additional information that seems to have not been exploited so far. I recommend that the authors have a look at the characters supporting the node corresponding to parviraptorids, the node corresponding to their sister taxon (which might contain characters that parviraptorids do not share with them), and the node corresponding to parviraptorids+their sister taxon.

2) There is no mention to why *Diabrophis* and *Portugalophis* are not included in the phylogenetic analyses. They seem to be quite relevant to the discussion, and their inclusion in the newly erected *Parviraptoridae* might be even questioned if not supported by the phylogenetic analyses, so if their fragmentary nature or any other reason (e.g. difficulties in accessing the material) is what precluded their inclusion, this should be explicitly stated in the text. A

3) One important consequence of reinterpreting parviraptorids as non-related to *Toxicofera* would be that it would not only remove the oldest records of (stem) snakes (which was the main point of Caldwell's 2015 study), but of toxicoferans in general. This is important for tracing the timing of origins of such groups. *Dorsetisaurus*, which might be oldest potential toxicoferan left, is not mentioned in the text, but surprisingly appears on the stem of squamata in the only dataset that includes it (Dataset 1, fig. 7A). I must admit, however, that a clade including *Hongshanxi*, *Eoscincus*, *Dorsetisaurus* and *Parviraptorids* is highly unexpected, and its position on the stem of Squamata is puzzling, because most (all but *Hongshanxi*) of them have been regarded as well nested in crown groups of lizards. I wonder if the tip-dating of Bayesian analyses has something to do with this change of position, as all of them are roughly contemporaneous and rather old taxa. This is another place where knowing the particular characters that support the group in the corresponding tree could prove interesting.

4) The diagnosis states that the vertebrae are procoelous in parviraptorids, but Evans (1994) referred amphicoelous and nothocordal vertebrae to *Parviraptor*, and interpreted that an ontogenetic series showed a transition between the amphicoelous and nothocordal vertebrae of juveniles and the adult procoelous morphology. I understand that the new specimen, which is interpreted as an adult and shows clearly procoelous vertebrae adds little to this point, but I would like to know if the authors regard Evans (1994) referral of amphicoelous vertebrae to *Parviraptor* as still valid. The only sentence of the manuscript that seems to tangentially provide an opinion on this is "Vertebrae of NMS G.2023.7.1 are procoelous (Figs 1C–I, S5), as in other adult parviraptorids (8,9 Evans 1994; Caldwell et al. 2015).", where one can indirectly imagine that vertebrae may not be procoelous in juveniles. I would suggest adding a sentence confirming this, because someone relying entirely on what is written in the present manuscript alone could discard the possibility that amphicoelous vertebrae belong to a juvenile parviraptorid unless they read previous papers. Also, the manuscript would benefit from some more data regarding the similarities and differences between the adult vertebral morphology of parviraptorids and those of snakes. Would the morphology of parviraptorids distinct enough to allow identification of isolated vertebrae? I am thinking about the identification of material coming from screen-washing.

I want to congratulate the authors for this huge work and for the finding of an extremely interesting specimen, and I hope that my comments help in improving the manuscript.

Referee #4

(Remarks to the Author)

The origin of snakes is a big question in macroevolution, including the relationship of snakes to other clades and the origin of the snake body-form. Parviraptorids have been reported to play a critical role in this context, yet many uncertainties about them remain, not least of which is the status of the type material. The authors present an important new specimen (and taxon) that helps to clarify some outstanding issues concerning this small group that could be highly relevant to understanding the evolutionary history of snakes, and more broadly, of a major squamate clade, *Toxicofera* (or not).

I think the paper is clearly written, its anatomical interpretations generally robust, and the broader conclusions approached with appropriate caution. The data associated with the paper (MorphoSource deposition of CT scans) will enable others to study these important specimens (the new one and type material of other taxa).

I have some suggestions for improvement.

(1) Most importantly: It's ultimately disappointing that the authors do not use this opportunity to delve more deeply into the other specimens that lie at the heart of the issue of the genotype, *Parviraptor*, of the new family. At least they make the CT scans available, which is helpful. At the same time, the authors offer in a throw-away statement "Given that the squamate bones on both Purbeck blocks all belong to parviraptorids (NHMUK PV R 8851; NHMUK PV OR 48388), they most likely also belong to single individuals". Since Caldwell et al. (2015) considered these specimens NOT to be individuals, I think that a more detailed treatment in the supplementary information is also necessary for the logic of the paper. Note that the authors write in the introduction that their specimen "resolves these issues" (which exactly?), seemingly referring to mosaic evolution.

I agree with the authors' summation of the arguments by Caldwell et al. (2015) for the attribution of elements originally referred to *Parviraptor estesi* by Evans. The former group wrote: "Since the type maxilla of *P. estesi* is clearly that of a snake we expect any other elements of the same taxon to be similarly 'snake-like' in morphology" (Supplementary Note 1). On the other hand, the authors state that "They retained only the left maxilla of NHMUK PV OR 48388 within *Parviraptor estesi*, on the basis that it is 'clearly that of a snake'". I don't think that's accurate. Caldwell et al. (2015) didn't exactly "retain" it but rather restricted the holotype to that element (the reasons aren't important).

The authors also state in Supplementary: "As with NHMUK PV OR 48388, all squamate bones on the NHMUK PV R8511 block are substantially similar to the corresponding bones of NMS G.2023.7.1 (Fig. S2), indicating that they belong to parviraptorids." This also does not seem to be sufficient.

Finally, the identification of a "snake-like" "suboptic shelf of the frontal" by Caldwell et al. (2015) in *Parviraptor estesi* should be addressed in more detail.

(2) The pterygoid of their new specimen is extremely strange (Fig. 3A). I am not aware of a pterygoid in which the central portion (around the *_epi_*pterygoid fossa, not *_ecto_*pterygoid, as in Fig. S1) so strongly dorsally. In crown *Xantusia*, I think this happens, but otherwise? That being said, the pterygoid does not seem so strange (or to have such a flexure) in Fig. S1. This should be clarified, especially since there's no break or missing portion in the pterygoid in Fig. S1 but there is in Fig. 3A.

Furthermore, on p. 5 - the "position of the maxillary facet on the jugal shows that the maxilla extended just posterior to mid-length of the orbit" - This statement must be buttressed by stronger anatomical arguments. Why could the jugal bend not be located far back in the orbit? This does not seem secure to me.

(3) I think the authors should offer a better justification for the proportions. The "lowness" of the skull, I understand, is reasonable given the maxilla of the genotype, but on what basis do the authors reconstruct the skull of the present specimen as "low" ?

Some smaller issues that should be looked into and addressed, as appropriate.

I would consider citing (and potentially using) Gauthier et al.'s (2020) definition of Pan-Squamata. Note that those authors considered that parviraptorids could be stem squamates.

According to the Reptile Database (Uetz et al.), there were over 12,000 species in September 2024, not 10000 in the abstract.

Under "included taxa" (note "." not ",") I'd write *Breugnathair elgolensis* "gen et sp. nov."

On the lizard-like proportions, the authors are referring specifically, it seems to vertebral count and limb size. I don't disagree with their conclusions, but I would include more here. (They inexplicably focus on *Pseudopus*.) On the precloacal vertebral count, note that not only *Pseudopus* but essentially all limbless (or nearly so) squamate clades show this: amongst relatives also *Ophisaurus* spp., *Ophiodes*, *Anniella*, but also *Chamaesaura* spp. (Lang 1991). Basically, the only limbless squamates (outside of *Serpentes* and *Amphisbaenia*) to exceed 80 presacral vertebrae are certain members of *Pygopodidae*, *Dibamidae* and *Acontinae* (Hoffstetter & Gasc, cited by authors).

p. 5 - The vertebral count, the authors seem to take, is 24, just as preserved. I think a better justification here is necessary, given the absence of so many cranial (and limb) elements. Why can the count not have been higher?

How was the fibula identified?

p. 6 - The authors write of "xantusiid anguimorphs" - What was really intended here? Also, "scincomorphs" occurs in the paper, but the term is not generally employed nowadays since the group is considered paraphyletic.

Vomerine teeth were also described by Smith and Habersetzer (2021) for *Paranecrosaurus feisti* (cf. p. 6) and by Rieppel et al. (2007) for *Eosaniwa koehni*, so *Pseudopus apodus* is not the only "crown-group squamate" (p. 10) that possesses them.

p. 7 - Why "LRST" for the lateral aperture of the recessus scala tympani? Gauthier et al. (2012) introduced LARST, which the authors (of course) need not use, but since it exists... I'd also indicate that the "broad" crista interfenestralis refers to its anteroposterior breadth.

"otooccipital" - Evans (2008, cited) uses "otooccipital"

p. 8 - The authors discuss the absence of tooth replacement in Portugalophis and _seem_ to draw a contrast to LeBlanc et al. 2020 (cited), but this should be stated more clearly.

Teeth as in "Heloderma horridum" - why specifically this species? Don't all Heloderma look similar?

Concerning the zygosphenes, the authors write that "This condition is widely distributed among squamates...." Here cite Gauthier et al. 2012 (char. 468). The state in NMS G.2023.7.1 appears to correspond to state 2 of that character; as the authors state, a zygosphene is well-developed but not like in snakes.

p. 9 - Concerning a "distinct hypapophysis", it's worth pointing out that Gauthier et al. (2012, char. 465) distinguished here a hypapophysis that has "fore and aft margins", which I think makes it clearer.

"results from different datasets leadS"

p. 10 - maybe "_undisputed_ stem squamate Bellairsia" ?

"varanid-like overall body and head proportions" - Please explain this more clearly. Why specifically "varanid"? Just the maxilla? Certainly the vertebral count is not as in varanids.

What is a "snake-like dental ecomorphology"?

The posterior process(es) on the parietal: Gauthier et al. (2012) treated the character commonly seen in Gekkota (char. 95) as distinct from that commonly found in Scincoidea (char. 97). Please elaborate, discuss or clarify.

Articulation of the cervical intercentra on the hypapophysis - this is not the case in "anguimorphs" generally, as implied.

p. 11 - "posterior to the fenestra exochoanalis" - please check

On the presence of a "distinct, medially projecting palatine process of the maxilla" - This feature is so widespread in squamate groups that I'd not bother citing any particular species (but citing Deep Scaly can stay).

Fig. S5 - instead of labelling "procoelous", I'd label here the morphological features (say, cotyle and condyle).

Macropredation - what does this mean? The term crops up from time to time, but it's not, I think, in widespread use. While it's not critical for the results of the paper in question, it should still be clarified. How does it relate to Greene's (1983) analysis of snake evolution? Losos & Green (1993) noted that even most Varanidae took mostly invertebrates most of the time. Smith & Habersetzer (2021) emphasized that it is not necessarily the case that the most common prey will dictate the morphology of the predator. But certainly in snakes, this should be discussed more clearly.

"whare" instead of "share many features" (p. 40).

I'm not aware that "syntype" is a possible original assignment of specimens in the 21st century. If Caldwell et al. (2015) meant that a bunch of specimens that they considered to be different individuals (from different lineages) were inaccurately ascribed to the same individual, "syntype" still doesn't seem to be the right word for it.

Please explain the "postparietal process" (p. 41), as this term does not appear in the main text and has not (to my knowledge) otherwise been used.

Nomen dubium status of *Eophis underwoodi*. There are two main characters of relevance here: the size and shape of the subdental lamina of the dentary." I think that this section is not sufficiently elaborated. As I understand from the ICZN, "nomen dubium" indicates ignorance: a taxon cannot be distinguished from another valid taxon. On p. 42 the authors refer to parviraptorids as showing "slow growth" - how was the rate of growth established? It was not discussed in the main text (see also p. 42 on *Portugalophis lignites*). Ontogenetic size variation was also not discussed in the text. Additionally, the authors refer only to the figure provided by Caldwell et al. and state that "differences ... were not quantified." It seems that if the authors wish to sink that taxon, they should provide the missing measurements and not just refer to the failure of another author to measure.

Material from the Kilmaluag Fm "were not relocated during current work." Can this be elaborated on?

References:

Gauthier, J. A. & de Queiroz, K. (2020) Pan-Squamata. pp. 1087-1092 in K. de Queiroz et al. *Phylonyms: A Companion to the Phylocode*. CRC Press.

Greene, H. W. (1983). Dietary correlates of the origin and radiation of snakes. *American Zoologist*, 23(2), 431-441.

Lang, M. (1991) Generic relationships within Cordyliformes (Reptilia: Squamata). *Bulletin de l'Institut Royal des Sciences Naturelles de Belgique*, 61, 121-188.

Losos, J. B., & Greene, H. W. (1988). Ecological and evolutionary implications of diet in monitor lizards. *Biological Journal of the Linnean Society*, 35(4), 379-407.

Smith, K. T., & Habersetzer, J. (2021). The anatomy, phylogenetic relationships, and autecology of the carnivorous lizard "Saniwa" feisti Stritzke, 1983 from the Eocene of Messel, Germany. *Comptes Rendus Palevol*, 20, 441-506.

Uetz, P. (2016). The reptile database turns 20. *Herpetological Review*, 47(2), 330-334. [or some newer reference]

Version 1:

Reviewer comments:

Referee #1

(Remarks to the Author)

The revised version of the manuscript represents a significant improvement, and I commend the authors for their thorough effort in addressing the reviewers' suggestions. I have only one additional minor correction to suggest: in Figure S3E, the labels for the postfrontal and prefrontal facets appear to be inverted and should be corrected.

Referee #2

(Remarks to the Author)

I read the complete and extensive rebuttal letter. I think all 4 reviewers agreed on the excellent work done by Dr Benson and collaborators. As a herpetologist interested in pale-herpetology, I anticipate this paper would be of high impact, not only in the herpetological and paleontological community, but also any other person interested in the evolution of vertebrates. This fossil with its contrasting phylogenetic positions highlight some interesting character combination that document the enigmatic origin of snakes. I also agree with the authors that this fossil is a fossil toxiciferan, although the possibility of being a Stem-Squamate cannot be ruled out. Current phylogenetic data sets seem to place many enigmatic groups at the base of Squamata, but this seems to be a limitation of the sampling and fossil record.

At this point I have no more comments, I think the authors did a tremendous effort for including most of the suggestions by reviewers, even many of mine that I suggested to be voluntary.

Referee #3

(Remarks to the Author)

The authors have dealt with all my requests and clarified any doubt raised in the previous round of reviews. The authors' diligence in incorporating not only my requests, but also those of the rest of reviewers, has resulted in a clearer text and more robust manuscript. The addition of descriptions for those elements for which they were missing is also acknowledged. I'm happy to recommend the publication of this manuscript in its present form. Please, just consider the following list of extremely minor observations.

Line 225 (and most or all other instances the otooccipital is mentioned). I think that, from the response to referees, the spelling intended was otooccipital (as used in Evans, 2008), but it now reads oto-occipital in most instances. If otooccipital is finally used, please note that in fig. S7A the label reads otoccipital.

Line 361. differs instead of differ?

Line 362. In the new sentence "and the presence of vomerine teeth differs from almost all crown squamates bar some living and extinct anguimorphs" check the word bar

Line 589: If V.F. is for Vincent Fernandez, who already is one of the authors of the manuscript, so shouldn't be thanked here

Line 609: regarding c.f. *Parviraptor estesi*, this is written as cf. *Parviraptor estesi* in other parts of the ms.

Line 1412: otooccipitals?

Figure S10, ages for dashed lines still missing in this figure (they were added in the rest).

Line 1633: Regarding "Only a three, anterior caudal vertebrae have been identified on NMS G.2023.7.1", the a between only and three should be removed.

Finally, I guess references format will be double-checked at some point, but I noticed a few journal names that are not abbreviated to the standard abbreviation or not abbreviated at all. Here is a short (non-exhaustive) list:

Line 426: Nature Communications to Nat. Commun.

Line 455: J. of Herpetol. to J. Herpetol.
Line 457: Journal of Herpetology to J. Herpetol.
Line 473: Science Advances to Sci. Adv.
Line 477: Acta Pal. Polonica to Acta Palaeontol. Pol.
Line 509: Nature Communications to Nat. Commun.
Line 522: Scientific Reports to Sci. Rep.
Line 527: Journal of Morphology to J. Morphol.
Line 543: Italics missing from microlanius.
Line 555: dot missing after Natl
Line 566: Journal of Systematic Palaeontology to J. Syst. Palaeont.

Referee #4

(Remarks to the Author)

I am generally satisfied with the revision. The new descriptions and study of other parviraptorids greatly enhances the present contribution. There are only a few small issues remaining.

line 672: "considered" not "consider"

Figs. 3, S4 ,S6: the authors here use "pineal foramen" whereas in the text, they write "parietal foramen". Although the former term regrettably occurs in Oelrich's (1956) extraordinary anatomical work, I think the latter is accurate (e.g., Evans, 2008, The Skull of lizards and tuatara, in Biology of the Reptilia, already cited). The organ occupying the space is the parietal organ (parapineal), not the pineal.

Line 1620: double period

In measurements, sometimes (most especially in NMS G.2023.7.1, Dorsal vertebrae) the authors present figures to the hundredth of a millimeter. That level of precision does not seem justifiable; I'd reduce to the tenth of a millimeter.

Thanks for sending our manuscript to review, and thanks also to the referees for their time in reviewing our work, and the high level of diligence and constructive commentary they provided. Their comments have certainly improved the manuscript.

We have outlined the changes we made to address their comments below. We aimed, in all cases, to make the evidence clear. Moreover, we spent quite a lot of time to carefully review phylogenetic scores, including adding *Dorsetisaurus* to our analyses. We also added a concise description to the SOM (as SOM2), added many panels to the Supplementary figures, and expanded on the taxonomic reasoning regarding other parviraptorids. Note that it isn't possible for us to cover every anatomical aspect that we'd like to in a short-format paper like the current one (we'd planned to publish a long description separately), so we had to focus on specific aspects. Nevertheless, we did try to do our best here and will continue to do so.

Thanks again for your ongoing consideration of our work for publication.

Referees' comments:

Referee #1 (Remarks to the Author):

The manuscript, entitled "Mosaic anatomy in an early fossil squamate," presents the description of a new Middle Jurassic squamate, *Breugnathair elgolensis*, from the Isle of Skye in Scotland. The specimen is confidently assigned to "parviraptorids," a group previously known only from highly fragmentary material, mostly comprising isolated cranial elements. Parviraptorids have long occupied a contentious position in discussions of squamate affinities and, more recently, in debates concerning the origin of snakes. Initially interpreted by some authors as stem squamates or anguimorphs, these fossils have more recently been linked to snakes due to a handful of jaw and dental characters, and proposed as a means of bridging a critical temporal gap in the early snake fossil record, extending their presence into the Jurassic. However, the limited and poorly preserved nature of most available specimens has significantly constrained the robustness of such interpretations. This issue has been further complicated by recent claims that the only relatively complete specimen attributed to this group may, in fact, be a chimaera—comprising cranial and postcranial elements from different, and possibly distantly related, taxa. Conversely, critics of a snake affinity have emphasized that reliance on a few snake-like traits—while disregarding postcranial anatomy that often appears inconsistent with snake morphology—casts doubt on the relevance of parviraptorids to snake evolution, leading to renewed scrutiny of their phylogenetic placement.

In this context, the new specimen described in the present manuscript constitutes a major advance. It provides key anatomical elements that decisively resolve previous uncertainties, effectively falsifying claims of chimaeric associations in known parviraptorid material. The authors argue that *Breugnathair* exhibits a striking mosaic of anatomical traits, combining primitive features, snake-like dental and jaw structures, and head-body proportions reminiscent of varanids. This combination is interpreted as strong evidence of widespread homoplasy and ecological experimentation during the early evolutionary radiation of squamates. The manuscript successfully addresses persistent conflicts of anatomical interpretations and contributes critical new data to the ongoing debate on squamate diversification and the origins of snakes.

Given the significance of the material, the quality of the anatomical analysis, and the broader implications of the findings, I fully endorse the results presented in this study and strongly recommend its publication in *Nature*, pending minor revisions and incorporation of the suggestions detailed below.

Hussam Zaher

We thank the reviewer for his positive comments

Lines 92, 111. Change *Diabolophis* to *Diablophis*.

Done throughout

Line 192 "present IN stem lepidosaurs, ...

Done

Lines 223-225: The authors may consider citing Zaher et al. (2022), who reviewed the loss of the metakinetic joint in snakes in the context of the braincase anatomy of the stem-snake *Sanajeh indicus*. Including this reference would provide additional context and strengthen the discussion of cranial kinesis evolution in early snakes.

Thanks. Good suggestion. We have added that.

Lines 231-234: Contrary to the statement presented by the authors, the development of a deep lamina of the

prearticular that walls the adductor fossa medially appears to be restricted to crown-Alethinophidia. In contrast, early fossil snakes such as *Najash* and *Dinilysia* retain the plesiomorphic condition of a shallow adductor fossa, similar to that of non-snake squamates (Garberoglio et al., 2019; Zaher & Scanferla, 2012). I suggest revising this portion of the text.

Thanks for pointing this out. We have removed this statement from the text

Lines 237-240: a pleurodont tooth implantation was also reported in stem-snakes *Najash* (Zaher et al., 2009) and *Sanajeh* (Zaher et al., 2022), and in crown-snake scolecophidians (Zaher and Rieppel, 1999).

We have added ‘...and some stem-snakes (Zaher et al., 2023)’.

Line 267 change Hypopophysis to hypapophysis.

Done

Line 736: change parviraptorids to parviraptorids, and toxicoferances to toxicoferans.

Done

Lines 736-738: The sentence “Because analyses of this matrix returned parviraptorids as toxicoferans, we ran three separate analyses of this dataset using different phylogenetic hypotheses for Toxicoferan ingroup relationships” would benefit from clarification. Specifically, could the authors elaborate on what is meant by “different phylogenetic hypotheses”?

Thanks. The text here now reads: “Because analyses of this matrix returned parviraptorids as toxicoferans we ran three separate analyses of this dataset, using topological constraints that represent different phylogenetic hypotheses for toxicoferan ingroup relationships: (1) snakes constrained as sister to anguimorphs; (2) snakes constrained as sister to iguanians; and (3) anguimorphs and iguanians constrained as sister taxa, to the exclusion of snakes”.

Were the three analyses conducted under topological constraints reflecting alternative toxicoferan relationships?

Yes. Correct. We clarified this now (see above).

And how was that accommodated with the larger constraint for the ingroup relationship of squamates mentioned in lines 745-748.

We have clarified this now, in the same place (formerly lines 745-748): “...by specifying a series of partial constraints matching those shown by 4 Talanda et al. (2022, extended data fig. 6). The resulting backbone constraint is consistent with the phylogeny of ref. 3 (Burbrink et al. 2020), with trichotomies representing areas of uncertainty among recent analyses (i.e. enforcing no specific relationships between the three constituent groups). For example, *Gekkota* and *Dibamidae* in an unresolved trichotomy with the clade including all other squamates. The toxicoferan clade (*Iguania*+*Anguimorpha*+*Serpentes*) was left as an unresolved trichotomy in analyses of datasets 1 and 2, but constrained to represent three different phylogenetic hypotheses of toxicoferan ingroup relationships for three separate analyses of dataset 3 (explained below).”.

Please specify the methodological approach used to implement these constraints and how they were derived.

We have clarified how the constraints were derived now (see above). For implementation: they are a series of partial constraints implemented in MrBayes, using the ‘constrain’ command multiple times to set up a series of partial constraints. We don’t explicitly state how that is done. However, there is only one way to do this within MrBayes and that information is evident in the scripts, which we have made available. So our analyses are easy to replicate.

Line 886: ..., and so it WILL not be discussed further ...

This line has been deleted based on another referee’s comment.

Lines 1014-1015: change *Diabolophis* to *Diabolphis*

Done

Referee #2 (Remarks to the Author):

Revision of Mosaic anatomy in an early fossil squamate by Benson et al.

This paper describes an incredible new fossil that sheds light to 1) Early evolution of Squamata, 2) the complex nature of the new defined family Parviraptoridae.

This paper is of interest for a general audience and is clearly presented and illustrated. The authors made an effort to test the position of this family by retrofitting this new taxon in three existing data sets.

I have some comments that can be considered to improve the paper, but the authors are free to include them or not.

1) The preservation of Breugnathair is very similar to the original block of Parviraptor. I am not an expert on taphonomy, but maybe the authors can add a section of taphonomy and discuss the similarity of the preservation of these taxa? Squamates are generally known by isolated elements and some rare articulated fossils (hard rock, or amber embedded), but it is curious the disarticulated, but complete nature of these fossils. Could similar depositional environment of the two fossils be interpreted as a similar ecological environment for Parviraptorids?

This is an excellent suggestion, and we have added a short discussion of the taphonomy of NMS G.2023.7.1, which is also intended to strengthen the argument that the specimen represents one individual.

We have added/edited text:

“This preservation is similar to that of specimens of *Parviraptor estesi* (NHMUK PV OR 41388) and c.f. *Parviraptor estesi* (NHMUK PV R8851), from the Early Cretaceous Purbeck Limestone Group (9 Evans 1994)”.

“NMS G.2023.7.1 is isolated from other skeletons recovered from the same bedding surface by at least two metres distance. Those skeletons belong to mammals, amphibians and fish, but not squamates. Thus, it is unlikely that any squamate elements reported here represent distinct squamate individuals or species, transported from adjacent areas of the lagoon floor”.

“The presence of large numbers of easily transported Voorhies Group 1 elements (e.g., vertebrae, ribs; Stoetzel et al. 2012) indicates that the skeleton was subjected to currents that were neither strong enough to remove these elements, nor strong enough to transport allochthonous Group 2 and 3 elements (i.e., limb bones and skull elements; Stoetzel et al. 2012) from other individuals into the bone scatter before it was buried. Other non-squamate vertebrate elements such as bone crumbs and a tritylodontid tooth are present in the region of the skeleton, but these Group 3 lag elements are most likely to have been present on the lagoon floor when the skeleton was deposited, and similar material is very common throughout vertebrate-bearing levels of the sequence and around other skeletons”.

It is worth remembering, however, that parviraptorids are also found as isolated elements in unconsolidated clays at Kirtlington Quarry Oxfordshire; in the lignitic beds of Guimarota in Portugal; and in the Morrison Formation. The key thing linking all parviraptorid localities is that they preserve small vertebrate remains in a fine matrix and relatively calm depositional conditions.

2) I made a comment on the phylogenetic part, and I still think that the authors exhausted the resources to test the position of Parviraptorids. I still think the data provided in Caldwell et al. (<https://www.nature.com/articles/ncomms6996>) should be either explored in the analyses or at least comment why these were not included.

Thanks for this suggestion. We reviewed the scores, and in-text comparisons of Caldwell et al. several times during the work, and tried to include relevant anatomical statements in our work – although we didn't clearly flag those. In our revision, we looked specifically at the scores again, and included a section in our description that addresses them, explaining why we have a different anatomical view, in some cases. We incorporated revisions to the phylogenetic scores, and added a concise discussion of this to SOM1.

3) Can you provide also a dorsal view of the articulated skull?

We have included this now in the SI (Fig. S6).

After reading the article, I ended a little bit disappointed that the position of Parviraptorids might remain still uncertain, but it is also common that fossils that show mixed traits are found in such alternative positions. *Oculudentavis* could be an interesting fossil to add, as it exhibits a similar behavior (Basal or with mosasaurs), at least in the Gauthier et al data set (Modified in Meyer et al.)

We're in the same boat! We spent several years trying to establish decisive evidence for either hypothesis. And we would have liked to confidently support a single hypothesis. In the end it seemed better to be transparent about the uncertainty. And that seems important scientifically. Recognising that trait evolution can be messy is important because it should provide a warning against dismissing characters or elements that don't fit a particular paradigm.

Oculudentavis is included in all three analyses. For us, it always comes out as a stem squamate, or thereabouts. It's interesting because it also shares some features with parviraptorids e.g. the long, low maxilla.

Specific comments:

Line 23: 11989 according to the Reptile data base
Changed to “almost 12,000”.

Line 28: although the ICZN don't have a rule that prevents writing species names in the abstract, it is generally not advisable to do so, cause it makes uncertain the place in the publication where the new species is described

While we agree with the concern of the reviewer, different journals have different rules on this. Given that we only describe one new species in the current work, this should not cause a problem.

Line 30: maybe say, The new taxon is place in a new Squamate family, Parviraptoridae, an... Or something like that.
We have followed this suggestion

Line 34: add: using different data sets
Added, as "analyses of multiple datasets"

Line 37, maybe convergence sound better than homoplasy?

We have added a statement about convergence, rather than replacing the word homoplasy. Convergence is one of several processes that could result in homoplasy. Under one of our phylogenetic hypotheses (parviraptorids as stem squamates), convergence is important. Under the other (parviraptorids as stem snakes), their mosaic of traits results from the retention of some surprising plesiomorphies that were then lost independently among the different toxiciferan groups. For that, 'convergence' is not quite accurate, but 'homoplasy' still works. This part of the text now reads: "These findings indicate high levels of homoplasy and evolutionary experimentation during the initial radiation of squamates, and the potential importance of convergent morphological transformations during deep divergences".

Line 43: Some molecular estimates of this divergence to 250 million years, or the end of the Permian.

See: <https://www.nature.com/articles/s41586-020-2561-9>

Thanks for this. The figure of 250 Ma here refers to the squamate-tuatara divergence, hence to the origin of the squamate stem lineage. Our figure of 190 million years referred to estimates of crown-squamates [no change made].

Line 51: Add *Megachirella wachtleri* to this list?

<https://www.nature.com/articles/s41586-018-0093-3>

We're focussing here on definite squamates whose positions within the crown (or just outside it) remain uncertain – which is basically the case for most Jurassic taxa. *Megachirella* might be a stem squamate or stem lepidosaur. So it is slightly different to what we're trying to give examples of here.

Line 71: Since the position of snakes in the combined analyses is generally stable within Toxicofera, I think the main issue is the mosaic nature of Parviraptorids.

We have clarified this text now: "This may provide evidence of mosaic evolutionary changes along the snake stem lineage, if parviraptorids are indeed stem snakes. Alternatively, it may indicate that parviraptorids are not closely related to snakes, but independently evolved some snake-like features."

Line 75, instead of constrains use maybe use: narrows down their phylogenetic position. Constrain in systematics refers mainly to ad hoc decision to force some groups .

We see the reviewer's point, especially as we have used constrain in our phylogenetic section. We have changed 'constrains' to 'limits'

Line 84: Since Lizard is pretty much a name that apply to Squamates (See Camp 1923), perhaps don't include Lizard-like squamates. Can you could just say Limbed squamates?

Thanks for the suggestion. We have deleted lizard-like here in the Diagnosis, and simply use "squamates (or stem squamates) with well-developed limbs". We do still use 'lizard-like' in one later instance. But there we are specific that this means with well-developed limbs. We considered alternatives to 'lizard-like', but nothing else seemed adequate and evocative to convey the point there.

Line 85: smooth instead on unsculptured?

We have left this as unsculptured. Some types of sculpturing could be described as smooth. So unsculptured seems to describe what we're trying to communicate more directly.

Line 98: something seems wrong on this sentence, it is defining the Formation and location, but also that is of special scientific interest. Is that the full name of the locality?

The locality is called "the Elgol Coast Site of Special Scientific Interest". A Site of Special Scientific Interest (SSSI) is a formal term in the UK for an area of outstanding importance (e.g. like the Jurassic Coast of Dorset). This is similar to saying e.g. "Yellowstone National Park". However, we can see that this could be confusing to read so we changed it to: "Elgol Coast SSSI (Site of Special Scientific Interest)". That also helps clarify later uses of "Elgol Coast SSSI" in the text.

Line 113: Are you adding a justification for the nomen dubium below, if so, maybe add See below?

Yes, that justification is in the Supplementary online materials 1 (SOM1). The text reads "Note that we regard Middle Jurassic *Eophis underwoodi* (8 Caldwell et al. 2015) as a nomen dubium (SOM1; Fig S3)".

Line 125 methods: are these references to sessions required by the journal?

Yes – listed at the end of the main text.

Line 133: make sure you clarify the numbers of vertebral regions.

We have made corrections to this section (and other places where this is discussed, including Supplemental Dataset S2) after reviewing all the material again.

Line 153: Is this difference on onset of humeral epiphyses, also seen in other limb bones? Could this be a fracture?

Forces strong enough to fracture a humerus would likely do so at a weaker point, rather than through the articular head unless the epiphysis was unattached or weakly attached. Non-fusion of limb bone epiphyses is consistent with the lack of co-ossification of pelvic components (and late development of the vertebral condyle in specimens from Kirtlington). The other limb bones are not as well preserved at their ends either.

Line 155: maybe change to basicranium?

Thanks for the Suggestion. 'Braincase' is quite widely used in vertebrate anatomy and thereby seems accessible to readers. Based on your comment, we considered changing it to 'endocranium' ("basicranium" is a term specifically meaning parabasisphenoid+basioccipital). But that seems unnecessarily technical. We'd like to keep this as 'braincase', if possible.

Line 158: A good reference for this would be: Stephenson NG. 1962. The comparative morphology of the head skeleton, girdles and hind limbs in the Pygopodidae. Journal of the Linnean Society (Zoology). 44: 627-644.

I also see your point, of all Gekkotans, Lialis is more comparable, however the similarities are just superficial. Lialis has smaller and more teeth, narrower skulls, different snouts, different palates, and different basicranium, just to cite a few.

Yes, we agree that the similarities are superficial. We have changed this to "similar to the skull proportions of..." to clarify that only that aspect is similar.

Line 169: True, but comparisons would make more sense to Early Pan-Gekkotans, such as Norellius and Eichstaettisaurus who have the foramen.

We have added: "and present in other squamates, including possible stem gekkotans (e.g. 27 Hoffstetter 1964)".

Line 170, see my comment on smooth. Also most gekkotans have smooth parietals see: Elizabeth Glynn, Juan D Daza, Aaron M Bauer, Surface sculpturing in the skull of gecko lizards (Squamata: Gekkota), Biological Journal of the Linnean Society, Volume 131, Issue 4, December 2020, Pages 801–813, <https://doi.org/10.1093/biolinnean/blaa144>

We agree, of course, that most gekkotans have smooth (or unsculptured) parietals. We prefer to say 'unsculptured' because it implies a comparison to the sculptured surface of other taxa. Our challenge in this paper is keeping within the length limits. So we have to be quite focussed as to what comparisons we include, and focussed on the ones that were more decisive in our analyses. Unsculptured parietals is a plesiomorphy that doesn't bear closely on the main hypotheses for where parviraptorids fit.

Line 188: Norellius also have palatine teeth.

Hmm. This could be a useful observation. However, we haven't found description of that in published descriptions. Conrad and Daza (2015) mention only pterygoid teeth, and Meyer et al. (2023) have scored them as absent. But let us know if we have this incorrect (and ideally send photos).

Line 207: Maybe use non-ophidian squamates, which seems to be more widely used.

We considered this when writing the original submission. The choice is between technical language (non-ophidian) and more accessible language (non-snake). One practical matter is that if we write 'non-ophidian' then we should define 'ophidian' earlier in the work somewhere. In the end, we opted to use 'snake' where it didn't really matter what node in the snake total-group we were referring to, and then stem snake or crown snake when referring to those specific parts of the tree. We prefer this, so we opted to keep it for now. But we will change it of course if editors insist.

Line 207: basicranium

Thanks, see above (braincase is widely used in this context among vertebrates).

Line 209: to be consistent with the figures, use alary process?

Thanks. Done.

Line 211: This is a great difference with snakes

Indeed it is

Line 213: I cannot tell if this is lack of definition or poorly preserved.

It is quite well preserved. The structure is not pronounced. One thing that will help with this: people will be able to access all the 3D models too.

Line 219: Maybe mention that this is also the case of many head first burrowers eg. amphibaenians, dibamids,
Added as suggested

Line 238: add this reference after Edmund: Zaher H, Rieppel O. 1999. Tooth implantation and replacement in squamates, with special reference to mosasaur lizards and snakes. American Museum Novitates 3271: 1-19.
Done

Line 283: The selection of data sets is appropriate to test the position of Parviraptorids, however, I have a recommendation: here you should either write a sentence of why the scores for the maxilla of Parviraptor done by Caldwell et al. (<https://www.nature.com/articles/ncomms6996>) and added to the Gauthier et al data set were not considered, are these not compatible with the data set of Meyer et al? Perhaps you can do an additional run including these scores?

Maybe this could be added to the supplementary material? The authors might disagree with this idea, but I know from personal experience that the group of Caldwell states all the time that their data is ignored. If this paper is published, what is going to happen is that they will re-score the data and provide a new analyses. It is hard to predict the results of this, but they can use argument to that Parviraptor is just a snake.

Thanks. This is a very good point. It's important that we at least discuss these. We have added discussion of these scores to SOM1.

It's worth bearing in mind that we find equivocal support for parviraptorids as either stem snakes or stem squamates. Some authors might prefer one hypothesis over another. We expect that. So it is inevitable that some people will argue they are stem snakes. Probably some people will also argue they are not. But this will be a fruitful scientific discourse, we'd hope.

Line 285: reword: But we recovered this group in different positions in the squamate tree.

We want to refer specifically to phylogenetic hypotheses, because it seems important that they are hypotheses and not firm knowledge. Hence "we found support for multiple different hypotheses of their relationships to other squamates" [no change made].

Line 494: ?

This was a reference we added when filling in the data summary sheet after submission. Given that we expected to re-number the references during revisions, it didn't seem worth spending the time to re-number them before that. It's done now.

Line 581: You stated that there are 32 vertebrae preserved, but from these 24 are attributed to presacral. Here you estimated 30 presacral, are the 2 additional sacral or caudal?

We have clarified this now, after re-counting: "Thirty-two vertebrae are preserved, of which 24 are definite presacrals, three are possible presacrals, three are definite caudals and two are possible caudals or sacrals (Fig. S4; Supplementary Data 2). This gives a minimum presacral count of 25-28 (including the missing atlas). The preserved materials indicate that at least seven–eight cervicals (six preserved, plus the atlas and one cervicodorsal) and 17–21 dorsals (17 preserved, plus the cervicodorsal and three possible dorsals) were present.

Also 30 presacral is in the higher end of many limbed squamates, groups, this number excludes Iguanians, limbed cordylids, Teiids and Gymnophthalmids. See: Daza JD, Bauer AM, Stanley EL, Bolet A, Dickson B**, Losos JB. 2018. An enigmatic miniaturized and attenuate complete lizard from the Mid-Cretaceous amber of Myanmar. *Breviora* 563:1–18

The estimated number of 27-30 is just to make the body length estimate. It's not supposed to be firm comparative data. For comparisons, for space reasons we're focussing on the point here that the number of presacrals is much less than the numbers seen in limbless squamates, but close to the numbers in limbed squamates. We are not suggesting it belongs to any one of these additional groups, and none of the phylogenetic analyses suggest this. So although this is accurate descriptive info. It doesn't bear directly on the questions we're asking. Although 27-30 may be at the higher end for limbed lizards, 27-30 is not outside the presacral range for many groups, according to Hoffstetter & Gasc (1969): geckos (23-29), xantusiids (26-29), iguanids (22-28), cordylids (24-27), gerrhonotids (27-30), lacertids (23-29), teiids (24-27), xenosaurs (26-27), shinosaurs (27), limbed anguids (29-36), helodermatids (33-35), lanthanotids (25-36), and varanids (28-30). Moreover, amongst early fossil taxa, *Ardeosaurus digitellus* (27), *Eichstaettisaurus* (31), *Jucaraseps* (31), unnamed Daohugou lizard (27), *Dalinghosaurus* (27) ...

Referee #3 (Remarks to the Author):

E. Conclusions: As a summary of my assessment of the paper results, I can say that I agree with the recognition

of Breugnathair as a new genus of parviraptorid and with the erection of parviraptoridae. The authors provide three alternative positions for parviraptorids: 1) stem-snakes (which would confirm the position proposed by Caldwell et al. 2015); 2) forming a polytomy with Anguimorpha and Serpentes, which would not regard them as stem-snakes but would confirm them as toxicoferans anyway; and 3) as stem-squamates. Although several decisions made through the manuscript seem to suggest they favour a non-stem-snake position for parviraptorids, the truth is that the final section of the manuscript is somewhat equidistant between possibilities. I think that being slightly more assertive in defending the stem-squamate position would have been clearer to the reader (if that position was actually favoured by the authors).

Thanks. We thought about this for a long time during the preparation of the paper. The evidence is really on a knife-edge to the point that, scientifically, it seems right to say that we can't distinguish between these hypotheses (although we can rule out many others, two remain after that). Based on an objective reading of this result, we consider it valuable and transparent to be open about the conflicting results so as to stimulate discussion and further work. The paper adds a lot of data on this enigmatic group, which we hope people agree is important.

The importance of the paper beyond the simple addition of a new genus of fossil squamate relies on 1) the ancient age of the deposits where it was found, because there are few confirmed squamates of this age, and only one contemporaneous genus (described by the same authors) is known from a more complete specimen; 2) parviraptorids, claimed to be stem snakes by Caldwell et al. (2015) after separation of elements associated in the same block as belonging to different taxa, and selection of only the snake-like bones, are here shown to represent instead squamates with a mosaic of features, some of which are ?convergently shared with snakes. The truth is that the authors are a bit inconsistent in their interpretation, because some decisions made seem to reveal that they favor interpretation of parviraptorids as stem-squamates (or at the very least, as forms not closely related to snakes), whereas other parts of the manuscript seem to try to remain equidistant between the three options (stem-squamate, stem-snake, or more broadly toxicoferan, see my comments below).

Our preference was to present the evidence as it stands. Most aspects of the skull and skeleton suggest a stemward position. Given that previous works have made the arguments for a stem snake position, based solely on jaw material, we wanted to be sure that these contradictory features were clearly enumerated here. However, in the cold light of evidence, we couldn't come decisively down on either side (but noting – we can rule out a lot of other positions, that previously were up in the air). We've tried to make sure that our phraseology does not come across as inconsistent, in the revision.

To give some idea of how we feel about various matters:

-Parviraptorids have a mosaic of features. We feel that this strongly rejects some of the taxonomic assertions of recent works, and undermines what might once have looked like decisive support for stem snake affinities.

-Once this is known, it is difficult to muster strong support for a specific phylogenetic hypothesis. Nevertheless, the number of possibilities is small (either as stem squamates or early toxicoferans and potentially as stem snakes thing Toxicofera).

Hopefully these points are communicated successfully in the revised ms.

Although I personally would favor interpretation of parviraptorids as not closely related to snakes, mainly in the view of the presence of well-developed limbs and other characters described by the authors, what worries me more is this slight lack of consistency. Note that removing parviraptorids from the snake stem would have deep consequences for our understanding of the evolutionary origins and timing of divergence of not only snakes, but also of toxicoferans (snakes, iguanians and anguimorphs) as a whole, because parviraptorids represented the oldest fossil record of Toxicofera.

Dorsetisaurus, which would be the oldest toxicoferan if parviraptorids are removed from the clade, is not mentioned in the manuscript.

This is a good point. We've carefully scored *Dorsetisaurus* into all three matrices now. *Dorsetisaurus* is part of a separate, ongoing project. So we have some good data on it. We don't want to focus on *Dorsetisaurus* in our ms. But you can see for interest that it comes out in a polytomy (with various other fossil taxa) among crown squamate groups (Talanda matrix; Fig. S7A), or as a stem iguanian within Toxicofera (Meyer matrix; Fig. S7B; Also Zaher & Smith matrix Fig. S8 – but note a lot of other fossils fall in this position).

The implications of this particular view, where parviraptorids are not toxicoferans or, at least, not stem-snakes, concern anyone involved in the evolution of squamates as a whole. It also hints at the importance of mid and late Jurassic squamates as examples of morphologies that do not easily fit any of the crown groups, something that is not new but is shown more clearly in the specimen described here than in others.

Yes, we hoped that people reading the paper would pick up on this important point. Parviraptorid anatomy is very uncomfortable with respect to current understanding of anatomical changes during deep squamate evolution. It is more striking here than in other taxa that have been described. Snakes are about as far away from the squamate stem as you could get.

F. Suggested improvements: Although I regard the study as very well executed and of great interest, not just to paleoherpetologists, but also to those interested in the evolution of Mesozoic faunas in general, the manuscript is not free of places that could get benefited of some review. More details are provided below, but aspects to be improved include: splitting the results of phylogenetic analyses shown in fig. 4A into different trees (or, alternatively, adding a warning about the fact that parts of the depicted topology are not 100% consistent with the obtained results and that the figure should be regarded as an informal tree).

Thanks. It's true that the main text summary tree could be potentially misleading. We've added a clear statement to the figure caption now, and the full trees of course are available in the SI. The new text in the figure caption reads as follows: "Summary of phylogenetic results provided in (Figs S7–S9), with general topology adopted from Fig. 8A (Iguania, Anguimorpha and Serpentes form a polytomy in other analyses)".

Additionally, the readers should be referred to the actual trees in figures S7-S9 to interpret the results;

We have made this as clear as possible now, in the second paragraph of the Phylogenetic Results, in paragraph 5 of the Discussion, and in the figure caption of the phylogenetic summary figure.

adding descriptions of elements of *Breugnathair* that are discussed in the text, but not described;

A new descriptive section has now been added to the supplementary online information, as SOM2.

adding some more weight on the importance of the possible reinterpretation of parviraptorids as unrelated to snakes;

We have added a final line in the phylogenetic part: "As a candidate stem snake, the parviraptorid *Breugnathair* may be the oldest crown toxicoferan. However, phylogenetic uncertainties suggest that it should be treated cautiously when choosing fossil calibrations for molecular clock studies".

maybe checking the synapomorphies supporting nodes of interest (or at the very least, check them for additional information that could be of interest in diagnosing *Breugnathair* in particular or parviraptorids in general, or that could help in deciding which of the alternative phylogenetic positions is more likely;

Thanks. The synapomorphies on the trees have been a major guiding force in deciding what description and comparison to include in the main text. Because we have multiple phylogenetic datasets, and they have somewhat different character sets, we've been focussing on the character states that show the clearest signals. Paragraphs 3, 4 and 5 of the Discussion summarise these traits. Basically it's dental traits and tooth-bearing bones that have snake-like features and a scatter of other bits of anatomy that have stem-squamate like features. We tried to capture this in our text.

To sum up, I strongly recommend the publication of this work, after careful consideration of comments raised in this review that I hope will help the authors improve a very interesting and strong manuscript.

We thank the reviewer for their support and helpful review

Specific comments:

Abstract. I am not sure that the expression early diverging toxicoferan is unambiguously conveying what the authors mean. They clearly mean a toxicoferan that is recorded early in the fossil record, but an early diverging toxicoferan can also be interpreted as one that diverges early along the evolution of toxicoferans (i.e. a stem toxicoferan) which is clearly not what they mean here. They might want to rephrase to avoid such ambiguity.

Thanks. Good point. We have changed this to "early toxicoferans".

There is some sense of contradiction in the fact that the authors seem to be much more cautious in their interpretation of the group placement (with e.g. sentences like "This may provide evidence of mosaic evolutionary changes along the snake stem lineage, or that parviraptorids are not closely related to snakes, but independently

evolved some snake-like features”).), than with the election of the name. What I mean is that in some parts of the manuscript it is rather clear that they do not favor a stem-snake position, but in others they do not seem to take parts between conflicting interpretations. Keeping a name that means “false snake” is perfectly fine, but it seems to be in slight contradiction with the decision made by the authors of avoiding taking sides between the alternative positions of parviraptorids in some parts of the manuscript. If the authors favor one of the interpretations, I recommend that they express it with more confidence, in order to avoid this sensation of inconsistency.

Thanks. It's useful to see this from an outside perspective. The genus name isn't supposed to come down on the side of a particular phylogenetic hypothesis. Parviraptorids have been and referred to as 'snakes' in the some papers or their titles, and depicted as having snake-like body forms in some paleoart. But we show that the body form was not snake-like, and someone looking naively at a living parviraptorid would probably call it a lizard. The 'false' could be taken to mean several different things, which is one of the reasons we liked it. For example, the 'false killer whale' isn't precisely a killer whale, but isn't very distantly related either. *Pterodactylus*, or *Triceratops*, or *Tyrannosaurus*, are stem birds but not “birds”. A fossil that was originally recognised as a bird from jaw elements but then turned out to be e.g. a dromaeosaurid could be a 'false bird' while simultaneously being a stem bird. *Avimimus* the oviraptorosaur is a 'bird mimic' while also being a stem bird.

The title feels, again, slightly ambiguous. I would tend to interpret the mosaic nature of the anatomy of parviraptorids (particularly Breugnathair, the only for which the presence of limbs has been demonstrated), as a mixture of genuinely plesiomorphic characters (consistent with a placement on the stem of Squamata) with derived snake-like characters acquired through convergence, possibly in relation to similar feeding habits. I guess that the alternative scenario, which would involve true snake characters mixed with plesiomorphic characters (at least at the level of Toxicofera) is still possible to be interpreted as a mosaic anatomy, but I think that in that case researchers tend to interpret retention of plesiomorphic characters as something expected from an early-diverging form of a lineage (in this case, snakes), and not truly a mosaic anatomy. My point here is that, again, the authors are not being too explicit in expressing an interpretation that underlies some of the decisions made, and that should be more clearly and consistently expressed.

Thanks for the discussion of this. It is helpful, and we broadly do feel that both scenarios can be described as 'mosaic'. “Mosaic evolution” was first framed in the 1860s in relation to *Archaeopteryx* having a mix of bird traits plus plesiomorphies shared with non-bird reptiles. So actually, our view had been that the term 'mosaic' worked best when considering parviraptorids as stem snakes. But that it also could be used in the general sense of an animal having a mix of traits consistent with different placements in squamate phylogeny.

Having said that. We actually preferred different titles originally. But the journal asks for very short title lengths. So we had to use words that were very effective at capturing an idea, whilst also being interpretable to a broad audience. That's how we ended up with “mosaic anatomy in an early fossil squamate”. We hoped to capture the primary significance of the work, as phrased by the referee earlier: “...the importance of mid and late Jurassic squamates as examples of morphologies that do not easily fit any of the crown groups”. The word “mosaic” is the best word we could think of to capture this.

Line 92, Diablophis is misspelled Diabolophis here and in lines 111, 1013 and 1014.

Corrected throughout.

Line 94. The taxon name Parviraptoridae has been previously informally used (e.g. Caldwell et al., 2015; Davis et al., 2018; Head et al., 2022), also in the form of parviraptorids (e.g. Pancirolli et al., 2020; Klein et al., 2021) or “parviraptorids” (e.g. Caldwell, 2020; Fachini et al., 2020; Marjanović, 2021; Zaher et al., 2023; Paulina-Carabajal et al., 2023; Čerňanský et al., 2023) although the clade had not formally been erected. The authors state this previous use of the term in the supplementary text, but it might be worth mentioning this as a note placed between the family composition and the new genus name, or at least somewhere in the main text.

We have done as suggested and inserted a comment after the family diagnosis and a reference to the SOM.

Line 109: closing parenthesis missing after S2

Done

Line 150. Compound bone is sometimes between quotes and sometimes not, please be consistent with one or the other.

We have removed the quotes

Line 168: "The parietals share features with early-diverging squamate groups, including the presence of a parietal foramen, which is absent in snakes and crown gekkotans (Fig. 3C)." Yes, but this is rather widespread, not only shared with early-diverging squamates.

We have added "and present in many other reptiles, including possible stem gekkotans (e.g. 27 Hoffstetter 1964) and most other squamates".

Line 171: the authors probably mean xantusiid AND anguimorphs, not xantusiid anguimorphs

Thank you. This is supposed to be just 'xantusiids'.

Line 188: regarding the sentence "Palatine teeth are also present stem lepidosaurs, rhynchocephalians and some early-diverging squamates (28,31,34 Whiteside 1986; Evans & Wang 2005; Griffiths et al. 2021) as well as most snakes and some anguimorphs (30 Cundall and Irish, 2008).", they are also variably present in several iguanians (see Mahler, L., Kearney, M. "The Palatal Dentition in Squamate Reptiles: Morphology, Development, Attachment, and Replacement," Fieldiana Zoology, 2006(108), 1-61).

This is correct. Thanks – we have added that, citing Evans 2008 here, to keep reference numbers down.

Line 192: "in" is missing between "present" and "stem"

Corrected

Line 246: "The marginal teeth of NMS G.2023.7.1 are conical and strongly recurved (Figs 3A, S6), similar overall to those of other parviraptorids (8,9 Evans 1994; Caldwell et al. 2015) as well as many snakes (8,29 Gauthier et al. 2012; Caldwell et al. 2015), mosasauroids (21 Russell 1967), amphisbaenians (41 Gans & Montero 2008), the pygopodid *Lialis* (42 Iordansky 2004), and some living (e.g. 20,23,32 *Anguis fragilis* Klembara et al. 2014; *Heloderma horridum*, many *Varanus*; Evans 2008; The Deep Scaly Project 2002–2019) and extinct (43 Norell et al. 1992) anguimorphs." Although the teeth in amphisbaenians can be considered conical and slightly recurved, they are in general very different from the rest of taxa mentioned here (they are much less recurved, closer to the condition in typical non-snake squamates). If what the authors are trying is to establish a comparison of parviraptorid with other squamates with typical features of macropredators, I would not include amphisbaenians here. If the authors want to keep amphisbaenians in the comparison, they should state "some amphisbaenians", because others (e.g. trogonophids) do not have recurved teeth at all. On the other hand, they might want to add the extant scincid *Sphenomorphus muelleri* (see Kosma R. 2004. The dentitions of recent and fossil scinciform lizards (Lacertilia, Squamata) - systematics, functional morphology, paleoecology. PhD dissertation, Hannover (Germany): University of Hannover, p. 187.) as an example of a squamate with recurved teeth (this taxon approaches more closely the dental morphology *Lialis* than amphisbaenians in general).

We take the reviewer's point. Amphisbaenian teeth are often recurved but, obviously, not universally so (e.g. in the acrodont trogonophids). We have therefore deleted amphisbaenians from this list. We are struggling to cite all the references that are necessary to support the work, including suggested additions from referees. So we have to focus on comparisons to groups that have potential relationships to parviraptorids. For this reason we didn't add the comparison to *Sphenomorphus*. But thank you for the suggestion.

Line 254. No mention to the overall snake-like look of the vertebral condyle, which is in some cases almost perfectly spherical.

Thanks. We've added this text now: "The cotyle and condyle are rounded in outline, similar to snakes, but the edges of the condyle are not expanded beyond the margins of the posterior centrum, unlike in teiids, varanoids, some mosasauroids and stem and crown snakes. Instead, the diameter on the condyle is smaller than the posterior centrum, as in scincoids and procoelus gekkotans (16 Hoffstetter & Gasc 1969)".

Some cranial bones (e.g. the squamosal, postfrontal) and most limb and girdle bones are mentioned in the main text, but unless I missed them, they are never described.

Some were figured in the SOM, but we have expanded the SOM to include further description of these elements.

Particularly in the case of the postcranial elements, which are extremely interesting in showing that at least this parviraptorid did not have a snake-like (limbless) postcranium, it is weird that there is no description at all. The same applies to the vertebrae. Do the caudals show a fracture plane for autotomy? Autotomy was inferred for *Parviraptor* on the basis of the fact that many caudals represented only one half of the vertebra (presumably due to split through the autotomal plane), so CT-scans slices would be a good place to check for such planes in both *Breugnathair* and the CT-Scanned *Parviraptor* (if caudals are preserved in the latter).

You're right of course that we should include some description of the limb elements. We have now included this in the SOM, to accompany the images in Fig.1 (on the block and vertebrae); and Fig.2 girdle and appendicular elements (humerus, femur, tibia). Unfortunately, only proximal caudal vertebrae are preserved on NMS G.2023.7.1, so it is not possible to investigate caudal autotomy, nor are they present on any of the associated specimens. In the disarticulated material from Kirtlington, attributed caudal hemi-vertebrae matching presacrals, suggest caudal autotomy was present.

Line 287. Regarding the sentence "We find parviraptorids either as stem squamates (in Datasets 1 and 2; Fig. S7), or as either stem snakes or on the stem of anguimorphs plus snakes (in Dataset 3; Fig. S8–S9)", the latest statement is not entirely true based on what is reported in the complete trees (Fig. S8–S9). What the trees show is a polytomy between parviraptorids, anguimorphs and snakes. One would need to check the original trees (prior to calculation of the consensus) in order to check which are the real positions, which I suspect would be related to anguimorphs in some trees and related to snakes in others.

Thanks for spotting this. We did various revisions to the matrices (e.g. adding *Dorsetisaurus*) and now the situation is more clearly resolved in the various iterations of our Zaher & Smith analyses. Now we actually find parviraptorids firmly on the stem of anguimorphs + snakes in some analyses, and as just stem snakes in others (see SI figures).

A position on the stem of Anguimorpha+Snakes is still possible, but not necessarily true, but I cannot check because I do not have access to the trees resulting from the full length analysis. In any case, what is stated (that the trees in figs. S8–S9 recover parviraptorids on the stem of anguimorphs plus snakes) in the text is inaccurate, and this same issue is replicated when summarizing the tree in Fig. 4a.

Thanks again. See above – this is now resolved by the results of our revised analyses.

Line 297. "Current analyses therefore suggest that crown squamates are not known with certainty until the Late Jurassic (e.g. 4,7,45,48,49,50 Simões et al. 2018; Meyer et al. 2022; Talanda et al. 2022; Whiteside et al. 2022, 2024; Brownstein et al. 2023)." What about previous claims of the presence of paramacellodids in middle Jurassic assemblages? I know that they are too fragmentary to be included in phylogenetic analyses but, paramacellodids are often regarded as crown squamates (in particular related to scincoids). Do the authors think that they could actually represent stem squamates too?

Yes – we consider attribution of the fragmentary Middle Jurassic material to paramacellodids as insecure. We added a note to this effect: "(including Middle Jurassic specimens sharing dental features with the candidate stem scincoid group Paramacellodidae e.g. cf. Paramacellodidae, from the ELGOL SSSI; 15 Panciroli et al. 2020)".

I see the position of *Eoscincus* in the results of the present manuscript, but as I say in another part of my review, I am not too confident that a clade formed by *Hongshanxi*, *Dorsetisaurus*, *Eoscincus* and parviraptorids is a real clade, because these taxa are very different from one another.

Thanks for this. We weren't very happy with that either. We have carefully checked the scores of *Dorsetisaurus* and *Eoscincus* in this analysis. Now they come out in a polytomy among subgroups of the squamate crown. That seems better than the previous situation in which they were a clade. It indicates some uncertainties based on the Talanda dataset at least, as might be expected given the slightly awkward anatomy of some of these animals. They often don't fit neatly into any particular group.

Line 332. Regarding "(3) the presence of palatine teeth, seen also in squamate outgroups (31,34 Evans 1980; Whiteside 1986; Griffiths et al. 2021) as well as some early-diverging squamates (28 Evans & Wang 2005)", as the authors themselves state in another sentence, palatine teeth are also present in some anguimorphs (not mentioned here), and I also stated in one of my comments above that some iguanians variably present palatine teeth too. ... Lines 326–338. I think that of the 5 characters supposed to represent characters "shared with both snakes and with some stem squamates or early-diverging squamate groups such as gekkotans", some (3, 4, and 5) are shared also with other members of crown groups (e.g. anguimorphs, iguanians), and this might need to be stated somewhere.

Palatine teeth in anguimorphs are mentioned elsewhere in the paper (paragraph 6 of the description) and we have added iguanians there as well. However, we have now added both in this sentence as well: "...and a few extant anguimorphs and iguanians (20 Evans 2008)".

Line 545. "and" is missing between "ribs," and "a"

Done

Line 580. Details on which taxa were used for the reconstruction are given, but what about the manus and pes? Are they based on any information of the fossil? If not, is it based on any extant taxon, and which one (possibly varanids?)?

Other than a few phalangeal elements, there is no data on the manus and pes. In the reconstruction, they are loosely based on *Varanus*. However, the reconstruction uses fairly generalised extremities - we avoided reconstructing the hand and feet like those of any specialist lineage. We have added this text to the relevant Methods: "Other aspects, such as hand and foot morphology, are not informed by any fossil evidence, and should be considered as generalised".

Line 606. The question mark besides postfrontal seems to convey a lack of confidence on identification, but it is missing in another part of the manuscript (line 542).

We have removed the question mark and also corrected the side – this is a left not right postfrontal.

Line 634. Check "adjusting the 63 parameters"

Corrected. Thanks. We also tidied up the description of methods here in general.

Line 663. Check "the overlapping part. Both final"

Thanks also here. Done.

Fig. 3. The postfrontal is the only bone that is not labeled.

We have now fixed this. Thanks.

Fig. 3 caption. I would place a mention stating that some bones have been mirrored to create copies of the complementary side, and specify which ones.

Done. Thanks.

Only the new taxon and two OTU's for *Parviraptor* are included in the analyses. What about *Portugalophis* and *Diablophis*? Since they are considered parviraptorids by including them in the erected family, the reason for not including them should be reported.

Thanks. We have added explanations of this to SOM1:

"*Portugalophis* was not included in our phylogenetic analysis because it is relatively incomplete and generally duplicates information already represented by the Skye and Purbeck specimens. We hope that the descriptions and illustrations of parviraptorid bones from other localities, provided here, will facilitate identification of further cranial and postcranial elements from Portuguese localities".

"Material of *Diablophis gilmorei* was not studied further during the present work, and was therefore not included in our phylogenetic analyses, pending further description".

We also added a comment in the first paragraph of Phylogenetic Results: "Parviraptorid specimens from the Late Jurassic of the USA (*Diablophis*) and Portugal (*Portugalophis*) were not included in the analyses, pending discovery or description of more comprehensive material (SOM1)". Our job in this manuscript is to describe the new Skye material, establish the association of cranial and postcranial elements previously rejected, and test the position of the family as a whole within squamates. We don't currently have good enough data on *Portugalophis* and *Diablophis*, which are currently known only from single jaw elements and vertebrae that do not add any character state that would likely alter the position of the group on the tree.

Line 735, "toxicoferans", not "toxicoferances"; "parviraptorids" instead of "parvriaptorids"

Corrected

Line 798. Since the summary of phylogenetic results includes the positions for Dataset 3, the caption should read Figs S7-S9, not S6-S7.

Corrected

See however, my comment about the fact that I have doubts about using a single tree to show the results of analyses with different constraints (this is, the two stars labeled as Dataset 3 are marking two positions, but it is not possible that they both came from a tree with the same relationships between Iguania, Anguimorpha and Serpentes, because the point of running three different analyses was to force such relationships to be different.

Thanks. See above. We added clarification of this to the figure caption. And the summary tree is accurate with respect to a specific tree from our results, now. "...with general topology adopted from Fig. 8A (Iguania, Anguimorpha and Serpentes form a polytomy in other analyses)...".

At the very least, if the authors want to keep this particular tree as the main figure for the phylogenetic analyses results, a warning stating that this is an informal tree and any interpretation of the results should be made on the supplementary trees (figs. S7-S9) should be added.

Thanks. See above.

Line 800. "and" should possibly be deleted, or the sentence needs rewording.

This now reads: "Digital renders of the parietals in dorsal view (C), the left palatine in lateral view (D) and the right vomer in medial view (E)"

Fig. S6. The label pointing at "two replacement teeth" should be better placed in G, rather than in E, because they are better seen in the former view.

We have labelled these on both panels now.

Fig. S7-S9. These trees are apparently timescaled (as a result of the use of tip-dating), but a numerical time scale (or labelling) is not provided.

Thanks. We've included this now.

Fig. S8A. Where the constraints are stated, it should read Iguania, not Iguana.

Thanks. Corrected.

Line 882. The authors state: "Taxonomic rank (as Parviraptor or parviraptorids) is a matter of subjective judgement that cannot be resolved objectively, but also has no impact on phylogenetic relationships or their implications for squamate evolution, and so it not discussed further here." However, they formally erect Parviraptoridae in the manuscript, what seems to be in slight contradiction (they actually confirm and adopt Caldwell et al.'s interpretation, at least regarding the composition of the clade, not necessarily the interpretation of its position if in the squamate tree).

Thanks. We can see the contradiction now and have removed the "not discussed further" part. The intention of the text was to say that it was just a nomenclatural change, with no impact on evolutionary hypotheses. But perhaps that's clear anyway.

Line 885. Caldwell et al. is sometimes cited as 2015, and sometimes as 2014.

That instance has been corrected

In most cases this does not matter because such citations will end up replaced by the corresponding reference number, but the authors should check that there are no places where this could generate a problem and, particularly, make sure that these two different years are not given different reference numbers. Also, at least on one instance (line 929), Caldwell et al. is given the reference number 9, when it should be 8;

Thanks. We have fixed all these now. There were some transpositions of the citation numbers of Caldwell et al (2015) and Evans (1994).

and in another one, line 971-972, the reference has no number.

Added

Evans (1994) seems to present a similar problem, where it is sometimes given reference number 9 (most cases), but sporadically number 8 (e.g. line 190).

We have corrected these.

line 895. "and group of skull bones" should read "a group of skull bones..."

Done

Line 896. I think that the "," should be a "."

Done

Line 900. Note that in other parts of the supplementary text, the links are in place, so the call to (...links...) should probably be replaced by the actual links

Yes. That will be done when (if) the paper is accepted. Until then the links don't work! But they're all listed in Supplementary Data 1.

Line 908. "show" instead of "shows".

Done

Line 918. Check if the "but" should be removed

No – retained

Line 984. Can the authors point to somewhere that justifies the statement about slow growth? Maybe a specific part of the text dealing with the osteohistology like line 142, and/or cite Evans (1994)?

The osteohistology paragraph states 'The prolonged duration of growth and late skeletal maturity resembles that of varanids....'. We now referred back to that in this part of the SOM.

We also have changed the sentence in the SOM and elsewhere to mirror this – changing 'slow growth' to 'prolonged growth/late skeletal maturity'

Line 988. I agree with the statement about *Eophis underwoodi* as a nomen dubium, but it might be worth explicitly stating to which taxon in particular are these specimens finally referred (in the caption of Fig. S3 only parviraptorid bones are mentioned, so this seems to suggest that the intended referral in the present study for these specimens would be Parviraptoridae indet., but this would be a change from the original referral to Parviraptor cf. *estesi* by Evans (1994), which is not dealt with in the text.

We agree that the appropriate attribution is Parviraptoridae indet., pending the full description of the Morrison material and also undescribed material from Guimarota. We have altered the text to clarify this: "We therefore regard *Eophis underwoodi* as a nomen dubium and Parviraptoridae indet"

Lines 994-1004. Although the fact that they regarded *Eophis* (but not *Portugalophis* or *Diablophis*) as nomen dubium, and the inclusion of the latter in the composition of parviraptoridae, make me think that the authors support both *Diablophis* and *Portugalophis* as separate genera from Parviraptor, it would possibly be worth adding a statement in this regard.

It is rather that we are maintaining the status quo pending the description of further material of both the Morrison and Guimarota parviraptorids (currently mainly represented by incomplete single jaws). Preliminary, unpublished, information on non-dental elements from the Morrison Formation suggests it is distinct from *Breugnathair elgolensis* but a detailed description of this material is outside the scope of the current paper. Similarly, we know there to be a great deal of undescribed squamate material from the Portuguese localities (SEE pers. obs and discussion with Portuguese colleagues), and expect that publication of our specimen will lead to the recognition of further elements from Guimarota and other late Jurassic Portuguese localities that would facilitate a more comprehensive comparison. We have made a statement to the effect that we are maintaining the status quo pending more material being found/described.

Line 1033. Remodelling, not remodeling

Corrected

Figure 1. Unless I am getting it wrong, the labels for procoelous on the vertebrae are pointing at the cotyle (concave anterior surface) in Fig1E and Fig1I. Since what makes a vertebra procoelous is the presence of a condyle (convex posterior surface), I would rather point the label at the condyle of either Fig1C or Fig1F and of either Fig1G or Fig1H. The same applies to figS5D.

Thanks. We've labelled 'cotyle' and 'condyle' now, to avoid any ambiguity.

Figure 3. Is it possible that what the authors have labeled as ectopterygoid fossa is actually the epipterygoid fossa? This would fit better with the scoring of the epipterygoid in both *P. estesi* and *Breugnathair* as present (although the dorsal view of the pterygoid is not figured for the latter, and there is no mention to such structure in the text).

Yes – sorry. This is correct in Fig.S1 and was corrected for Fig.3, but the wrong version was uploaded.

Figure 4. As I said before, I have a couple of problems with the tree corresponding to figure 4A. One is that, as

currently displayed, what are labeled as stem squamates are actually drawn in a polytomy with crown squamates.

Thanks. Yes. It is useful to hear this. The figure is attempting to summarise results, rather than depict them directly (for which, see SI Figs). But obviously that's difficult when the source trees have some differences. We tried to maintain the important aspects, in light of specific hypotheses, in the summary tree. Thanks for the suggestion to add a warning about the fact that parts of the depicted topology are not 100% consistent with the obtained results (above). We've clarified the figure captions now.

This is what happens in fig. S7A, but not in the rest of trees, and drives me to a more profound problem, which is if it is acceptable to draw the multiple positions of parviraptorids in a topology that does not fit all the results. I mean, it is not the same to draw two alternative positions of a wildcard taxon in a tree that is, for the rest of taxa, fundamentally the same (at least when taxa are collapsed into large groups), than to select one of the topologies and place all the alternative positions there. The problem is that, by doing that, the authors are showing relationships that do not necessarily occur in the results.

Just as an example, the position of parviraptorids as stem snakes occurs in figures S8B and S9, but in these figures Anguimorpha and Serpentes are not sister taxa as shown in Fig. 4A.

Actually, fig. S8B shows the tree resulting from an analysis where Anguimorpha and Iguania are forced to be sister taxa, and in fig. S9, Iguania is forced to be sister of Serpentes, even if such relationships are not evident in the final tree because the movement of other taxa (possibly fossils, which are not included in the constraints) create a huge polytomy including Iguania and Anguimorpha. The topology regarding toxicoferan internal relationships shown in Fig. 4A corresponds to one of the results, the other two having different combinations of relationships for these three clades, so while one of the stars shows a true result of an analysis, the other star never happened in this particular topology (it happened in a topology that is completely different).

In my opinion, this can be fixed in two different ways: one would be to use a different tree for each of the positions;

the other one would be to place a warning in the figure stating that the different positions are placed in one particular topology, indicate which one, and place a warning that this is a simplification of the actual relationships recovered, and point towards the actual results in figures S7-S9.

We have opted for the second alternative, with more explanation in the figure caption.

Figure S6. Since the retroarticular process is scored in the matrix as present, it is worth labelling in this figure, and maybe add a mention to its presence in the text too.

We have added this now

Figures S7-S9. I am not sure if this is done on purpose, but it is weird that the OTU's in the trees correspond to the specimen number, and not to the taxon name to which such specimens are referred. This would be more justifiable in the case of specimens referred with low confidence, but it is particularly weird in the case of *Breagnathair*, because it results in trees where the new taxon is not easy to find. Why not naming them *Breagnathair*, *Parviraptor estesi* (holotype) and *Parviraptor estesi* (referred specimen number xxxx) that would be much clearer? I think that naming them with the taxon name (and maybe *Breagnathair* in bold or signaled somehow) would improve readability.

This was because the final decision on the new genus name was made after the analyses were completed. But it's correct now. We also put a star to indicate the position of *Breagnathair* now too.

Also, Anguimorpha is labeled in just one of the trees. I think that there is space to label Iguania and Anguimorpha in all trees, but if the authors do not want to do it, it might be worth being consistent and remove the label for Anguimorpha from FigS8A.

Thanks. Anguimorpha are important in that tree specifically because they are included in our description of the result. We've now labelled Anguimorpha in all the trees.

Just for figure S8A. Although Fig. 4A depicts Parviraptoridae on the stem of Anguimorpha+Serpentes, what the majority rule consensus tree shows in Fig. S8A is a polytomy between parviraptorids, anguimorphs and snakes. It is not possible for me to check the alternative positions of parviraptorids because the authors did not provide the resulting trees. It is quite important that the authors confirm that parviraptorids are actually recovered on the stem of Anguimorpha+Serpentes, and not just as sister to Anguimorphs in some trees, and sister to Serpentes in others (what would effectively place them in a polytomy with them, as shown in Fig. S8A, but not on the stem as shown in Fig.4A). Related to this, have the authors checked the allcomp consensus tree? Just to see which

relationships are favoured in the presented polytomies. This might be particularly informative for the polytomy of Fig. S8A.

Thanks for all the discussion of this. The situation is largely resolved now in our current analyses since adding *Dorsetisaurus* (and some other minor changes), because parviraptorids are not in polytomies any more.

There is not a single actual photograph of the new specimen. Although this is not mandatory, I would recommend providing at least one photograph (as a supplementary figure) of the specimen, where one can see the colors and other properties of both the bones and surrounding rock matrix.

That's true. It was too easy for us to overlook that fact after many years of looking primarily at the digital data. We have included photos now in the SI figures.

There is a mention to the deletion of *Tetrapodophis* in the statement report (and at least there is a line in the scripts of dataset3 that effectively deactivates the taxon), but this is not mentioned in the methods. Similarly, *Taytalura* is mentioned as an addition to datasets 2 and 3, but it is deactivated in the scripts of such datasets, so it is not effectively used. This should be mentioned in the methods.

Thanks for this. We have added explanations of these removals to the Methods now. For *Taytalura* – that wasn't included in the source matrices but we added it in. Later, we became concerned that it might not be a lepidosaur. So it seemed safest to remove it. We have now written: "We also did not include the proposed stem lepidosaur *Taytalura* (56 Martinez et al. 2021), because of phylogenetic uncertainties (3 Tałanda et al. 2022)". For *Tetrapodophis*, we have written now: "We omitted the candidate stem snake *Tetrapodopis* from analyses due to substantial doubts about its anatomy and phylogenetic affinities (65,66 Martill et al. 2015; Caldwell et al. 2021)".

Of the three scripts for dataset3, in the script where the constraints enforce the monophyly of snakes+anguimorphs to the exclusion of iguanians, the second line reads "constraint:snakes+anguimorphs to the exclusion of anguimorphs". The second instance where anguimorphs is written, it should read iguanians instead.

Thanks for paying attention to such things. We have corrected this now.

This has no effect on the results of the analyses because this is not a line of code, but an informative statement describing which constraint will be enforced below, but I thought that the authors might want to fix this anyways. After the authors replaced the old matrices by the final updated ones I ran short versions of the analyses and everything went smoothly. My short versions of the analyses recover results that are in line with the full-length analyses results reported in the manuscript (recovering parviraptorids as stem-snakes in some occasions), but not the polytomy reported by the authors in fig. S8A). However, this is possibly related to the fact that my analyses did not run long enough to reach convergence. This is unfortunate, because I haven't been able to check the exact positions of parviraptorids that result in this polytomy (something that the authors should be able to do with the .t files of their results). This is important because such positions might contradict what is stated in the text (see my other comments regarding this).

Thanks for trying to check this. We agree that it is an important matter. It seems to be resolved in our current results since the revisions.

I checked the scorings for parviraptorids, and I only have some minor questions:

1) At least in Dataset 1, both the postorbital and postfrontal are scored as present (and in Dataset 2 are scored as separate). A bone in the approximate position of those bones appears in fig. 3A but it is not labelled and only the postfrontal is mentioned in the text (line 543). Other places where the postfrontal is mentioned, it is accompanied by a ? (line 607). If the presence of the postorbital is scored indirectly (e.g., inferred on the basis of the presence of a facet on the postfrontal/parietal), this should be mentioned in the text. Even in that case, the authors should look for inconsistencies in how the identification of the postfrontal is expressed in the text (with or without a question mark).

The issue of the ? was discussed above, and has been dealt with (thanks). The presence of a postorbital is, indeed, inferred from facets – on both the postfrontal and on the squamosal. A sentence has been added to the main text – and this is developed further in the SOM.

2) There are other instances where the score is possibly assessed indirectly (e.g., char 68 of dataset 1, the presence of frontal parietal tabs in NMS G.2023.7.1, a specimen that does not preserve the frontals, and the presence of which is possibly deduced from facets on the parietal). The same occurs with the presence of the ectopterygoid (char. 113 of dataset 1) or the epipterygoid (char 115 of dataset1). It might be worth mentioning in the text when such indirect scorings are used, mainly in those cases (e.g. presence of the epipterygoid) in which the character state is not shared with snakes. Breugnathair is scored as lacking contact between ectopterygoid

and maxilla (dataset 1, char 362), but how is this known if none of these elements is preserved in NMS G.2023.7.1?

We have now extended the SOM to include more description. The ectopterygoid character was scored based on a facet for the ectopterygoid on the jugal, posterior to the facet for the maxilla.

3) The scored presence of frontal tabs in the parietal of NMS G.2023.7.1 (char 78, dataset 1) is not mentioned in the description of the element.

The short format of Nature papers does not permit this level of detail, and we were planning to follow with a separate descriptive paper, fully illustrated with views from each element. For now, we used parts of that to extend the SOM to include a summary description of individual elements and key features.

4) In character 123 of dataset 2: Maxilla_pivots_on_prefrontal_to_erect_fang / absent present, none of the taxa are scored 1, and only a few of them (Paliguana, Limnoscansor and Magnuviator) are scored ?. I know this is a snake character, but how the authors know about this state in Breugnathair if the maxilla and prefrontal are not preserved? The same applies to other maxilla characters.

The structure of the palatine, and its articular surface for the maxilla (ch.122), would preclude pivoting of the maxilla and, from the structure of the jugal, the maxilla was clearly not of the short morphology found in derived snakes with pivoting fangs. Nonetheless, we have changed this to ? in the reanalysis

5) How is the quadrate lateral conch (Dataset 2, character 188) scored as absent (state 1) if the quadrate is not preserved in Breugnathair?

This was an error and has been corrected in the new analyses.

6) In Dataset 2, characters 373 and 374, if Dentary_subdental_gutter_is scored absent, then character 374 (Dentary_subdental_gutter_development) should be unscorable (but see my comment 11 below).

See response to 11.

7) The coronoid is scored as present in dataset 2 (character 404) and other datasets, but this bone (or corresponding facets on other lower jaw bones) are not mentioned in the text for Breugnathair.

A facet for the coronoid was labelled on Figure S6. We've included this in the SOM description now also

8) In Dataset 2, character 583: 583 '570. Osteoderms on body (and/or tail)' / not_imbricate 'imbricate, with gliding surface anteriorly' 'imbricate anteroposteriorly, but interdigitate laterally', shouldn't this character be scored – instead of 0 for Breugnathair?

Yes – correct, this has been changed to ?

9) In Dataset 3, I am not sure why character 54 Medial frontal pillar, relationship to subolfactory process, is scored 0 instead of missing for Breugnathair.

Thanks for spotting this, as the frontal is not preserved this should be '?'. We have fixed this.

10) In Dataset 3, character 221 presence of lateral conch is scored as missing in Breugnathair, which I think is correct but it is in contradiction with character 188 of dataset 2.

Yes – this has been corrected in dataset 2

11) In Dataset 3, subdental shelf/gutter is scored as pronounced for Breugnathair, whereas in Dataset 2 it was scored first absent (character 373 state 1) and then pronounced (character 374, state 1). According to this, I now realise that regarding what I said in my point 6 above, the score that is possibly wrong is that character 373 should be 0, and then character 374 is fine.

Yes – we agree, the coding of 373 has been changed to 0 in dataset 2

12) In Dataset 3, character 724, again, osteoderms are scored 0 (not imbricate), when they should possibly be scored -.

The programme does not handle '-' but we rescored this as '?'

13) If in Dataset 3, as it is stated in the text, new character 789 corresponds to the presence/absence of gastralia, I do not understand how this was scored in the matrix, because all extant squamates and at least those fossil squamates preserving a portion of the postcranium should have been scored 1 (absent), whereas most taxa (including extant squamates known to lack gastralia!) are scored ?. Additionally, many snakes are scored 0.

Possibly with exception of the last point, which might hint at something that needs revision, I regard these as minor potential issues that do not necessarily mean that the phylogenetic analyses should be run again, but might require some further explanations in the text.

We checked this in our spreadsheet and also in the scripts. There must be some misunderstanding. This was scored with extant squamate as 1 (gastralia absent), including in snakes. So we didn't fix anything. But it seemed to be correct already.

Other points that might deserve attention:

1) I wonder what is the favored interpretation of the authors regarding the position of parviraptorids as either stem squamates, stem snakes, or a less defined toxicoferan? It does not matter if the phylogenetic analyses provide three different placements, the authors can have a preferred interpretation of the observed morphology in the fossils. On the other hand, they could possibly have a look at the synapomorphies supporting each of the results, because some characters are more clearly distributed across phylogenies than others and might help in tipping the balance towards one interpretation or another. I am not requesting a deep discussion of characters on a one by one basis, but a quick look at the characters (which can be easily mapped, and it might be worth providing as additional supporting figures) could provide additional information that seems to have not been exploited so far. I recommend that the authors have a look at the characters supporting the node corresponding to parviraptorids, the node corresponding to their sister taxon (which might contain characters that parviraptorids do not share with them), and the node corresponding to parviraptorids+their sister taxon.

Thanks for this suggestion. We're only allowed 10 supporting figures. So mapping characters isn't really possible. But we did map the characters during the work, and that informed the main text. We discussed these characters in the Discussion. Primarily jaw/teeth characters support the hypothesis that parviraptorids are stem snakes. Other characters support the hypothesis that parviraptorids are stem squamates.

In terms of providing opinion. We collectively feel that both hypotheses remain viable. So we tried to reflect that in the manuscript.

2) There is no mention to why *Diablophis* and *Portugalophis* are not included in the phylogenetic analyses. They seem to be quite relevant to the discussion, and their inclusion in the newly erected *Parviraptoridae* might be even questioned if not supported by the phylogenetic analyses, so if their fragmentary nature or any other reason (e.g. difficulties in accessing the material) is what precluded their inclusion, this should be explicitly stated in the text.

We have added an explanation of that now, which was also requested by other reviewers. They were not included in the analysis because (a) they are currently too incompletely known (and the isolated jaws/vertebrae add no data to the family diagnosis) or incompletely described. Our main concern is the positioning of parviraptorids within squamates as a whole and nothing in the current descriptions of the Morrison and Portuguese material would add relevant data. They are discussed in the SOM. We have added some extra text to explain our position.

3) One important consequence of reinterpreting parviraptorids as non-related to *Toxicofera* would be that it would not only remove the oldest records of (stem) snakes (which was the main point of Caldwell's 2015 study), but of toxicoferans in general. This is important for tracing the timing of origins of such groups.

We agree. The text now reads: "Current analyses therefore suggest that crown squamates are not known with certainty until the Late Jurassic (e.g. 3,6,45,48,49,50 Simões et al. 2018; Meyer et al. 2022; Talanda et al. 2022; Whiteside et al. 2022, 2024; Brownstein et al. 2023). As a candidate stem snake, the parviraptorid *Breugnathair* may be the oldest crown toxicoferan. However, phylogenetic uncertainties suggest that it should be treated cautiously when choosing fossil calibrations for molecular clock studies."

Dorsetisaurus, which might be oldest potential toxicoferan left, is not mentioned in the text, but surprisingly appears on the stem of *Squamata* in the only dataset that includes it (Dataset 1, fig. 7A). I must admit, however, that a clade including *Hongshanxi*, *Eoscincus*, *Dorsetisaurus* and parviraptorids is highly unexpected, and its position on the stem of *Squamata* is puzzling, because most (all but *Hongshanxi*) of them have been regarded as well nested in crown groups of lizards.

Thanks. This is important. We've now included *Dorsetisaurus* in all the matrices, and carefully reviewed its scores. The Talanda et al. matrix (Dataset 1) finds this in a polytomy with several other fossils, plus the stems of most major squamate subgroups (including *Iguania*, *Anguimorpha* and snakes). The other matrices found it as a stem iguanian — so within *Toxicofera*. We don't want to focus on this in our text because *Dorsetisaurus* really

requires a proper descriptive review. But it seems to play an important role influencing the phylogenetic position of parviraptorids as well.

I wonder if the tip-dating of Bayesian analyses has something to do with this change of position, as all of them are roughly contemporaneous and rather old taxa. This is another place where knowing the particular characters that support the group in the corresponding tree could prove interesting.

Thanks. Probably this is somewhere where a few incorrect scores could make a difference. We carefully reviewed the scores of *Dorsetisaurus* and we don't find support for this clade any more.

4) The diagnosis states that the vertebrae are procoelous in parviraptorids, but Evans (1994) referred amphicoelous and nothocordal vertebrae to Parviraptor, and interpreted that an ontogenetic series showed a transition between the amphicoelous and nothocordal vertebrae of juveniles and the adult procoelous morphology. I understand that the new specimen, which is interpreted as an adult and shows clearly procoelous vertebrae adds little to this point, but I would like to know if the authors regard Evans (1994) referral of amphicoelous vertebrae to Parviraptor as still valid.

Thanks. We've checked the text now and clarified, where needed e.g. "procoelous vertebrae in adults" and "Vertebrae of NMS G.2023.7.1 are procoelous (Figs 1C–I, S5), as in other adult parviraptorids (7,8 Evans 1994; Caldwell et al. 2015), although immature vertebrae may retain an open notochordal canal (8 Evans 1994)."

We do maintain that those other vertebrae from Kirtlington belong to parviraptorids We have new data even: our CT scans revealed the presence of a patent notochordal canal within the centrum, and some of the Skye specimens retain a dimple in the condyle which we regard as an ontogenetic remnant of the closed canal. The problem with including this is that the paper is for a short format journal, and we didn't want to go into so much depth about referrals of this material when it doesn't influence our analyses etc. We wanted to maintain a relatively high degree of focus, and these vertebrae require additional evidence to be presented. So we prefer not to include that because we have limited space for figures and for text.

The only sentence of the manuscript that seems to tangentially provide an opinion on this is "Vertebrae of NMS G.2023.7.1 are procoelous (Figs 1C–I, S5), as in other adult parviraptorids (8,9 Evans 1994; Caldwell et al. 2015).", where one can indirectly imagine that vertebrae may not be procoelous in juveniles. I would suggest adding a sentence confirming this, because someone relying entirely on what is written in the present manuscript alone could discard the possibility that amphicoelous vertebrae belong to a juvenile parviraptorid unless they read previous papers.

See above. And we have added clarifying text to the description in the SOM.

Also, the manuscript would benefit from some more data regarding the similarities and differences between the adult vertebral morphology of parviraptorids and those of snakes. Would the morphology of parviraptorids be distinct enough to allow identification of isolated vertebrae? I am thinking about the identification of material coming from screen-washing.

We have added more in the SOM. The procoelous vertebrae from Skye enclose a patent notochord at the centre, and some have a dimple where the canal has closed. Isolated vertebrae have been attributed to parviraptorids e.g. in the Morrison Formation. It would be hard to confuse parviraptorid vertebrae with those of snakes – especially in terms of zygosphenes development and the shape and position of the rib facets. This comparison is expanded in the revised manuscript.

I want to congratulate the authors for this huge work and for the finding of an extremely interesting specimen, and I hope that my comments help in improving the manuscript.

We thank the reviewer for their support and careful feedback

Referee #4 (Remarks to the Author):

The origin of snakes is a big question in macroevolution, including the relationship of snakes to other clades and the origin of the snake body-form. Parviraptorids have been reported to play a critical role in this context, yet many uncertainties about them remain, not least of which is the status of the type material. The authors present an important new specimen (and taxon) that helps to clarify some outstanding issues concerning this small group that could be highly relevant to understanding the evolutionary history of snakes, and more broadly, of a major

squamate clade, Toxicofera (or not).

I think the paper is clearly written, its anatomical interpretations generally robust, and the broader conclusions approached with appropriate caution. The data associated with the paper (MorphoSource deposition of CT scans) will enable others to study these important specimens (the new one and type material of other taxa).

I have some suggestions for improvement.

(1) Most importantly: It's ultimately disappointing that the authors do not use this opportunity to delve more deeply into the other specimens that lie at the heart of the issue of the genotype, Parviraptor, of the new family. At least they make the CT scans available, which is helpful. At the same time, the authors offer in a throw-away statement "Given that the squamate bones on both Purbeck blocks all belong to parviraptorids (NHMUK PV R 8851; NHMUK PV OR 48388), they most likely also belong to single individuals". Since Caldwell et al. (2015) considered these specimens NOT to be individuals, I think that a more detailed treatment in the supplementary information is also necessary for the logic of the paper.

Thanks for this insight. We have added substantially to the descriptions and comparisons of this material in the SI. In the section for each of these specimens, we discuss first the original attributions, then Caldwell's removal of some bones from the hypodigm (naming them), then: the new evidence that those bones belong to parviraptorids based on comparison to the anatomy of our new specimen. We also added images of the frontal that weren't previously included.

We didn't intend this to read as a throw-away statement. We'd hoped to communicate that new evidence from the anatomy of *Breugnathair* shows that the bones on the Purbeck blocks are consistent with the hypotheses that each represents a single squamate individual. We were surprised to read that someone could read this as a throw-away remark. Assuming we can accept the underlying anatomical observations that are provided in the current version of the ms, our reasoning is as follows: Caldwell et al's primary reason for rejecting the other elements on the Purbeck blocks, and thus arguing that they (and material from Kirtlington) were chimaeric, seems to have been that those elements (i.e. other than the tooth-bearing bones) did not fit their concept of what a snake should look like. *Breugnathair* demonstrates that elements rejected by Caldwell et al. (parietals, palatines, pterygoids, etc) have morphology that is consistent with attribution to parviraptorids. After that, there is no reason to argue that either Purbeck specimen is a mixed assemblage – especially as Purbeck lizard fossils generally are not mixed (e.g. the holotype of *Dorsetisaurus*). That line of reasoning still seems logical to us. And hopefully it is clear to you also, based on the revised text.

We have also strengthened the argument that the bone accumulation from Skye represents a single individual by adding extra comment on the taphonomy of the specimen in Methods, as suggested by Reviewer 2.

Note that the authors write in the introduction that their specimen "resolves these issues" (which exactly?), seemingly referring to mosaic evolution.

The text in question is as follows: "However, discussion of these evolutionary hypotheses has been eclipsed by concerns that some of the more informative specimens may be chimaeric associations of multiple taxa (7 Caldwell et al. 2015; SOM1). Here, we report a relatively complete specimen of an early parviraptorid that **resolves these issues**". The intended meaning is that the new specimen resolves the 'chimaera' issue by showing that parviraptorids indeed have a surprising trait combination. The trait combination is stated concisely in the abstract "It displays a mosaic of anatomical traits not seen in any living group, with head and body proportions similar to varanids (monitor lizards), and snake-like features of the teeth and jaws, alongside primitive traits shared with early-diverging groups such as gekkotans", and is further unpacked in the discussion. Do let us know if you have a different suggested phrasing for the part at the end of the introduction. We didn't want to repeat too hard on what was said already elsewhere.

I agree with the authors' summation of the arguments by Caldwell et al. (2015) for the attribution of elements originally referred to *Parviraptor estesi* by Evans. The former group wrote: "Since the type maxilla of *P. estesi* is clearly that of a snake we expect any other elements of the same taxon to be similarly 'snake-like' in morphology" (Supplementary Note 1). On the other hand, the authors state that "They retained only the left maxilla of NHMUK PV OR 48388 within *Parviraptor estesi*, on the basis that it is 'clearly that of a snake'". I don't think that's accurate. Caldwell et al. (2015) didn't exactly "retain" it but rather restricted the holotype to that element (the reasons aren't important).

We have clarified this now: “Of the elements present on NHMUK PV OR 48388, they retained only the left maxilla within *Parviraptor estesi* (as the holotype), stating that it was “clearly that of a snake” (8 Caldwell et al. 2015, supplementary materials). Based on the absence of clear snake-like features, they removed the pterygoid, parietals and palatine of NHMUK PV OR 48388 from *Parviraptor estesi*, and referred them to cf. Squamata indet”

The word ‘retain’ seems important and appropriately descriptive to us. All the squamate bones that are present on NHMUK PV OR 48288 were originally referred to *Parviraptor estesi* by Evans. Of these, Caldwell only kept one, the maxilla, within the hypodigm (as the ‘holotype’, although note this would accurately be a ‘lectotype’). This was included because all the others were removed. The word ‘retain’ captures this situation very accurately. Most bones were removed, one was retained.

Evidence of this is clear throughout that paper (Caldwell et al 2015): e.g. the maxilla listed as the holotype in their main text, and no other parts of the specimen listed as referred specimens — also shown in their Supplementary Figure 1 (referrals to *P. estesi* shown in blue – only the maxilla), and in the details of their SI.

We think that the reasons for the taxonomic decision of Caldwell et al. are important because they provide context to the reader of our work, and are central to understanding the basis for our suggestion that the Purbeck specimens are not chimaeras. Caldwell et al.’s conviction that they had identified an early snake led them to reject associated elements that did not fit their morphological hypothesis. We show instead that parviraptorids have a mosaic of snake-like and non snake-like features. This causes us to accept the associations to single individuals.

The authors also state in Supplementary: “As with NHMUK PV OR 48388, all squamate bones on the NHMUK PV R8511 block are substantially similar to the corresponding bones of NMS G.2023.7.1 (Fig. S2), indicating that they belong to parviraptorids.” This also does not seem to be sufficient.

We have expanded the SOM to help with this, providing much greater exposition.

Finally, the identification of a “snake-like” “suboptic shelf of the frontal” by Caldwell et al. (2015) in *Parviraptor estesi* should be addressed in more detail.

The subolfactory lamina is deep in the referred specimen from Purbeck – but the interpretation of a ‘sub-optic process’ is based on the supposed anterior margin of an optic foramen. A CT scan of the specimen shows that this supposed margin is not smooth, it is irregular and clearly the result of damage to the edge of the shelf. Secondly, we know from the parietals that there is no corresponding descending lamina on the parietal. This is a misinterpretation. We have added an image of the digital frontal, pointing out the relevant region. We also added an explanation in the SOM:

“The frontal of NHMUK PV R8511 has a deep subolfactory process that appears notched on its posterior margin. Caldwell et al. (8) interpreted this notch as the anterior margin of an optic nerve foramen, thereby interpreting the specimen as having a snake-like suboptic shelf. However, the CT scans of the specimen show that the supposed notch has irregular margins and is due to breakage there is no corresponding lamina on the parietal (Fig. S2).”

(2) The pterygoid of their new specimen is extremely strange (Fig. 3A). I am not aware of a pterygoid in which the central portion (around the epipterygoid fossa, not ectopterygoid, as in Fig. S1) so strongly dorsally.

Thanks. The pterygoid is broken into parts and here may have been some torsion. We adjusted the articulation of these parts in the digital image now (revised Fig. 3A).

In crown Xantusia, I think this happens, but otherwise? That being said, the pterygoid does not seem so strange (or to have such a flexure) in Fig. S1. This should be clarified, especially since there’s no break or missing portion in the pterygoid in Fig. S1 but there is in Fig. 3A.

The reviewer may not realise that these are two different pterygoids. The pterygoid in Fig. S1 is that of the holotype of *Parviraptor*, the pterygoid in Fig. 3 is that of the new Skye specimen. The Skye bone is, unfortunately broken with only a small contiguous portion between anterior and posterior halves. But the morphology is represented faithfully – other than questions about how precisely the broken parts should be arranged.

Furthermore, on p. 5 - the “position of the maxillary facet on the jugal shows that the maxilla extended just

posterior to mid-length of the orbit" - This statement must be buttressed by stronger anatomical arguments. Why could the jugal bend not be located far back in the orbit? This does not seem secure to me.

We have clarified this now, to: "The position of the maxillary facet on the jugal shows that the maxilla terminated anterior to the postorbital bar". We are confident about this because the jugal forms the ventral part of the postorbital bar, to which we can compare the position of the maxillary facet of the jugal. We now have labelled this specifically on Fig S5.

(3) I think the authors should offer a better justification for the proportions. The "lowness" of the skull, I understand, is reasonable given the maxilla of the genotype, but on what basis do the authors reconstruct the skull of the present specimen as "low" ?

Thanks for this question. It is good to have the opportunity to include the relevant information. We have added a short explanation to the Methods: "The reconstructed skull has long, low proportions. Evidence for this comes from the braincase dimensions, relative to the lengths of the combined palatal elements and mandibular elements. We allowed additional vertical height at the back of the skull to accommodate slight crushing of the braincase." Please note that crushing would have to be fairly extreme before the skull would stop being long and low. So we're actually quite confident that it was long and low in some form.

Some smaller issues that should be looked into and addressed, as appropriate.

I would consider citing (and potentially using) Gauthier et al.'s (2020) definition of Pan-Squamata. Note that those authors considered that parviraptorids could be stem squamates.

We have cited this now in the Introduction, and also replaced 'Squamata' with Pan-Squamata in the Systematic Palaeontology.

According to the Reptile Database (Uetz et al.), there were over 12,000 species in September 2024, not 10000 in the abstract.

This has been updated following one of the other reviewers

Under "included taxa" (note "." not ",") I'd write *Breugnathair elgolensis* "gen et sp. nov."

OK - done

On the lizard-like proportions, the authors are referring specifically, it seems to vertebral count and limb size. I don't disagree with their conclusions, but I would include more here. (They inexplicably focus on *Pseudopus*.) On the preloacal vertebral count, note that not only *Pseudopus* but essentially all limbless (or nearly so) squamate clades show this: amongst relatives also *Ophisaurus* spp., *Ophiodes*, *Anniella*, but also *Chamaesaura* spp. (Lang 1991). Basically, the only limbless squamates (outside of *Serpentes* and *Amphisbaenia*) to exceed 80 presacral vertebrae are certain members of *Pygopodidae*, *Dibamidae* and *Acontinae* (Hoffstetter & Gasc, cited by authors).

We were trying to write concisely, given the format. So we have to pick examples carefully. *Pseudopus* was cited as an example of a vertebral count at the lower end of the range for a limbless taxon. We have clarified this in the text now: "at the lower end up to 68 in the anguid *Pseudopus*; and more in other taxa...". The big comparison here is between limbed and limbless forms. Variation among limbless forms is large of course. However, given the format we must focus on the comparisons that provide context for our specimen specifically.

p. 5 - The vertebral count, the authors seem to take, is 24, just as preserved. I think a better justification here is necessary, given the absence of so many cranial (and limb) elements. Why can the count not have been higher?

This wasn't our intention. We wrote in the Methods that, for the body length estimate, we assumed 27–30 presacral vertebrae. Our estimate, based on the consistent distribution of vertebrae across the block, is that the likely original number was 27-30. It could have been higher – but not significantly higher if the animal is to function. Given the format, we were trying to avoid repeating text. So, here in the summary description we wrote '... are preserved', to indicate that this was a preserved count, not a full estimate.

To make this explicit, we have added some text now that reads "...even given that some vertebrae are likely missing...". And "Thirty-two vertebrae are preserved, of which 24 are definite presacrals, three are possible presacrals, three are definite caudals and two are possible caudals or sacrals (Fig. S4; Supplementary Data 2). This **gives a minimum presacral count of 25-28** (including the missing atlas)".

How was the fibula identified?

As a bone more slender than the tibia and close to the femur and pelvis. It contains no phylogenetically relevant features, so there isn't much at stake here. The text now reads "possible partial fibula".

p. 6 - The authors write of "xantusiid anguimorphs" - What was really intended here?

This should have been just 'xantusiids'. We've corrected it now.

Also, "scincomorphs" occurs in the paper, but the term is not generally employed nowadays since the group is considered paraphyletic.

As used in the past, this is a paraphyletic grouping and we agree with the reviewer that it is best avoided. Unfortunately some authors re-use it in a different context. Here we have changed it to scincoid and lacertoid to reflect taxa (e.g. some teiids and scincids) with this morphology.

Vomerine teeth were also described by Smith and Habersetzer (2021) for *Paranecrosaurus feisti* (cf. p. 6) and by Rieppel et al. (2007) for *Eosaniwa koehni*, so *Pseudopus apodus* is not the only "crown-group squamate" (p. 10) that possesses them.

It wasn't out intention to say that *Pseudopus* was the only crown group squamate with vomerine teeth. Indeed, we said they were absent in almost all crown squamates, except for three groups: "the extant anguine *Pseudopus* (32 Klembara et al. 2017), some Eocene anguimorphs (33 Rieppel et al. 2007) and the Jurassic stem scincoid *Eoscincus* (6 Brownstein et al. 2022)".

We specifically cited Rieppel et al. (2007) here, in reference to *Eosaniwa*. We said 'some Eocene anguimorphs' instead of naming that taxon specifically – because more than one Eocene anguimorph has this. In the original text, before we applied the citation limit, we also cited Smith and Habersetzer (2021). However, we had to remove it in order to fit the limit of 50 citations.

p. 7 - Why "LRST" for the lateral aperture of the recessus scala tympani? Gauthier et al. (2012) introduced LARST, which the authors (of course) need not use, but since it exists... I'd also indicate that the "broad" crista interfenestralis refers to its anteroposterior breadth.

The most important thing here seems to be that the abbreviation is explicitly defined, and then used consistently in the current work. There are precedents for 'LRST'. Evans used the abbreviation LRST in 2008 (ref.20)

We have added antero-posteriorly to the description of the crista.

"otoccipital" - Evans (2008, cited) uses "otooccipital"

Yes – corrected, it was correct on the line below.

p. 8 - The authors discuss the absence of tooth replacement in *Portugalophis* and seem to draw a contrast to LeBlanc et al. 2020 (cited), but this should be stated more clearly.

For convenience, this is the text under discussion: "Replacement pits are absent, as in the Late Jurassic parviraptorid *Portugalophis*, which has been interpreted as showing early initiation of internal tooth resorption, indicating snake-like tooth replacement (39 LeBlanc et al 2020)".

We're not contradicting LeBlanc here. These taxa lack replacement pits regardless. We thought it was important to recognise that they described a possible mechanism for teeth replacement in parviraptorids. But we are agnostic as to LeBlanc's interpretation of *Portugalophis* replacement as being snake-like as it is based on a small difference in perceived thickness of a tooth section from a CT scan. Because we have not examined the specimen itself, we wanted to leave this issue open.

Teeth as in "*Heloderma horridum*" - why specifically this species? Don't all *Heloderma* look similar?

Yes – the living species are similar. *H. horridum* was the taxon to hand for comparative observations. We have deleted the species name here because it isn't useful.

Concerning the zygosphenes, the authors write that "This condition is widely distributed among squamates...."

Here cite Gauthier et al. 2012 (char. 468). The state in NMS G.2023.7.1 appears to correspond to state 2 of that character; as the authors state, a zygosphenes is well-developed but not like in snakes.

Done

p. 9 - Concerning a "distinct hyapophysis", it's worth pointing out that Gauthier et al. (2012, char. 465) distinguished here a hyapophysis that has "fore and aft margins", which I think makes it clearer.

Done

"results from different datasets leadS"

Done

p. 10 - maybe "_undisputed_ stem squamate Bellairsia" ?

Done

"varanid-like overall body and head proportions" - Please explain this more clearly. Why specifically "varanid"? Just the maxilla? Certainly the vertebral count is not as in varanids.

We have clarified this now, in brackets: "a long, low skull without loss of limbs". It is also possible that it has a higher cervical count – similar to anguimorphs. But we aren't 100% confident so that isn't stated.

What is a "snake-like dental ecomorphology"?

We have changed this to just "snake-like dental morphology".

The posterior process(es) on the parietal: Gauthier et al. (2012) treated the character commonly seen in Gekkota (char. 95) as distinct from that commonly found in Scincoidea (char. 97). Please elaborate, discuss or clarify.

In the main text, we have added a citation to Gauthier et al. 2012, and we have discussed this more fully in the extended SOM. The two characters (95,97) are less easy to distinguish where the parietal is paired. We scored (1) for both characters 95 and 97 in this dataset, because when mirrored and reconstructed, there is a slight gap between the two postparietal (posteromedian) processes. However, in life, they may have formed a single undivided median process. As neither form is common to toxicoferans, our statement in the main text remains valid — the structure, in either form, is shared with early-diverging squamate groups.

Articulation of the cervical intercentra on the hypapophysis - this is not the case in "anguimorphs" generally, as implied.

OK – changed to 'some anguimorphs'

p. 11 - "posterior to the fenestra exochoanalis" - please check

Well spotted – this should be anterior

On the presence of a "distinct, medially projecting palatine process of the maxilla" - This feature is so widespread in squamate groups that I'd not bother citing any particular species (but citing Deep Scaly can stay).

OK. Thanks. We appreciate saving some words.

Fig. S5 - instead of labelling "procoelous", I'd label here the morphological features (say, cotyle and condyle).

Yes – this has been changed

Macropredation - what does this mean? The term crops up from time to time, but it's not, I think, in widespread use. While it's not critical for the results of the paper in question, it should still be clarified. How does it relate to Greene's (1983) analysis of snake evolution? Losos & Green (1993) noted that even most Varanidae took mostly invertebrates most of the time. Smith & Habersetzer (2021) emphasized that it is not necessarily the case that the most common prey will dictate the morphology of the predator. But certainly in snakes, this should be discussed more clearly.

We have changed such instances just to 'predatory' or 'feeding mode', to avoid this issue with the word 'macropredation'. Reviewing the papers recommended by the reviewer here (and thanks), these authors seem to agree that these sorts of dental traits, in snakes and some anguimorphs, evolved to capture larger prey, heavier prey, or more energetic prey i.e. vertebrates, even if those prey sizes aren't the predominant part of the diet. The term 'macropredation' is used more often in some other vertebrate groups, maybe not as much in squamates. We had to work hard in this revision to accommodate people's suggestions to what we could add to the paper, and what citations we should add. But the space limit is tight of course and adding detailed reasoning on the

frequency of taking large prey, or implications for early snake evolution (when we're not confident that it is a stem snake), is not possible. We hope it is acceptable if we just take it as a given that snakes, varanids, *Heloderma*, parviraptorids etc evolved teeth like this in relation to predation. That seems to be the broad consensus, even if there are nuances in the precise details.

"whare" instead of "share many features" (p. 40).

This has been corrected

I'm not aware that "syntype" is a possible original assignment of specimens in the 21st century. If Caldwell et al. (2015) meant that a bunch of specimens that they considered to be different individuals (from different lineages) were inaccurately ascribed to the same individual, "syntype" still doesn't seem to be the right word for it.

Here, we are providing context by describing what Caldwell et al. (2015) said. We've put 'syntypes' in quote marks now, to help communicate that. We wanted to be accurate, but also to avoid digging too much dirt on issues that aren't central to the scientific arguments of the paper.

Please explain the "postparietal process" (p. 41), as this term does not appear in the main text and has not (to my knowledge) otherwise been used.

Thanks for noticing this inconsistency. We have clarified this now, at its first use (in the Diagnosis): "postparietal (posteromedian) process". After that, we have ensured that it appears consistently in the text and figures.

This is the term frequently used for the posteromedian process and which is used in the diagnosis and Figure S2.

Nomen dubium status of *Eophis underwoodi*. There are two main characters of relevance here: the size and shape of the subdental lamina of the dentary." I think that this section is not sufficiently elaborated. As I understand from the ICZN, "nomen dubium" indicates ignorance: a taxon cannot be distinguished from another valid taxon.

Thanks for pointing this out. We included a more complete discussion of this now, in SOM1, including comparative measurements of the subdental lamina.

On p. 42 the authors refer to parviraptorids as showing "slow growth" - how was the rate of growth established? We have changed 'slow growth' to prolonged growth/delayed maturity, although this effectively means the same thing, the arguments for which are discussed in relation to the combination of osteohistology (large counts of growth marks) and failure of the epiphyses to fuse to the shaft (humerus).

It was not discussed in the main text (see also p. 42 on *Portugalophis lignites*).

Growth was discussed in the main text osteohistology: "The bone histology of the humerus and femur shows slow, cyclical growth similar to many similar-sized living squamates... the humeral epiphyses remain unfused to the shaft, consistent with prolonged continuation of growth".

We have made that clear in the SOM text now too: "However, size may not usefully distinguish between species given that individuals grow larger through ontogeny, and especially given the evidence of prolonged growth (e.g. growth continued over at least nine years in *Breugnathair*, see main text osteohistology; Fig. 2). Further evidence that size differences may not be useful for parviraptorid taxonomy comes from ontogenetic size variation within the Kirtlington assemblage of parviraptorid bones (e.g. vertebrae, 9 Evans 1994)".

Ontogenetic size variation was also not discussed in the text. Additionally, the authors refer only to the figure provided by Caldwell et al. and state that "differences ... were not quantified." It seems that if the authors wish to sink that taxon, they should provide the missing measurements and not just refer to the failure of another author to measure.

We agree. The problem was that we couldn't physically get to see the *Portugalophis* specimen. But we were trying to get hold of a 3D model, which we now have. That allowed us to take measurements. We have written the following: "This leaves only the relatively shallower subdental lamina as a distinguishing feature. However, this does not seem to be especially narrow in the holotype of *E. underwoodi* (NHMUK PV R12355). The ratio of the subdental lamina height below the fourth alveolus, to the straight line distance from the dorsomesiolingual point of the symphysis, to the midpoint of the posterior margin of the fourth alveolus is 0.29 in *Eophis*, 0.30 in *Breugnathair*, and 0.23 in *Portugalophis*. The ratio of the subdental lamina height below the fourth alveolus, to the straight height of the lateral surface of the dentary at the level of the fourth alveolus is 0.38 in *Eophis*, 0.35 in

Breugnathair, and 0.36 in Portugalophis. The differences between these values are small, and a robust diagnosis is not possible based on such incomplete material. We also cannot exclude that metric differences are possibly linked to small size or immaturity.”

Ultimately, using a section of edentulous dentary symphysis less than 2mm in length as a holotype is hard to justify unless the features it shows are truly remarkable. Evans (1994) did not name the Kirtlington parviraptorid precisely because there was no tooth-bearing element, or diagnostic mature element, that could provide a valid, diagnosable, holotype. However, we have discussed this a little more fully in the SOM. We agree that the subdentary lamina is thinner in the mid-dentary Kirtlington fragment, but the animal is also much smaller.

Material from the Kilmaluag Fm “were not relocated during current work.” Can this be elaborated on?

We have changed the text to read: “These were not reported with specimen numbers and have not been discussed further in subsequent literature and could not be relocated during the current work”.

References:

- Gauthier, J. A. & de Queiroz, K. (2020) Pan-Squamata. pp. 1087-1092 in K. de Queiroz et al. *Phylonyms: A Companion to the Phylocode*. CRC Press.
- Greene, H. W. (1983). Dietary correlates of the origin and radiation of snakes. *American Zoologist*, 23(2), 431-441.
- Lang, M. (1991) Generic relationships within Cordyliformes (Reptilia: Squamata). *Bulletin de l'Institut Royal des Sciences Naturelles de Belgique*, 61, 121-188.
- Losos, J. B., & Greene, H. W. (1988). Ecological and evolutionary implications of diet in monitor lizards. *Biological Journal of the Linnean Society*, 35(4), 379-407.
- Smith, K. T., & Habersetzer, J. (2021). The anatomy, phylogenetic relationships, and autecology of the carnivorous lizard "Saniwa" feisti Stritzke, 1983 from the Eocene of Messel, Germany. *Comptes Rendus Palevol*, 20, 441-506.
- Uetz, P. (2016). The reptile database turns 20. *Herpetological Review*, 47(2), 330-334. [or some newer reference]

We thank the referees for their consideration of our revised manuscript, and are pleased to receive their positive remarks. Our changes to the manuscript are included in blue type below.

Thanks,

Roger

Referee #1 (Remarks to the Author):

The revised version of the manuscript represents a significant improvement, and I commend the authors for their thorough effort in addressing the reviewers' suggestions. I have only one additional minor correction to suggest: in Figure S3E, the labels for the postfrontal and prefrontal facets appear to be inverted and should be corrected.

>>Thanks for catching that error. We have corrected this in the figure now.

Referee #2 (Remarks to the Author):

I read the complete and extensive rebuttal letter. I think all 4 reviewers agreed on the excellent work done by Dr Benson and collaborators. As a herpetologists interested in paleoherpetology, I anticipate this paper would be of high impact, not only in the herpetological and paleontological community, but also any other person interested in the evolution of vertebrates. This fossil with its contrasting phylogenetic positions highlight some interesting character combination that document the enigmatic origin of snakes. I also agree with the authors that this fossil is a fossil toxicoferan, although the possibility of bean a Stem-Squamate cannot be rule out. Current phylogenetic data sets seem to place many enigmatic groups at the base of Squamata, but this seems to be a limitation of the sampling and fossil record.

At this point I have no more comments, I think the authors did a tremendous effort for including most of the suggestions by reviewers, even many of mine that I suggested to be voluntary.

>>Thanks, much appreciated.

Referee #4 (Remarks to the Author):

I am generally satisfied with the revision. The new descriptions and study of other parviraptorids greatly enhances the present contribution. There are only a few small issues remaining.

line 672: "considered" not "consider"

>>Thanks for spotting that. It is corrected now

Figs. 3, S4 ,S6: the authors here use “pineal foramen” whereas in the text, they write “parietal foramen”. Although the former term regrettably occurs in Oelrich's (1956) extraordinary anatomical work, I think the latter is accurate (e.g., Evans, 2008, The Skull of lizards and tuatara, in Biology of the Reptilia, already cited). The organ occupying the space is the parietal organ (parapineal), not the pineal.

>>Thanks. We have changed all instances to ‘parietal foramen’.

Line 1620: double period

>>Thanks. We did find/repalce to change all such instances.

In measurements, sometimes (most especially in NMS G.2023.7.1, Dorsal vertebrae) the authors present figures to the hundredth of a millimeter. That level of precision does not seem justifiable; I'd reduce to the tenth of a millimeter.

>>Thanks. We changed this to tenths of millimetres.